# A patterned human primitive heart organoid model generated by pluripotent stem cell self-organization

Brett Volmert [1,2], Artem Kiselev [1,3,4], Aniwat Juhong [5,6], Fei Wang [7], Ashlin Riggs [1,2], Aleksandra Kostina [1,2], Colin O'Hern [1,2], Priyadharshni Muniyandi [1,2], Aaron Wasserman [1,2], Amanda Huang [1,2], Yonatan Lewis-Israeli [1,2], Vishal Panda [3,8], Sudin Bhattacharya [3,8], Adam Lauver [3], Sangbum Park [1,3,4], Zhen Qiu [5,6], Chao Zhou [7] & Aitor Aguirre [1,2] ✉

Pluripotent stem cell-derived organoids can recapitulate significant features of organ development in vitro. We hypothesized that creating human heart organoids by mimicking aspects of *in utero* gestation (e.g., addition of metabolic and hormonal factors) would lead to higher physiological and anatomical relevance. We find that heart organoids produced using this self-organization-driven developmental induction strategy are remarkably similar transcriptionally and morphologically to age-matched human embryonic hearts. We also show that they recapitulate several aspects of cardiac development, including large atrial and ventricular chambers, proepicardial organ formation, and retinoic acid-mediated anterior-posterior patterning, mimicking the developmental processes found in the post-heart tube stage primitive heart. Moreover, we provide proof-of-concept demonstration of the value of this system for disease modeling by exploring the effects of ondansetron, a drug administered to pregnant women and associated with congenital heart defects. These findings constitute a significant technical advance for synthetic heart development and provide a powerful tool for cardiac disease modeling.

Cardiovascular diseases (CVDs), including disorders of the heart and blood vessels, are the leading causes of death globally, contributing to an estimated 17.9 million deaths annually[1]. Laboratory models of the heart are used to better understand the etiology and mechanisms of CVDs in high detail. Several model systems to research CVDs are used, ranging from primary and induced pluripotent stem cell (iPSC)-derived cardiomyocyte cultures to animal models and 3D-culture systems such as spheroids and engineered heart tissues[2–4]. Nevertheless, many of these systems fail to fully recapitulate aspects of the complex nature of the human heart due to a variety of reasons, including the absence of endogenous extracellular matrix (ECM) and non-cardiomyocyte cardiac cell types, as well as the lack of

[1]Institute for Quantitative Health Science and Engineering, Division of Developmental and Stem Cell Biology, Michigan State University, East Lansing, MI, USA. [2]Department of Biomedical Engineering, College of Engineering, Michigan State University, East Lansing, MI, USA. [3]Department of Pharmacology and Toxicology, College of Human Medicine, Michigan State University, East Lansing, MI, USA. [4]Division of Dermatology, Department of Medicine, College of Human Medicine, Michigan State University, East Lansing, MI 48824, USA. [5]Institute for Quantitative Health Science and Engineering, Division of Biomedical Devices, Michigan State University, East Lansing, MI, USA. [6]Department of Electrical and Computer Engineering, College of Engineering, Michigan State University, East Lansing, MI, USA. [7]Department of Biomedical Engineering, Washington University in Saint Louis, Saint Louis, MO, USA. [8]Institute for Quantitative Health Science and Engineering, Division of Systems Biology, Michigan State University, East Lansing, MI, USA. ✉e-mail: aaguirre@msu.edu

physiological morphology and cellular organization. In addition, animal models possess distinct, non-human physiology, metabolism, electrophysiology and pharmacokinetic profiles which often do not predict human-relevant responses accurately[5].

The introduction of human-relevant models is paramount to the discovery of effective, clinically translatable solutions to CVDs. Over the last 10 years, innovations in human induced pluripotent stem cell (hiPSC)[6] and organoid[7] technologies have advanced techniques and provided platforms to better model and study human systems with increasing precision. Recently, we and others have described methodologies to create human heart organoids from pluripotent stem cells. These methods enable the study of human heart development and disease[8–10] in a dish to a degree unseen before due to their cellular complexity and physiological relevance. Yet, these systems still fall short of recapitulating important aspects of human heart development and the late embryonic human heart, such as anterior-posterior patterning, coronary vascularization and lack important cell populations contributing to heart structure (e.g., neural crest). Thus, there is a pressing need to develop more sophisticated in vitro heart organoid model systems to better understand human heart development and disease pathology.

Here, we report an advanced set of developmentally inspired conditions to induce further developmentally relevant cellular, biochemical, and structural changes in human heart organoids in a high throughput setting by complete self-assembly, bringing heart organoids one step closer to 6-10-week-old gestational hearts. The methods introduced here take inspiration from developmental and maturation timeline paradigms which elicit distinct intra-organoid cellular compositions, transcriptomes, functionalities, and metabolic profiles generated in part through a retinoic acid morphogen gradient present only in our most advanced developmental maturation strategy. We show that this protocol is reproducible across multiple human pluripotent stem cell lines, including embryonic and induced pluripotent stem cells. We also provide proof-of-concept of its utility for disease modeling and pharmacological studies by investigating the morphological and electrophysiological effects of ondansetron, an FDA-approved drug administered to pregnant women and associated with congenital heart defects[11–13], on embryonic heart development. Our methodology is highly automatable, scalable, and overall amenable to high-throughput screening approaches for the investigation of human heart development, cardiac diseases, toxicity testing and pharmacological discovery.

## Results

### Extending synthetic heart development through developmental induction

We recently detailed a protocol for the generation of self-organizing early embryonic human heart organoids which constitutes the starting step for the methodology described below[8]. Heart organoids were differentiated from hiPSC embryoid bodies to the cardiac lineage between days 0 and 7 through a timewise 3-step Wnt pathway modulation strategy, and then cultured until day 20 in RPMI/B27[8]. To examine the effect of more advanced organoid culture strategies mimicking *in utero* conditions on heart organoid development, we took day 20 early embryonic-like heart organoids and employed four different developmental induction strategies from day 20 to day 30 (Fig. 1a). These strategies are partly based on previous human and animal developmental studies[14–19] and represent gradual increasing steps in complexity relevant to *in utero* conditions (in order of less complex to more complex: control, maturation medium, enhanced maturation medium 1, enhanced maturation medium 2/1). Our control strategy represents a continuation of organoid culture in the base medium used for organoid formation, RPMI/B27. The maturation medium (MM) strategy uses RPMI/B27 with added fatty acids (an embryonic relevant concentration of oleic acid, linoleic acid, and

palmitic acid)[14] and L-carnitine to facilitate a developmentally relevant transition from glucose utilization to fatty acid metabolism characteristic of the fetal human heart. The MM strategy also uses T3 hormone, a potent activator of organ growth during embryonic development and metabolic maturation that has been shown to stimulate cardiovascular growth[16]. The enhanced maturation medium 1 (EMM1) strategy uses the same basal composition as MM but decreases the concentration of glucose to cardiac physiological levels[17] (from 11.1 mM to 4 mM to further encourage the transition to fatty acid oxidation) and adds ascorbic acid as a reactive oxygen species scavenger to counteract the increased oxidative stress. Enhanced maturation medium 2/1 (EMM2/1) strategy utilizes a combination of two different media formulations. During days 20-26, EMM2 media is utilized which uses the same basal composition as EMM1 and contains IGF-1. IGF-1 plays important roles during embryonic and fetal development in tissue growth and maturation, especially in the heart, as proven in murine and human studies[18,19]. From day 26 onwards, EMM1 media is utilized in the EMM2/1 strategy. EMM2/1 constituted our most advanced condition and mimicked *in utero* heart development to the greatest extent. More detailed descriptions of all developmental induction strategies along with concentrations of respective media formulations can be found in the Methods.

Following our original protocol[8], organoids experienced a period of rapid growth from day 0 to day 10, increasing in diameter while retaining their spherical structure (Fig. 1b), and then began to condense and become more defined up to day 30 in all maturation conditions. Organoids developed distinct elliptical morphologies after day 20, elongating as observed by brightfield microscopy and growing long diameters between 1000 and 1600 μm, with short diameters ranging from 600 to ~1000 μm on day 30 (Fig. 1b, c). Organoid area revealed similar trends for each condition, between 0.6 mm$^2$ to ~0.9 mm$^2$ (Fig. 1d). By day 30 of culture, nearly 100% of organoids in every condition were beating across five independent experiments (Fig. 1e, Supplementary Movies 1–4). Transmission electron microscopy (TEM) images indicated the presence of well-developed myofibrils and the formation of sarcomeres (Fig. 1f) in all conditions, with sarcomeres in the EMM1 and EMM2/1 conditions displaying significantly increased lengths of $1.58 \pm 0.323$ μm and $1.41 \pm 0.187$ μm, respectively, compared to Day 15 organoids (Fig. 1g). qRT-PCR revealed the expression of hallmark cardiomyocyte marker genes from day 20 to day 30 as expected (*MYL2*, *MYL7*, *MYH7*, and *MYH6*) at various timepoints of differentiation (Fig. 1h). Additionally, using a transgenic Flip-GFP human pluripotent stem cell line (Flip-GFP fluoresces only upon apoptotic cascade activation)[20], we found no significant levels of apoptosis in heart organoids from day 20 to day 30 (Supplementary Fig. 1a) and found no differences in apoptosis between any of the tested conditions (Supplementary Fig. 1b), in agreement with previous observations[8]. A 48-h doxorubicin treatment was used as a positive control and displayed high levels of fluorescence (Supplementary Fig. 1c).

### scRNA-seq of human heart organoids under developmental induction reveals cell type complexity and differences in cellular composition

To characterize the cellular and transcriptomic composition of heart organoids in each of our developmental induction conditions, we performed scRNA-seq on day 34 of organoid culture and plotted the integrated UMAP clustering for each condition (Fig. 2a). Ventricular and atrial cardiomyocytes (VCMs and ACMs, respectively), valve cells (VCs), proepicardial derived cells (PEDCs), epicardial cells (ECs), stromal cells (SCs), cardiac progenitor cells (CPCs), conductance cells (CCs), and endothelial cells (ECs) were revealed in all conditions of heart organoids. The abundance of several significant cell groups varied according to the developmental medium conditions. Control organoids were composed of 17% VCMs, 17% ACMs, 3% VCs, 17% PEDCs,

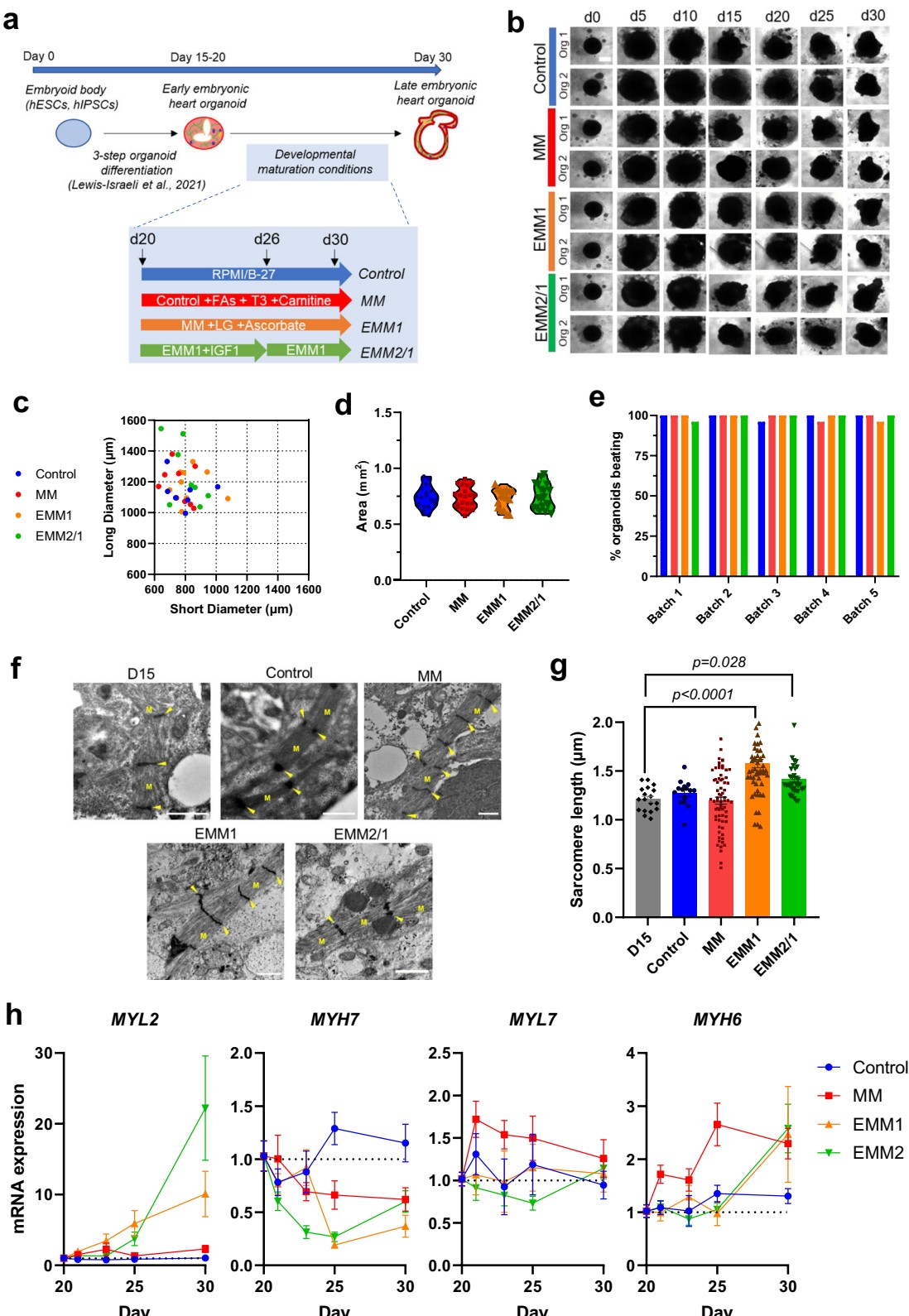

1% EPCs, 18% SCs, 10% CPCs, 5% CCs, and 1% ECs (Fig. 2b). Relative to control, MM organoids displayed an increased percentage of both VCMs and ACMs (27% and 34%, respectively), increased VCs (10%), decreased PEDCs (12%), 1% EPCs, decreased SCs (9%), decreased CPCs (6%), and decreased CCs (1%). Relative to control, EMM1 organoids contained an increased percentage of VCMs (22%), increased ACMs (31%), increased VCs (10%), decreased PEDCs (16%), increased EPCs

(4%), decreased SCs (9%), decreased CPCs (7%), and decreased CCs (1%). Relative to control, EMM2/1 organoids exhibited a decreased VCM percentage (13%), increased ACMs (20%), increased VCs (18%), decreased PEDCs (15%), increased EPCs (3%), 18% SCs, 10% CPCs, and decreased CCs (2%). Differential gene expression analyses determined signature genes which were used to identify clusters (Fig. 2c, d). ACMs possessed high expression of *MYH6, MYL7, NPPA, and GJA5*[21,22]. VCMs

**Fig. 1 | Developmental induction methods for improving human heart organoid developmental modeling. a** A schematic diagram depicting the differentiation protocol for creating human heart organoids and the media conditions for the four maturation strategies (control, MM, EMM1, and EMM2/1). Created with BioRender.com. **b** Brightfield images of organoids throughout the 30-day culture period. Two representative organoids are shown for each condition (data representative of 23–24 independent organoids per condition across five independent experiments). Scale bar = 400 μm. **c** Quantification of organoid long diameter and short diameter at day 30 of culture for each maturation strategy ($n = 7$–9 independent organoids per condition). **d** Quantification of organoid area at day 30 of organoid culture for each maturation strategy ($n = 17$–20 independent organoids per condition across two independent experiments). Data presented as a violin plot with all points, one-way ANOVA with Dunnett's multiple comparisons test. **e** Quantification of percentage of organoids visibly beating under brightfield

microscopy in each condition from 5 different organoid batches ($n = 22$–24 independent organoids in each condition across five independent experiments). **f** TEM images displaying sarcomeres, myofibrils (M) and I-bands (arrows) in day 15 organoids and in organoids from each maturation condition at day 30 ($n = 4$ independent organoids per condition). Scale bars = 1 μm. **g** Quantification of sarcomere length within TEM images ($n = 4$ independent organoids per condition, $n = 18$, 15, 69, 58, and 35 measurements for D15, control, MM, EMM1, and EMM2/1, respectively). Data presented as mean ± s.e.m, one-way ANOVA with Dunnett's multiple comparisons tests. **h** mRNA expression of key sarcomeric genes involved in cardiomyocyte maturation between days 20 and 30 of culture for each condition ($n = 14$ independent organoids per day per condition per gene across three independent experiments). Data presented as $\log_2$ fold change normalized to Day 20. Values = mean ± s.e.m. Source data are provided as a Source Data file.

displayed high expression of *MYL3, MYH7, TNNC1, and HSPB7*[21,22]. PEDCs showed high expression of *PDGFRB, SEMA3D, POSTN*, and *TCF21*[23]. EPCs shared slight similarity with PEDCs, yet also presented higher differentially expressed genes including *WT1, TBX18, ITLN1*, and *TNNT1*[21]. CCs displayed high expression of *STMN2, CHGA, SCG2*, and *INSM1*; genes that are involved in neuron growth, development and neuroendocrine signaling[24] and shared similarity with neural crest and Schwann cells identified in human embryonic heart datasets[21]. ECs possessed high expression of *PECAM1, ESAM, SOX18*, and *FLT1*[21]. SCs were identified by expression of *SOX2, ANXA4, SOX9, CD24*[21]. VCs were identified via the expression of *DLK1, ID2, DKK2*, and *WNT7B*[22,25,26]. Taken together, these results showed that heart organoids possessed similar cell types as those that are found in the 1st trimester developing human heart, and suggested, in agreement with previous studies on cardiac development[21,22], that developmental induction conditions can exert dramatic effects on the expansion and maturation of cardiac cell types to better reflect in vivo heart development.

Marker genes corresponding to various other cardiac cell types and processes were identified in all conditions, such as cardiac fibroblasts, left-right asymmetry, proliferation, first heart field (FHF) and second heart field (SHF). Genes specific for cardiac fibroblasts were identified within the PEDC cluster, showing expression of *DCN, LUM, OGN*, and *POSTN, and COL1A1*[21] (Supplementary Fig. 2). Organoids also recapitulated key genes involved in left-right asymmetry in all conditions such as *PITX2, PRRX2, LEFTY1 and PRRX1* (Supplementary Fig. 3). Additionally, organoids displayed high upregulation of proliferation markers such as *MKI67, PCNA, AURKB*, and *CDK1* in all conditions, indicating that important growth and remodeling are still undergoing at day 34 of differentiation (Supplementary Fig. 4). Cells of the first heart field (FHF) and second heart field (SHF) contribute to linear heart tube expansion and subsequent chamber formation and are important for proper cardiac morphogenesis. Various FHF and SHF markers were observed in organoids in all conditions (Supplementary Fig. 5, 6). FHF markers *HAND1, TBX5*, and *HCN4* were all upregulated in the VCM and ACM clusters for all conditions, while *HAND1* appeared to be more highly expressed by VCMs, *TBX5* more highly expressed by ACMs, and *HCN4* more restricted to ACMs in the EMM2/1 condition. Interestingly, these patterns of gene expression appear to mimic cardiac developmental principles, with *HAND1* playing a large role in overall cardiac morphogenesis, particularly with ventricular development[27], with *TBX5* being expressed in the left ventricle and atria throughout embryonic development[28], and with *HCN4* acting as a marker of the first heart field and as a transient marker of conduction precursors[29]. Additionally, genes relating to the SHF such as *GATA4, TBX1, MEF2C, OSR1*, and *WNT11* were widely expressed in heart organoids. *GATA4* was expressed by ACMs, VCMs, PEDCs and EPCs, *TBX1* was expressed in all clusters, *MEF2C* was upregulated in the ACM, VCM, and PEDC clusters, and *OSR1* and *WNT11* were expressed in the ACM, VCM, EPC, and PEDC clusters for all conditions. These gene networks in our heart organoids appear to closely mimic those observed in human heart development,

as *GATA4* has been shown to be expressed in cardiac lineage cells and to be essential in the formation of the proepicardium and epicardium[30], and with MEF2C being required for proper cardiac morphogenesis[31], and with *OSR1* being implicated in atrial septation[32]. In addition, genes related to outflow tract development such as *RSPO3, WNT5A, TBX2, and TBX3* were shown to be upregulated in various clusters for all conditions (Supplementary Fig. 7). Further, additional markers for valve development, such as *FN1, THBS1, CXCL12*, and *CCND1*[25,26] are shown (Supplementary Fig. 8). Additional quantification of *WT1, TNNT2, PECAM1, ALDH1A2*, and *TBX18* shows the individual distribution of gene expression across major clusters (Supplementary Fig. 9).

Extending these analyses, we utilized publicly available data from the Human Cell Atlas Project[21], using data from gestational day 45 (GD45) hearts and from 5 weeks to 13 weeks of gestation[22] to compare our human heart organoids to that of developing human hearts (Supplementary Fig. 10). Based on their time in culture, our human heart organoids should be closest to GD45 or 6-7 gestational week human fetal hearts (Supplementary Fig. 10a). We integrated these scRNA-seq datasets and discovered high overlap between our cell type annotations and that presented from the Human Cell Atlas Project (Supplementary Fig. 10b), with atrial and ventricular cardiomyocytes, proepicardial-derived cells (named fibroblast-like, smooth muscle cells, and epicardium-derived cells in the Human Cell Atlas Project dataset), endothelial cells, and epicardial cells displaying a high degree of clustering between datasets. ACMs, VCMs, and EPCs in the organoids showed similar percentages of total cellular makeup (13–32%, 15–26%, and 1–4%, respectively, per condition) compared to gestational human hearts[21], with ACMs, VCMs, and EPCs in GD45 gestational human hearts showing 32%, 14%, and 3.8% of total makeup, respectively. Interestingly, our valve cells mapped closely to capillary endothelial cells and our stromal cells mapped closely to immune cells, which may be due to shared gene programs surrounding endothelial-to-mesenchymal transition processes[33] (valve cells and capillary endothelial cells) or transcriptional pathways implicated in the interplay between stromal cells and immune cells during cardiac repair and regeneration[34]. Our conductance cell cluster was mapped to an unannotated cluster of cells from the embryological datasets, even though our conductance cell cluster displays similar gene expression profiles to the cardiac neural crest and Schwann progenitor cell cluster in the Human Cell Atlas Project dataset[21], which may be due to a lack of extra-cardiac signaling pathways in the organoids. We then used these datasets to compare gene expression profiles at the single-cell level (Supplementary Fig. 10c and Supplementary Fig. 11). Using the top 1000 differentially expressed genes in each dataset from the VCM, ACM, PEDC, and EPC-mapped clusters, we showed a high degree of similarity between organoids from each condition and that from embryonic hearts (Supplementary Fig. 10c), with control organoids and EMM2/1 organoids clustering closely to week 6 embryonic hearts, whereas MM and EMM1 organoids clustered closer to

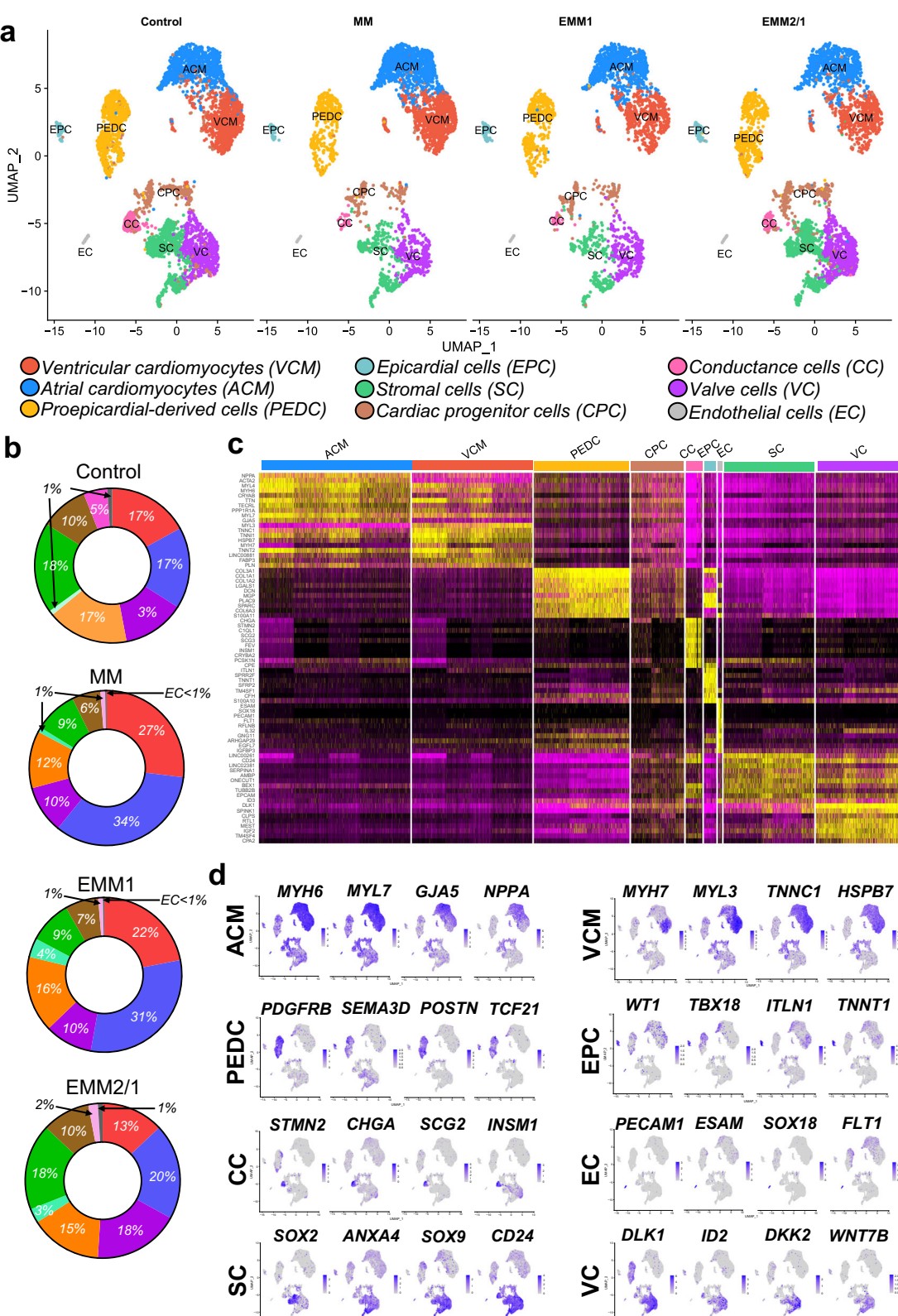

**Fig. 2 | Single-cell RNA sequencing of human heart organoids reveals distinct cardiac cell populations. a** UMAP dimensional reduction plots of integrated single-cell RNA sequencing data for each condition in day 34 organoids. Cluster identities are in the legend below. **b** Quantification of total cell count percentages per cluster. Colors of regions correspond to those found in the legend in (**a**). **c** Differential expression heatmap displaying the top 10 differentially expressed genes for all clusters. **d** Feature plots displaying key marker genes for each cluster. Color intensity represents the relative value of gene expression per gene.

weeks 7–13 embryonic hearts, which may indicate that the MM and EMM1 strategies accelerate in vitro development at a rate higher than biological development (Supplementary Fig. 10a). We also assessed individual gene expression levels and show a high degree of similarity between embryonic hearts and human heart organoids across all major clusters (Supplementary Fig. 11a, b, c, d, e, f), including hallmark genes for atrial cardiomyocytes (*MYL7*, *NPPA*, and *GJA5*), ventricular cardiomyocytes (*MYH7*, *MYL3*, and *IRX4*), proepicardial-derived cells (*PDGFRB*, *POSTN*, and *TCF21*), epicardial cells (*WT1*, *TBX18*, and *TNNT1*), endothelial cells (*PECAM1*, *ESAM*, and *SOX18*), and valve cells (*SOX9*, *UGDH*, and *FLRT2*). Organoids were shown to be committed to the mesodermal fate with null commitment toward ectodermal or endodermal fates (Supplementary Fig. 12)[8] (GSE153185). Additionally, we show that genes in the conductance cell cluster and stromal cell cluster show resemblance to embryological datasets (Supplementary Fig. 13) and show that metabolic genes across organoid and embryological datasets share similar expression profiles across cell types and clusters (Supplementary Fig. 13). To complement the above scRNAseq analyses, dot plots describing the average and percent expression of key lineage-defining differentially expressed genes for individual clusters are depicted for each developmental induction condition, illustrating the cellular complexity of our heart organoids at day 34 (Fig. 3a). As we and others have shown before[7–10], the high cellular complexity of the organoids drives self-organization and cell-cell communication. We performed computational analysis of cell-cell communication networks for key genes found in the organoids. We identified various complex receptor-ligand communication pathways within our human heart organoids in each condition (Fig. 3b). Receptor-ligand networks include JAG1-NOTCH1, PDGFRs, IGF2-IGF2R, INSR, and VEGF, among others. We also performed Gene Ontology (GO) analyses for biological process terms for each cluster corresponding to the top differentially expressed genes for each cluster in each condition (Supplementary Fig. 14, 15). To further investigate cell-cell communication networks, we utilized scRNA-seq data to highlight key receptor-ligand pairs within the organoids from each condition (Supplementary Fig. 16–19). This data highlights the ability and sensitivity of our organoids to respond to various developmental maturation stimuli surrounding cell-cell communication paradigms.

## Mitochondrial maturation and oxidative metabolism in human heart organoids under developmental induction conditions

The early developing human heart relies heavily on glycolysis for energy expenditure. As it continues to grow, it decreases its reliance on glycolysis and switches to fatty acid oxidation for the bulk of energy consumption[35]. Therefore, we sought to determine the effect that our developmental induction conditions exerted on mitochondrial growth and metabolic transcriptional activity within heart organoids. Through the addition of MitoTracker, a mitochondrial-permeable fluorescent dye, we visualized live mitochondrial content within heart organoids at day 30 of culture (Fig. 4a). Control organoids displayed few and diffuse mitochondria, while EMM2/1 organoids possessed the most developed mitochondrial content of all conditions (abundance, morphology) (Fig. 4a, b). An increasing trend of mitochondrial content in MM, EMM1, and EMM2/1 organoids (fold change of $1.73 \pm 0.10$, $2.60 \pm 0.11$, and $3.10 \pm 0.18$, respectively) was quantified relative to control, suggesting that developmentally matured organoids had an increasingly higher capacity for aerobic respiration and responded positively to maturation stimuli (Fig. 4b). TEM revealed high-magnification detail on mitochondrial presence within organoids at day 30 of culture (Fig. 4c). Compared to day 15 mitochondrial size, control organoid mitochondrial size was similar (Fig. 4d). However, mitochondrial size within MM, EMM1 and EMM2/1 organoids dramatically increased relative to that of control organoids. We employed qRT-PCR at different timepoints from day 20 to day 30 of organoid culture to explore the differential gene expression of two key OXPHOS genes in cardiac metabolic

maturation: *PPARGC1A*, a master regulator of mitochondrial biogenesis, and *CPT1B*, a critical rate-limiting fatty acid transporter element (Fig. 4e). *PPARGC1A* expression was up to 2.5-fold higher in EMM2/1 organoids from days 21 to 25 relative to control and ended at a fold change of 1.5-fold higher by day 30. Expression for MM and EMM1 organoids also exhibited an increase from days 21 to 25 relative to control, albeit not as high as EMM2/1. By day 30, expression in EMM1 organoids remained similar to control while MM and EMM2/1 organoids displayed 1.2-fold and 1.7-fold higher levels, respectively. *CPT1B* expression increased 1.5-fold at day 30 in the EMM2/1 condition relative to control, yet expression in MM remained similar or decreased for EMM1 organoids.

To investigate real-time metabolic parameters, we performed Seahorse Mito Stress Test assays with organoids in each condition at day 30 (Fig. 4f). Organoids in the EMM2/1 condition displayed marked increases in basal respiration (Fig. 4g), maximal respiration (Fig. 4h), and percent spare respiratory capacity (Fig. 4i) compared to control; aligning closely with the metabolic enhancement present in EMM2/1 organoids displayed in previously shown mitochondrial and metabolic data.

Supporting these findings using scRNA-seq data, key genes involved in cardiac metabolism were found to be upregulated in organoids from the MM, EMM1, and EMM2/1 conditions in the ACM and VCM clusters (Fig. 4j), including: *CKMT2*, a gene that encodes a mitochondrial creatine kinase and is important for metabolic efficiency and implicated in cardiac maturation[36]; *NMRK2*, a gene active in high energy states and involved in both cardiac maturation and lipid metabolism[37]; and *KLF9*, a gene related to adipogenesis and cardiac metabolic maturation[37]. We then used gene expression data from the ACM and VCM clusters in each condition to look for a wider set of metabolic markers as the organoids developed in the different conditions. We found that organoids in the EMM2/1 condition expressed much higher levels of key metabolic genes compared to control, including those involved in fatty acid metabolism, amino acid metabolism, TCA cycle, and mitochondrial dynamics (Fig. 4k). Furthermore, we performed computational transcriptomic analysis and mapping to KEGG metabolic pathways using Pathview[38] (Supplementary Figs. 20–27). In agreement with our other metabolic data, EMM2/1 organoids showed reduced activity of glycolytic complexes (Supplementary Figs. 20–23) and increased activity of mitochondrial respiratory complexes (Supplementary Figs. 24–27), indicative of progressive developmental maturation. Overall, these results suggested that developmental induction strategies can elicit distinct metabolic growth patterns and that EMM2/1 organoids exhibit the most marked recapitulation of significant aspects of cardiac metabolic growth in vitro.

## Developmental induction conditions promote progressive electrophysiological maturation in human heart organoids

The emergence and presence of the cardiac conduction system, including specific ion channels and membrane receptors, such as those surrounding calcium, potassium, and sodium currents, represent critical elements of the cardiomyocyte action potential and fetal heart development[39]. We sought to characterize the functionality of heart organoids under developmental induction conditions through electrophysiology and immunofluorescence for key markers. We assessed calcium transient activity of individual cardiomyocytes within human heart organoids at day 30 using the membrane-permeable dye Fluo-4 (Fig. 5a, Supplementary Movies 5–8). Organoids in all conditions exhibited distinct and regular calcium transient activities with varying peak amplitude and action potential frequencies (Fig. 5 b, c). Control and MM organoids presented smaller peak amplitudes when compared to EMM1 and EMM2/1 organoids, indicating less robust contractions (Fig. 5b), and presented similar beat frequencies -1.5 Hz. EMM1 organoids displayed abnormally high beating rates (-2.5 Hz) for the heart at this stage, while EMM2/1 organoids showed beat frequencies at -1–1.5 Hz (Fig. 5c). In general, and except for EMM1

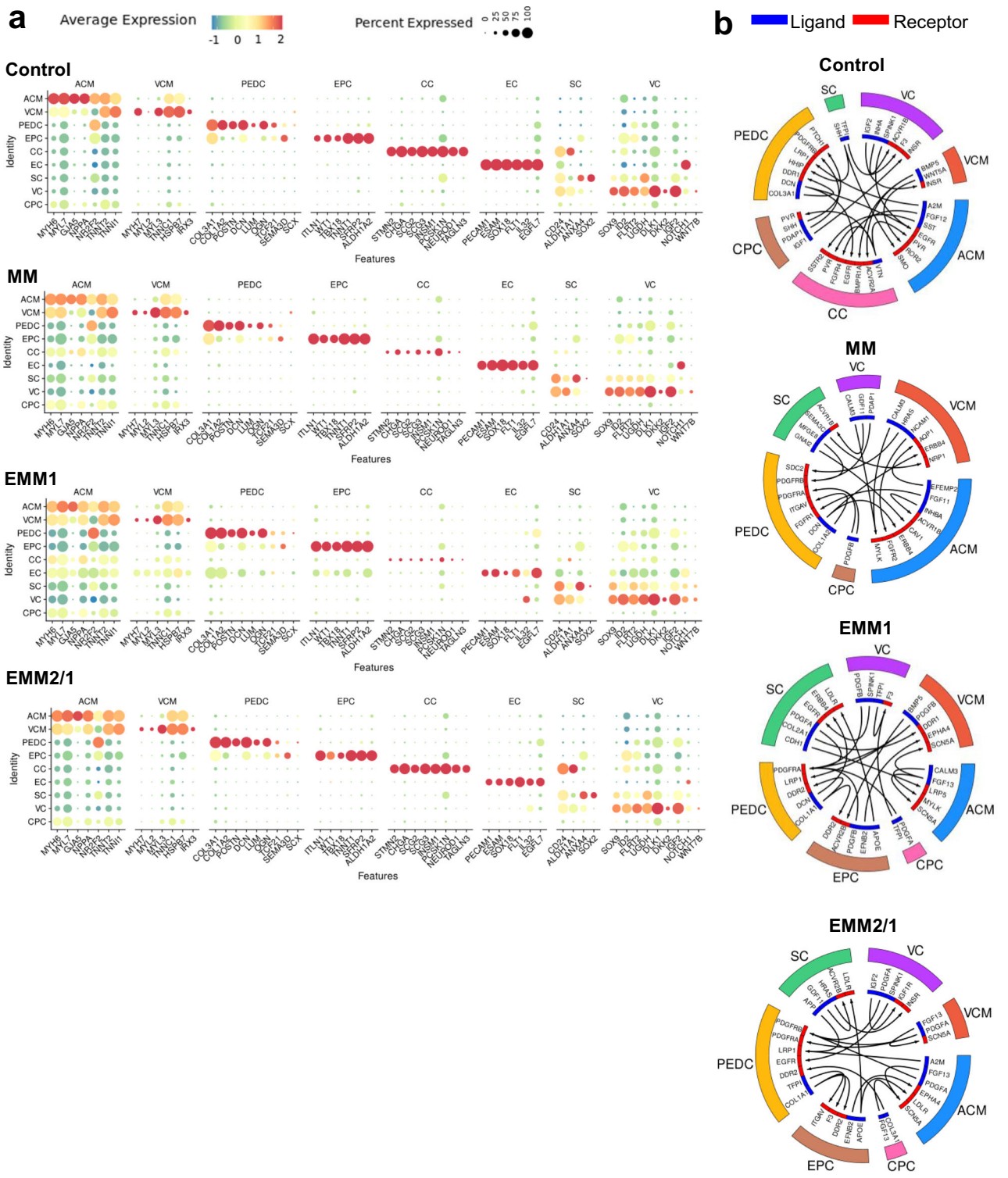

**Fig. 3 | Cluster identity and cell-cell communication networks highlight the importance of self-organization in heart organoid development. a** Dot plots of differentially expressed marker genes in each cluster for each condition. Color is indicative of the average expression level across all cells, and the size of the circle is indicative of the percentage of cells within a particular cluster that express the respective gene. **b** Visualization of cell-cell ligand-receptor communication networks for each condition. Colors of clusters (exterior) match that of UMAP projections. Ligands are indicated as blue bands and receptors are indicated by red bands. Arrows within depict pairing from ligands to receptors.

organoids, developmentally induced organoids presented beating rates compatible with what has been described for early human embryos at GD45[40] (60–80 beats per minute). Calcium traces from organoids in all conditions were shown to be reproducible (Supplementary Fig. 28).

Cardiomyocyte action potential activity encompasses the complex orchestration of various ion currents, such as calcium, potassium, and sodium, and supporting channels such as ryanodine receptors. We investigated the expression levels of various electrophysiologically-relevant genes in heart organoids and

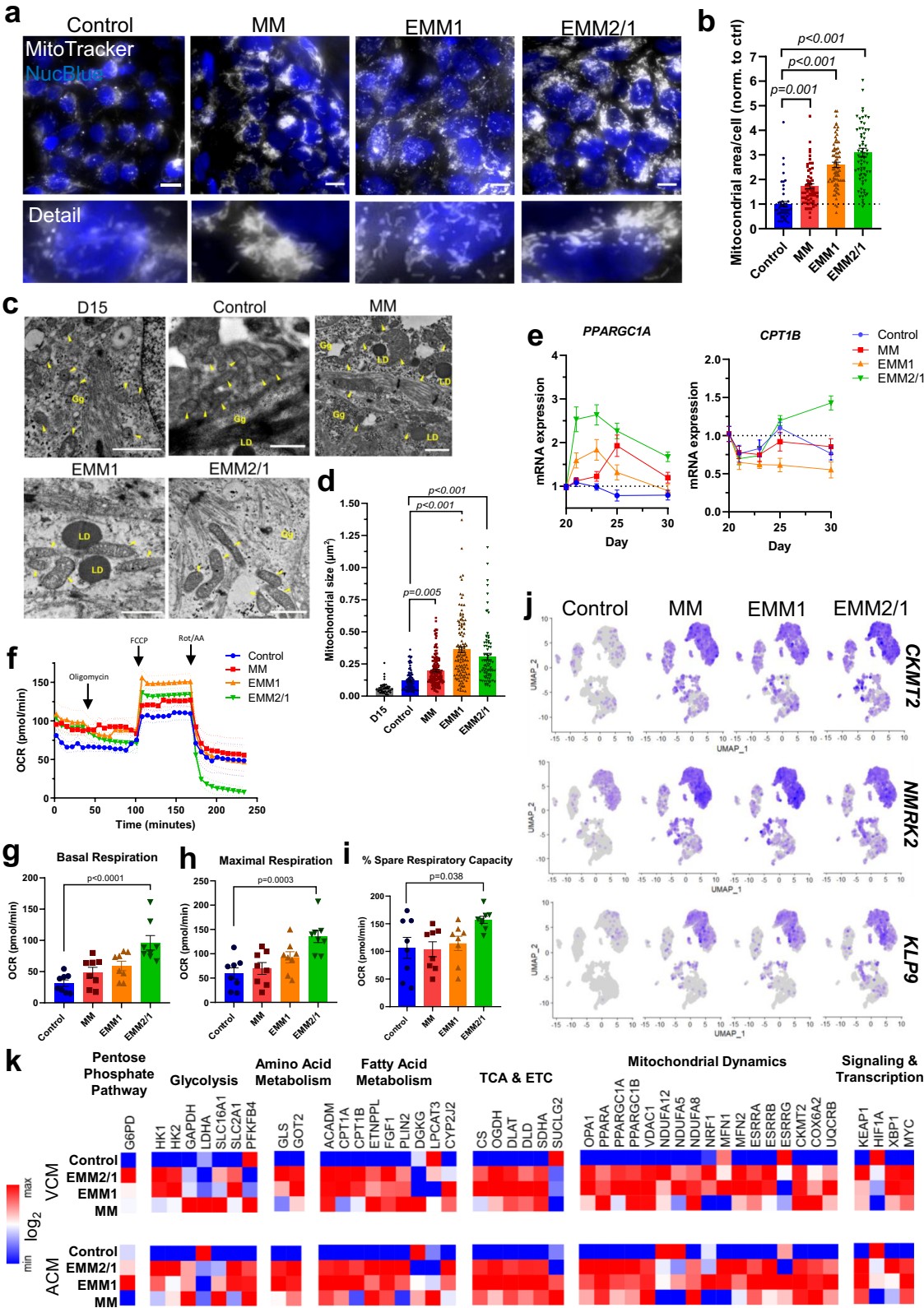

discovered a robust expression pattern in ACM and VCM clusters across all conditions 5Fig. 5d), including *RYR*, *ATP2A2*, *SCN5A*, *KCNJ2*, and *KCNH2*. Expression levels for all genes appeared to increase slightly to moderately for the EMM2/1 condition relative to control. Notably, *KCNJ2* expression increased dramatically for all maturation conditions relative to control, particularly in the EMM2/1 condition. An additional ion channel of critical importance is the hERG channel encoded by the gene *KCNH2*. Mutations and perturbations in this channel can lead to shortening or prolongation of the QT interval, and drug interactions with this channel can lead to cardiac arrythmia which represents a critical bottleneck surrounding drug discovery and development[41]. Interestingly, *KCNH2* displayed high expression levels within the ACM and VCM clusters in all conditions of our organoids (Fig. 5d).

**Fig. 4 | Human heart organoids develop increasingly mature metabolic profiles following developmental induction conditions. a** Mitochondrial labeling within day 30 human heart organoids in each condition ($n = 6$ independent organoids per condition). White = Mitotracker, blue = NucBlue. Scale bars = 10 µm. Detailed images of mitochondria are shown below each main image. **b** Quantification of mitochondrial area surrounding each individual nucleus ($n = 6$ independent organoids per condition, $n = 50, 59, 74$, and $70$ measurements for control, MM, EMM1, and EMM2/1, respectively). Values = mean ± s.e.m., one-way ANOVA with Dunnett's multiple comparisons tests. **c** TEM images displaying mitochondria in day 15 organoids and in organoids from each maturation condition at day 30 ($n = 4$ independent organoids per condition). Yellow arrows indicate mitochondria, LD = lipid droplets, Gg = glycogen granules. Scale bars = 1 µm. **d** Quantification of mitochondrial area from TEM images. Values = mean ± s.e.m., one-way ANOVA with Dunnett's multiple comparisons tests ($n = 4$ independent organoids per condition, $n = 40, 91, 154, 117$, and $81$ mitochondria measured for D15, control, MM, EMM1, and EMM2/1, respectively). **e** mRNA expression of metabolic genes *PPARGC1A* and *CPT1B* between days 20 and 30 of culture for each condition ($n = 8$ independent organoids per condition across three independent experiments). Data presented as $\log_2$ fold change normalized to Day 20. Values = mean ± s.e.m. **f** Oxygen consumption rate measurements from Agilent Seahorse XFe96 metabolic stress test assay in all conditions ($n = 8$ independent organoids per condition across two independent experiments). Values = mean ± s.e.m. Quantifications from oxygen consumption rate assay ($n = 8$ independent organoids per condition. Values = mean ± s.e.m., one-way ANOVA) including **g** Basal respiration, **h** Maximal respiration, and **i** Spare respiratory capacity. **j** Feature plots displaying key metabolic genes upregulated in the VCM and ACM clusters. **k** Expression heatmaps of key metabolic genes in the VCM and ACM clusters in each condition. Data displayed as $\log_2$ and is normalized to each column (for each gene and cluster). Source data are provided as a Source Data file.

Further, autonomic control of the cardiac conduction system through adrenergic signaling plays a significant role in heart physiology, and underlies a range of CVDs from heart failure and hypertension to arrythmia[42]. We identified the presence of critical beta-adrenergic receptor genes, *ADRB1* and *ADRB2*, encoding beta-adrenergic receptors 1 and 2 within our organoids in each condition (Supplementary Fig. 29). While *ADRB2* was expressed in both the ACM and VCM clusters in each condition, *ADRB1* showed expression within the ACM and VCM clusters in MM, EMM1 and EMM2/1 conditions, but was only expressed in the ACM cluster in the control condition. *ADRB3* was sparsely expressed relative to *ADRB1* and *ADRB2*, which stays true to cardiac physiology[21]. To investigate the temporal dynamics of key ion channels through the application of our developmental maturation strategies, we utilized qRT-PCR from day 20 to day 30 of organoid culture to assess levels of calcium (*ATP2A2*), sodium (*SCN5A*), and potassium (*KCNJ2*) transporters (Fig. 5e). *ATP2A2* expression increased in all conditions relative to control, with EMM2/1 exhibiting the most marked upregulation of fourfold (relative to control) at day 25 and day 30. *SCN5A* expression was upregulated for all conditions from day 21 to day 30 relative to day 20. Notably, MM and EMM2/1 organoids displayed a 3-fold increase at day 30 relative to day 20, while control organoids only displayed a 2-fold increase. *KCNJ2* expression steadily decreased in the EMM1 condition relative to control, with MM organoids exhibiting upregulation at day 30. Meanwhile, EMM2/1 displayed upregulation compared to control throughout the culture period up until day 30. We investigated the voltage activity within control and EMM2/1 heart organoids via the potentiometric dye di-8-ANEPPS and identified unique actional potentials within individual cardiomyocytes indicative of the presence of specialized atrial- and nodal-like cells but, interestingly, ventricular-like action potentials were only observed in EMM2/1 organoids (Fig. 5f).

Proper excitation-contraction coupling, depolarization, and repolarization of cardiomyocytes depend on specialized invaginations of the sarcolemma (t-tubules), which are indicative of cardiomyocyte maturation[43]. We assessed t-tubule presence in human heart organoids at day 30 via caveolin-3 immunofluorescence imaging (Fig. 5g) and discovered t-tubules among and surrounding sarcomeres (TNNT2⁺) within organoids in each condition, with increasing t-tubule density quantified in the EMM2/1 condition (Fig. 5h). We also utilized fluorescently labeled wheat germ agglutinin (WGA) to assess t-tubules in human heart organoids at day 30 (Supplementary Fig. 30a, b) and discovered similar results indicating EMM2/1 organoids possessed marked increases in t-tubule density. We assessed the presence of KCNJ2 via confocal microscopy (Fig. 5i). KCNJ2⁺ puncta were observed in each condition, with a 2-fold increased presence in the EMM2/1 condition relative to control (Fig. 5j), supporting previous data displaying increased amounts of KCNJ2 transcripts in EMM2/1 organoids. Together, this data shows that our developmentally matured organoid platform, specifically the EMM2/1 strategy, produces organoids that recapitulate significant electrophysiological aspects of cardiac development, physiology, and disease.

## Developmental induction promotes the emergence of a proepicardial organ and formation of atrial and ventricular chambers by self-organization

We have shown that developmentally induced heart organoids present improved cellular, biochemical, and functional properties when compared to their control counterparts and exhibit multiple features reminiscent of GD45 human fetal hearts. However, previous heart organoid attempts have lacked anatomically relevant cardiac structure and morphology to a great extent, including our previous work[8–10]. Given the significant changes observed through applying the EMM2/1 strategy, we decided to characterize morphological changes that took place under this improved condition. Organoids were harvested on day 30 of culture and stained for WT1 (proepicardium and epicardial cells) and TNNT2 (cardiomyocytes) (Fig. 6a). Organoids in each developmental induction condition displayed TNNT2⁺ and WT1⁺ cells, consistent with our previous observations[8], indicating the presence of epicardial and cardiomyocyte populations distributed throughout the organoids. Assessing both surface and interior planes of the organoids, organoids in all conditions possessed two distinct "chambers" marked via WT1⁺ and TNNT2⁺ cells. TNNT2⁺ cells were densely packed and formed a thick myocardial wall in the lower chamber, while also present in the upper region in a less dense arrangement directly underneath WT1⁺ cells (Supplementary Fig. 31). In EMM2/1, WT1⁺ cells were found densely covering the outer surface of the upper region. This clear organization pattern was not observed in the control, MM, or EMM1 culture conditions. We quantified the area of WT1⁺ and TNNT2⁺ chambers across all maturation conditions (Fig. 6b, c). We found no difference in TNNT2⁺ chamber area in MM and EMM1 organoids relative to control but found that EMM2/1 organoids display a 1.54-fold increased area relative to control. Additionally, we found no difference in WT1⁺ chamber area in MM organoids relative to control, whereas EMM1 and EMM2/1 organoids displayed increased areas (fold change) of 1.77 and 1.98, respectively. This data shows that organoids in all conditions undergo significant morphological organization, especially in the EMM2/1 condition, which included the emergence of an organoid with advanced myocardial dual-chamber morphology as well as a proepicardial pole.

We could determine that ventricular (MYL2) and atrial (MYL7) myosins were spatially restricted, particularly in EMM2/1 organoids (Fig. 6d). All organoids expressed MYL7 throughout the bulk of the organoid, but expression was strongly localized to the upper chamber in EMM2/1, suggestive of an atrial-like chamber. In the control and MM conditions, organoids possessed MYL2 in a high variety of locations that were not restricted to a polar end of the organoid or to either chamber, suggesting disorganization. On the other hand, organoids in the EMM1 and EMM2/1 conditions displayed an increased presence of

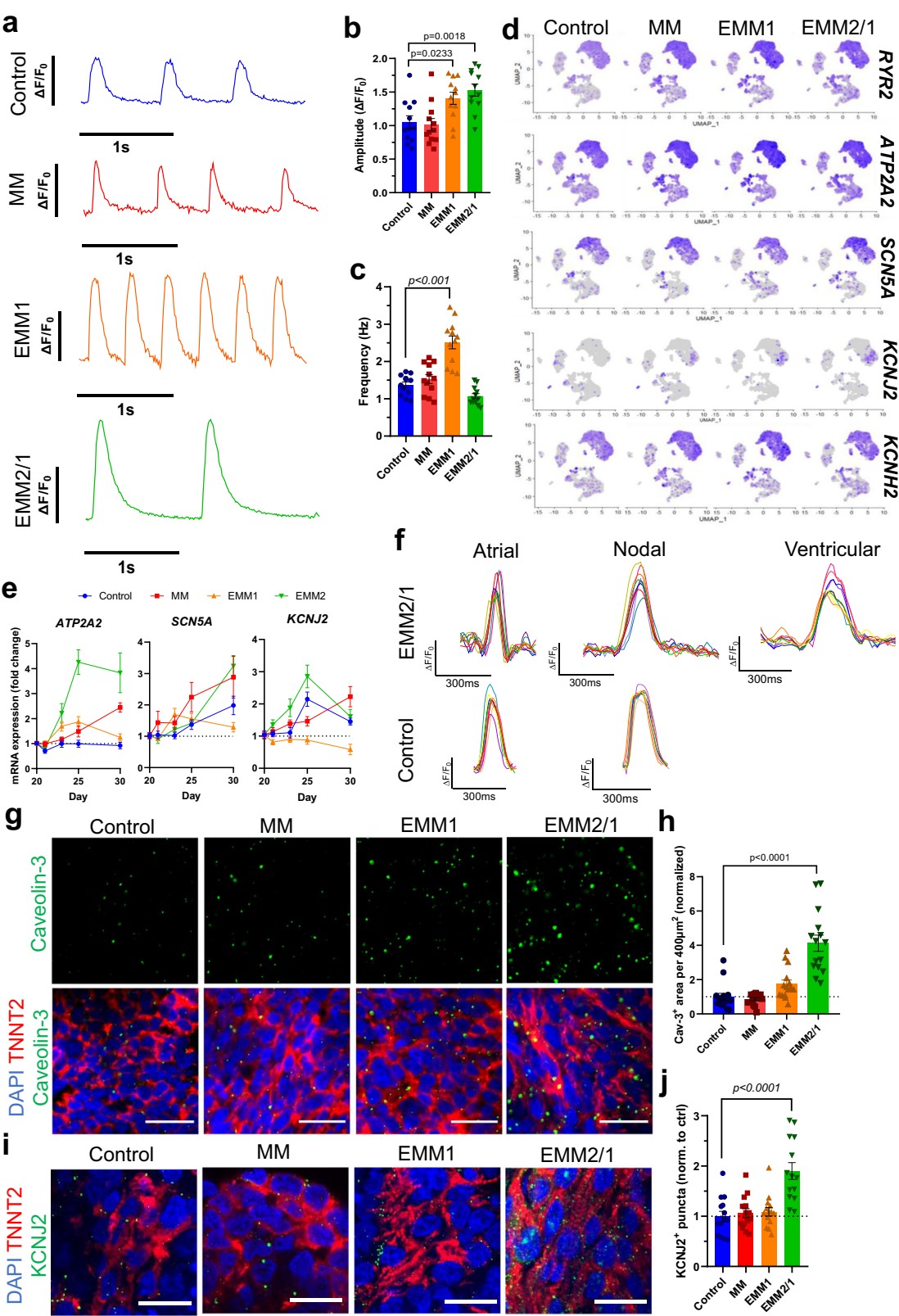

MYL2+ staining and degree of organization compared to control, showcasing MYL2 restricted to one polar end of the organoids and with EMM2/1 organoids displaying a 5.5-fold increase in MYL2+ area (Fig. 6e); suggesting the formation of a ventricular-like chamber. However, as these proteins are known to be inconsistently expressed at all stages of heart development, further investigation utilizing more specific proteins and genes for ventricular and atrial chamber identity

are necessary to truly define these two cellular populations and their localization within the organoids (see Fig. 6f–j). Overall, this organization is reminiscent of heart tube anterior-posterior patterning present in utero during development (see Fig. 7a for a schematic).

To further investigate the identity of ventricular- and atrial-like chambers in the human heart organoids, we performed staining for additional atrial and ventricular chamber markers NR2F2 (atrial) and

**Fig. 5 | Developmental induction conditions promote progressive electro-physiological maturation in human heart organoids. a** Representative calcium transient traces within d30 human heart organoids from each condition ($n = 12$ independent organoids per condition across three independent experiments). Traces represent data from an individual cardiomyocyte within human heart organoids. **b** Quantification of peak amplitude of calcium transient traces from each condition ($n = 12$ across three independent experiments). Values = mean ± s.e.m. **c** Quantification of calcium transient peak frequency for each condition ($n = 12$ independent organoids per condition across three independent experiments). Values = mean ± s.e.m. **d** Feature plots displaying key electrophysiological genes differentially expressed in the VCM and ACM clusters in each condition. **e** mRNA expression of key electrophysiological genes between d20 and d30 for each condition ($n = 8$ independent organoids per day per condition per gene across three independent experiments). Data presented as $\log_2$ fold change normalized to d20. Values = mean ± s.e.m. **f** Representative voltage tracings of organoids in the EMM2/1 and control conditions depicting atrial-, nodal-, and ventricular-like action potentials ($n = 8$ (EMM2/1, Ventricular), 9 (EMM2/1, Atrial and Nodal; Control, Atrial), and 10 (Control, Nodal) individual cells from three independent organoids across three independent experiments). Representative immunofluorescence images (**g**) and quantification (**h**) of caveolin-3 puncta for each condition ($n = 15$ independent organoids per condition across three independent experiments). Green = caveolin-3, red = TNNT2, blue = DAPI. Scale bar = 20 µm. Data presented as fold change normalized to control. Values = mean ± s.e.m., one-way ANOVA with Dunnett's multiple comparisons tests. Representative immunofluorescence images (**i**) and quantification (**j**) of KCNJ2+ puncta for each condition ($n = 14$ independent organoids per condition across three independent experiments). KCNJ2 = green, TNNT2 = red, DAPI = blue. Scale bar = 20 µm. Data presented as fold change normalized to control. Values = mean ± s.e.m., one-way ANOVA with Dunnett's multiple comparisons tests. Source data are provided as a Source Data file.

MYL3 (ventricular) (Fig. 6f). EMM2/1 organoids displayed a distinct, increased degree of separation between the two chambers, while control organoids showed a larger overlap of these two proteins (Fig. 6g); indicating that EMM2/1 organoids possess a greater degree of specification and maturity of chamber development (Supplementary Fig. 32). Remarkably, these results were highly reproducible in other PSC lines, including BYS0111 (iPSC) and H9 (ESC) (Supplementary Fig. 33a). While L1 control and EMM2/1 organoids were showed again for reproducibility and comparative purposes (Supplementary Fig. 33a, b), control BYS0111 organoids displayed similar overlap of NR2F2 and MYL3, while EMM2/1 BYS0111 organoids showed distinct separation of NR2F2 and MYL3 (Supplementary Fig. 33a, c), with MYL3+ cells highlighting thick myocardial walls in the EMM2/1 condition. Control H9 organoids displayed decreased expression of both NR2F2 and MYL3 compared to EMM2/1 H9 organoids, with EMM2/1 H9 organoids exhibiting distinct separation of NR2F2+ and MYL3+ chambers (Supplementary Fig. 33a,d). To support these immuno-fluorescence results describing the potential identity of atrial and ventricular chambers in our heart organoids, we investigated gene expression patterns using scRNA-seq data in the ACM and VCM clusters (Fig. 6h–j). ACMs displayed increased gene expression for hallmark atrial chamber identity markers such as *NR2F2, TBX5, NPPA*, and *NR2F1*[44] compared to VCMs (Fig. 6i). Meanwhile, VCMs showcased increased gene expression for hallmark ventricular chamber identity markers such as *MYL3, HEY2, IRX4*, and *HAND1*[27] compared to ACMs (Fig. 6j). These results highlight not only the recapitulation of post-heart tube and primitive heart morphology in our heart organoid platform, but also the reproducibility of our findings.

Optical coherence tomography (OCT) was used to provide detailed characterization of the chambers in live organoids over time, and to measure the growth and monitor dynamics of chamber development under developmental induction conditions via a custom-made OCT microscopy system amenable to high-content screening[45,46] (Supplementary Fig. 34a). We found that chambers exhibited dynamic behavior initially and coalesced into larger structures over time. EMM2/1 conditions led to the largest internal chambers within our human heart organoids between day 20 and day 30 of culture, with typically two large internal chambers as previously observed by confocal microscopy (Supplementary Fig. 34b, c, and Supplementary Movies 13–16). While MM organoids displayed a single internal chamber, organoids grown in the control, EMM1 and EMM2/1 conditions possessed multiple, smaller, interconnected chambers. Control and EMM2/1 organoids possessed chambers throughout the bulk of the organoids while EMM1 organoids showed chambers predominantly towards one side of the organoid. These data confirmed the formation of well-established cardiac chambers and further supported our observations on the effects of developmental induction conditions. The process by which chambers seem to form might be a limitation of our model, as it does not follow the biological paradigm, but this is also to be expected as there is no vasculature or external circulation to support chamber development in a more physiological manner.

We also assessed vasculature formation in developmental induction conditions. Endothelial cell (PECAM1+) vasculature formation was examined at day 30 of culture via immunofluorescence and confocal microscopy (Supplementary Fig. 35, Supplementary Movies 9–12). Assessment of organoids on surface and interior planes revealed the presence of endothelial cells amongst the myocardial regions of all organoids (Supplementary Fig. 35a). Organoids in the EMM1 and EMM2/1 conditions presented less PECAM1+ cells than control and MM organoids. Control and MM organoids displayed robust, interconnected endothelial cell networks and throughout myocardial (TNNT2+) tissue (Supplementary Fig. 35b). We quantified total PECAM1+ area, and MM organoids showed no significant difference compared to control organoids, while EMM1 and EMM2/1 organoids possessed only 52% and 61% of PECAM1+ area compared to control, respectively (Supplementary Fig. 35c), consistent with scRNA-seq data that shows a decrease in endothelial gene expression in EMM1 and EMM2/1 organoids compared to control (Fig. 3a). High magnification images of organoids further show the morphological transitory state of the endothelial cells within cardiomyocyte-rich regions (Supplementary Fig. 35d). Overall, this data suggests that vascularization of the organoids might be partially eclipsed by factors in EMM1 and EMM2/1 conditions, possibly due to timing or concentration of growth factors, and will require further investigation to fine-tune medium conditions.

### An endogenous retinoic acid gradient is responsible for spontaneous anterior-posterior heart tube patterning in EMM2/1 organoids

The emergence of a retinoic acid gradient originating at the posterior pole of the heart tube (produced by the epicardium and primitive atrium) is a critical developmental step in mammalian cardiogenesis[47]. This gradient establishes the anterior-posterior axis that provides cues for the formation of the ventricles, atria and inflow and outflow tract, while also contributing to the specification of cardiogenic progenitors and other structures[48] (Fig. 7a). To determine whether the post heart tube stage structure observed in EMM2/1 organoids (Fig. 6) was induced by endogenous retinoic acid signaling, we performed Raman microscopy to detect its molecular signature using a microscope designed for this purpose (Supplementary Fig. 36). We identified the presence of myosin, troponin T, tropomyosin, collagen I and other related molecular signatures in organoids in all conditions as expected, but the presence of retinoic acid was specifically identified only in EMM2/1 organoids (Fig. 7b). Retinoic acid synthesis is carried out largely by retinaldehyde dehydrogenase 2 (ALDH1A2) during embryogenesis[48]. We utilized qRT-PCR at day 30 of organoid culture to assess levels of *ALDH1A2* in all conditions (Fig. 7c). *ALDH1A2* expression

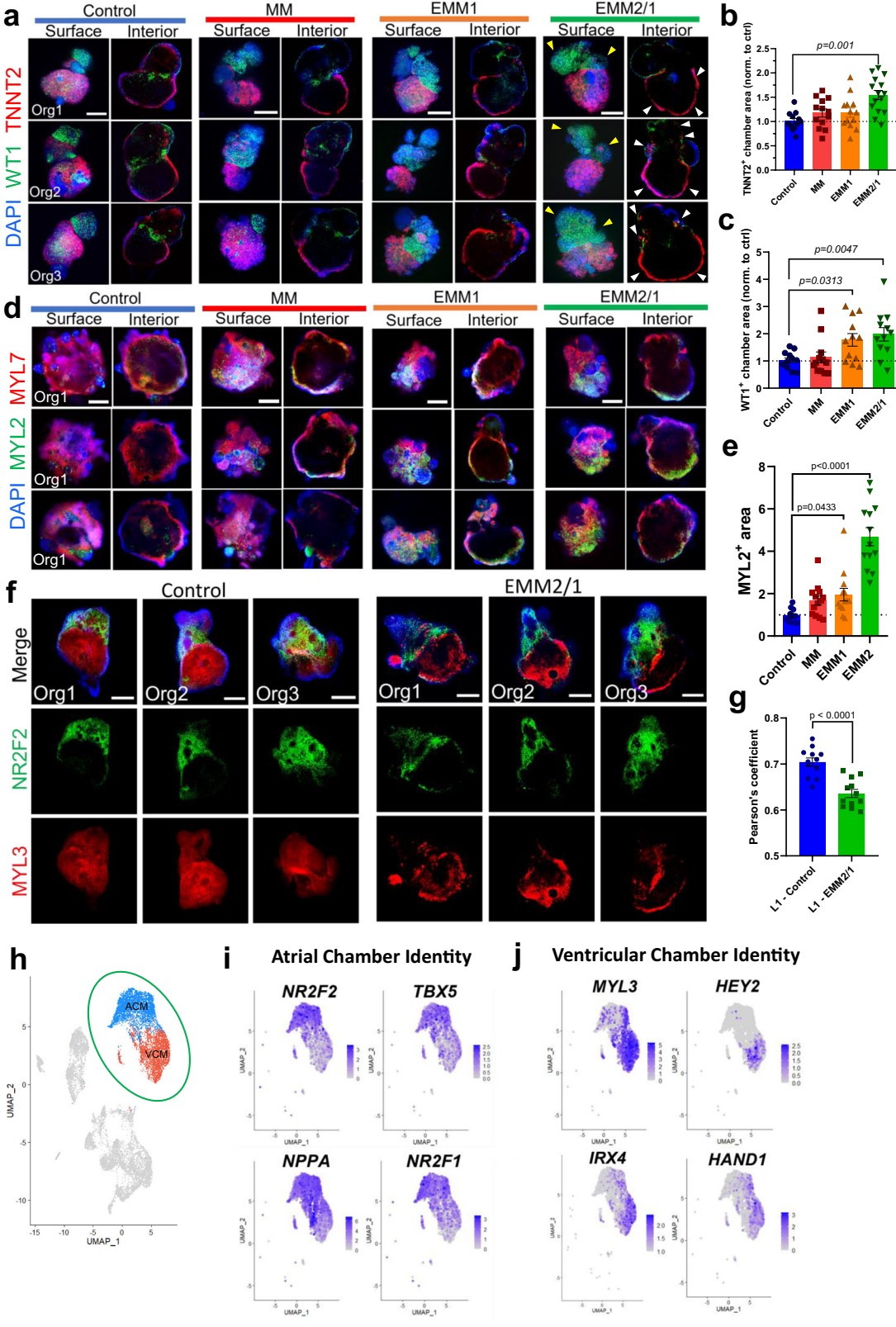

was shown to increase by ~2.2-fold in the EMM2/1 condition relative to control, with no significant changes in expression displayed in the MM or EMM1 conditions.

To assess the cell-specific dynamics of retinoic acid production in our organoids, we used scRNA-seq data to show that *ALDH1A2* is expressed in EPCs, PEDCs, and ACMs in our organoids (Fig. 7d); consistent with expression patterns reported in vivo[47]. To complement

this analysis and to further investigate the localization of retinoic acid production in our organoids, we performed immunostaining with antibodies for ALDH1A2 and TBX18 (an epicardial transcription factor, to label the proepicardial/atrial pole)[49] for organoids in all conditions at day 30 (Fig. 7e, Supplementary Fig. 37). We found that EMM2/1 organoids possess a localized, polarized expression of ALDH1A2 which colocalized with proepicardial organ TBX18+ cells, confirming that the

**Fig. 6 | Developmental induction promotes the emergence of a proepicardial organ and formation of distinct atrial and ventricular chambers by self-organization. a** Representative surface and interior immunofluorescence images of individual day 30 organoids in all conditions displaying WT1 (green), TNNT2 (red), and DAPI (blue). Three organoids are displayed for each condition ($n = 12$–$15$ independent organoids per condition across two independent experiments). Yellow arrows represent WT1+ cells on outer surface. White arrows represent TNNT2+ cells on lower chamber wall. Scale bars = 200 m. Quantification of TNNT2+ chamber area (**b**) and WT1+ chamber area (**c**) in each condition from images presented in (**a**) ($n = 12$, 13, 13, and 15 (**b**) and $n = 12$ (**c**) independent organoids for control, MM, EMM1, and EMM2/1, respectively, across two independent experiments). Values are presented as fold change normalized to control. Values = mean ± s.e.m., one-way ANOVA with Dunnett's multiple comparisons test. **d** Representative surface and interior immunofluorescence images of individual day 30 organoids in all conditions displaying MYL2 (green), MYL7 (red), and DAPI (blue). Three organoids are displayed for each condition ($n = 13$ independent organoids per condition across three independent experiments). Scale bars = 200 μm. **e** Quantification of MYL2+ area in each organoid in each condition from images presented in (**d**) ($n = 13$ independent organoids per condition across three independent experiments). Values are presented as fold change normalized to control. Values = mean ± s.e.m., one-way ANOVA with Dunnett's multiple comparisons test. **f** Representative immunofluorescence images of individual day 30 organoids in all conditions displaying atrial marker NR2F2 (green), ventricular marker MYL3 (red), and DAPI (blue). Three organoids are displayed for each condition ($n = 12$ organoids per condition across two independent experiments). Scale bars = 200 μm. **g** Quantification of colocalization (Pearson's coefficient) between NR2F2 (green) and MYL3 (red) from images presented in (**f**). Values = mean ± s.e.m., unpaired $t$ test. **h** Feature plot highlighting scRNA-seq VCM and ACM clusters. Feature plots displaying hallmark atrial chamber identity genes (**i**) and ventricular chamber identity genes (**j**) that are differentially expressed in the ACM (**i**) or VCM (**j**) cluster. Source data are provided as a Source Data file.

retinoic acid gradient patterning the organoids was coming from the proepicardial/atrial pole (posterior pole of the heart tube in utero) (Fig. 7a, e, f). We confirmed that the proepicardial pole is also the atrial pole by performing immunostaining for WT1 and MYL3 - showing that the proepicardial pole is opposite to that of the ventricular pole (Fig. 7h). Control, MM and EMM1 organoids did not display ALDH1A2 expression. We quantified the area of colocalization between ALDH1A2 and TBX18 and show that EMM2/1 organoids were significantly more responsive to the induction of retinoic acid synthesis (Fig. 7g). Furthermore, these results were reproduced in the two additional PSC lines BYS0111 and H9 (Supplementary Fig. 38a). While L1 control and EMM2/1 organoids were displayed once again for reproducibility and comparative purposes (Supplementary Fig. 38b), control and EMM2/1 BYS0111 organoids displayed similar patterns of ALDH1A2 and TBX18 expression as was shown for L1 organoids. EMM2/1 BYS0111 organoids also displayed markedly increased amounts of polarized ALDH1A2⁺TBX18⁺ cells compared to control BYS0111 organoids (Supplementary Fig. 38a, c). H9 organoids displayed a similar degree of recapitulation, with EMM2/1 H9 organoids showcasing marked increases in ALDH1A2⁺TBX18⁺ cells compared to control H9 organoids (Supplementary Fig. 38a, d). Control and EMM2/1 organoids from all three cell lines also displayed similarly robust and reproducible transcriptomic signatures (Supplementary Fig. 39a, b, c, d, e, f, g) for *ALDH1A2* and other important genes such as *MYL2, MYL7, WT1*, and *PPARGC1A*, as determined by qRT-PCR.

To further demonstrate the functional importance of the endogenous retinoic acid signaling observed in EMM2/1 organoids, we inhibited ALDH1A2 and retinoic acid production using the highly specific inhibitor DEAB, and through immunostaining for NR2F2 and MYL3, we showed that ALDH1A2 inhibition led to significantly impaired heart organoid patterning in EMM2/1 conditions (Fig. 7i, j, k). Organoids with inhibited ALDH1A2 displayed 0.35-fold and 0.42-fold reductions in MYL3⁺ and NR2F2⁺ areas relative to untreated organoids, respectively (Fig. 7j, k). Furthermore, we also showed that addition of exogenous retinoic acid to EMM2/1 conditions did not lead to further differences in patterning, suggesting that EMM2/1 organoids produce sufficient retinoic acid for normal patterning on their own (Fig. 7i, k). Together, these data show the ability of EMM2/1 organoids to endogenously synthesize retinoic acid in a spatially restricted manner colocalized with the epicardium (TBX18), a phenomenon that closely mimics the processes observed in in utero heart development and heart tube patterning.

### Ondansetron treatment during heart organoid development captures congenital heart disease phenotypes associated with its use in the clinic

Organoids possess the unique capacity to better model and investigate human development, organogenesis, and disease at an unprecedented scale and with greater precision compared to existing platforms. However, in the contexts of organogenesis and disease modeling, until now, human heart organoids have only been used to model developmental perturbations in diabetes-induced cardiomyopathy during pregnancy[8], gene knockout studies[10], and developmental cryoinjuries[9]. Therefore, while heart organoids show promise towards unraveling unanswered questions surrounding cardiogenesis and pathology, critical areas such as investigating developmental drug toxicity and broader morphological perturbations in cardiac pathologies remain ripe for discovery.

We sought to investigate the effects of ondansetron during human heart organoid development (Fig. 8). Ondansetron is a 5-HT₃ receptor antagonist and antiemetic used for treating nausea and vomiting[50]. Despite the lack of safety studies during pregnancy, ondansetron (also known as Zofran) is also the most common prescription medication for preventing nausea and vomiting during pregnancy, with up to 25% of pregnant women taking it off-label during this period[51]. Ondansetron has been epidemiologically linked to congenital heart defects (particularly ventricular septal defects) and orofacial defects[11,13], although consensus in the field is divided[52]. While the electrophysiological effects of ondansetron have been studied in vivo using animal models[12,53] and on cardiomyocytes in vitro[54,55], well-designed studies to thoroughly and directly investigate its safety as it relates to the development of the human heart are lacking. We used clinical data on ondansetron dosing strategies to determine relevant concentrations for heart organoid studies[11]. We applied ondansetron at three different concentrations to heart organoids from day 9 until day 30 in the EMM2/1 strategy and noticed stark differences in beating behavior and gross morphology at day 30, with organoids in the 10 μM and 100 μM conditions exhibiting a marked decrease in beating frequency (Supplementary Movies 17–20). Due to ondansetron's chemistry as a 5-HT₃ receptor antagonist, the electrophysiological effects in this respect were expected. We then assessed organoid morphology for atrial and ventricular cells, using MYL7 and MYL2 markers respectively (Fig. 8a), at day 30. A clear dose-dependent reduction in MYL2⁺ ventricular cells was found when ondansetron was applied (Fig. 8a). We quantified these results and showed that MYL2⁺ area decreased to 0.55-fold and 0.18-fold in organoids in the 10 μM and 100 μM conditions, respectively, relative to Untreated organoids (Fig. 8b), while MYL7⁺ area remained unchanged across all conditions (Fig. 8c). Furthermore, organoids in the 100 μM condition also appeared to be structurally less organized with less defined chamber walls and loose chamber separation compared to Untreated. To support these results, we performed qRT-PCR on organoids in all conditions and show that *MYL2* expression decreased to 0.58-fold and 0.40-fold in the 10 and 100 μM conditions, respectively, relative to Untreated (Fig. 8d). Together, these data suggest that ondansetron perturbs critical steps of ventricular heart development reminiscent of

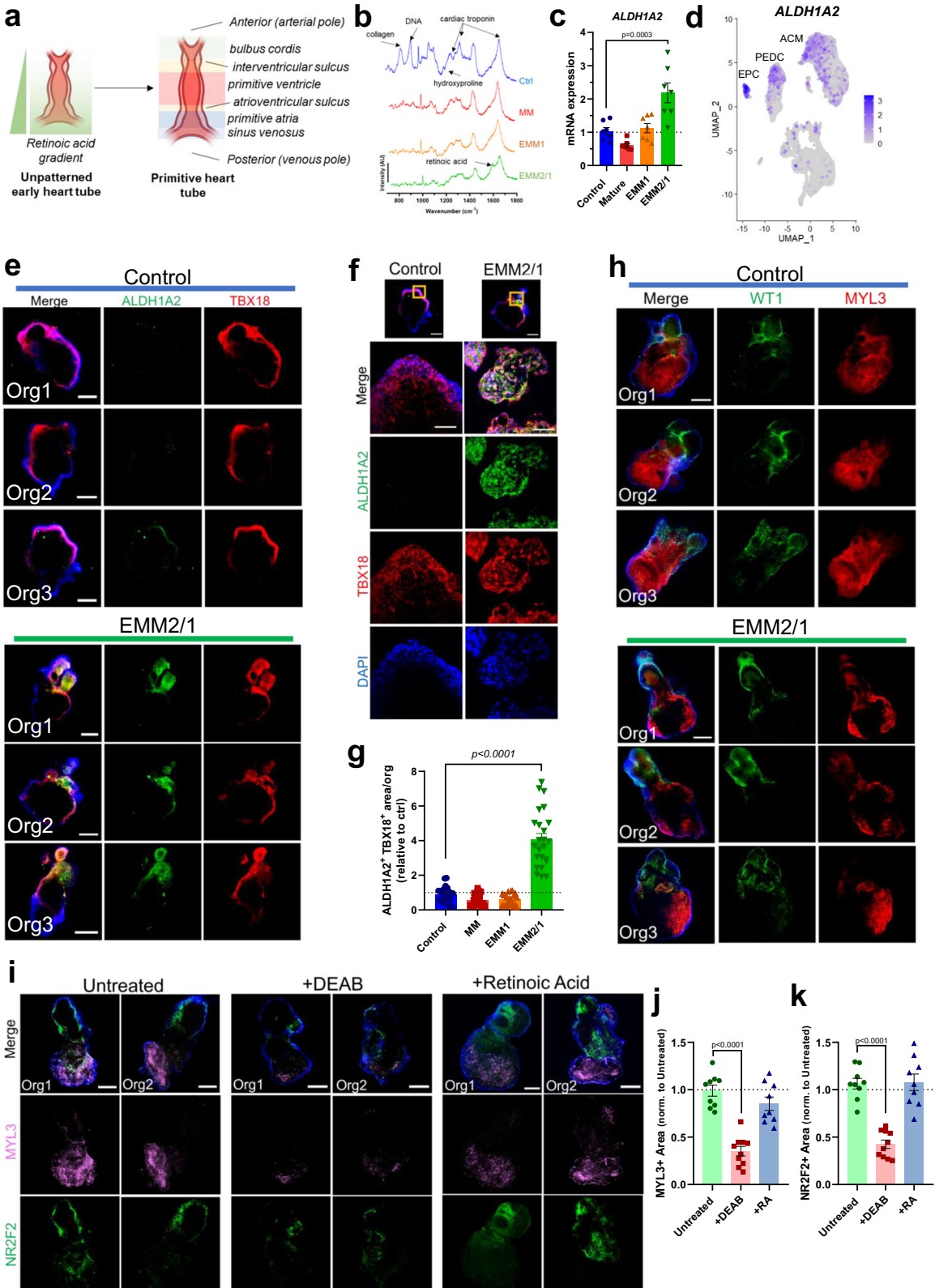

clinical phenotypes epidemiologically associated with its use, namely, ventricular septal defects.

Ondansetron has also been implicated in inducing arrythmias and in prolonging the QT interval[12,53,54]. Thus, we investigated the electrophysiological effects of ondansetron on heart organoid development (Fig. 8e-j) via the potentiometric dye di-8-ANEPPS. Action potentials for organoids in the 10 and 100 μM conditions were markedly different

compared to Untreated (Fig. 8e,f), displaying decreased frequencies (Fig. 8g), amplitudes (Fig. 8h), and increased APD30/90 (Fig. 8i, j), which is consistent with the known effects of ondansetron on hERG sodium channel blockade[56]. We also found that ondansetron did not contribute to apoptosis in human heart organoids (Supplementary Fig. 40a, b), but over time it did induce progressive loss of beating (Supplementary Fig. 40c), suggesting that prolonged ondansetron use

**Fig. 7 | An endogenous retinoic acid gradient is responsible for spontaneous anterior-posterior heart tube patterning. a** Schematic of in vivo heart tube formation, highlighting the localization and intensity of retinoic acid from anterior (arterial pole) to posterior (venous pole) of the primitive heart tube. Created using BioRender.com. **b** Raman spectroscopy intensity plots for all conditions at d30. Data representative of $n = 3$ organoids per condition. **c** mRNA expression of *ALDH1A2* in all conditions at d30 ($n = 7$ independent organoids/condition, two independent experiments). Data presented as $\log_2$ fold change normalized to control. Values = mean ± s.e.m., one-way ANOVA with Dunnett's multiple comparisons test. **d** Feature plot displaying expression of *ALDH1A2*. **e** Immunofluorescence images of individual day 30 organoids in the control and EMM2/1 conditions displaying ALDH1A2 (green), TBX18 (red), and DAPI (blue). Three organoids displayed per condition ($n = 22$-$24$ independent organoids per condition, three independent experiments). Scale bar = 200 μm. **f** High magnification images of organoids shown in (**c**), displaying ALDH1A2 (green), TBX18 (red), and DAPI (blue). Yellow square (Scale bar = 200 μm) represents the area of high magnification. Scale bar = 50 μm. **g** Quantification of ALDH1A2⁺ TBX18⁺ area within organoids in each condition from (**e**) and in Supplementary Fig. 30 ($n = 22, 22, 22,$ and 24 independent organoids for control, MM, EMM1, and EMM2/1, respectively, across three independent experiments). Data presented as fold change normalized to control. Values = mean ± s.e.m., one-way ANOVA with Dunnett's multiple comparisons test. **h** Immunofluorescence images of d30 organoids displaying WT1 (green), MYL3 (red), and DAPI (blue). Three organoids displayed per condition ($n = 12$ organoids/condition, two independent experiments). Scale bars = 200 μm. **i** Immunofluorescence images of d30 EMM2/1 organoids following exposure to either deoxyaminobenzaldehyde (DEAB), retinoic acid (RA), or Untreated. Staining was performed for ventricular marker MYL3 (pink), atrial marker NR2F2 (green), and DAPI (blue). Two organoids displayed for each condition (n = 9, 10, and 9 independent organoids for Untreated, +DEAB, and +RA, respectively, across two independent experiments). Scale bar = 200 μm Quantification of MYL3+ area (**j**) and NR2F2+ area (**k**) from organoids presented in (**i**). Values = mean ± s.e.m., one-way ANOVA with Dunnett's multiple comparisons test. Source data are provided as a Source Data file.

might have deleterious effects for electrophysiological maturation of the embryonic/fetal heart. Collectively, these data provide insight into the morphological and electrophysiological safety of ondansetron during human heart development and provide proof-of-concept for future investigations towards improving both the safety and efficacy of gestational medications and the pathology of congenital heart diseases.

## Discussion

Laboratory models of the human heart have made considerable progress over the last several decades, beginning with animal models and primary cardiomyocyte culture and moving onwards to induced pluripotent stem cell-derived cardiac tissues (e.g., cardiomyocytes) and tissue engineering approaches (3D printing, biomaterials). The latest advances in human heart models are heart organoids generated from pluripotent stem cells[8–10]. While these model systems have certainly yielded transformative research findings in the fields of cardiac disease, heart development, and cardiac toxicity testing[2–4,8], these systems do not possess the true complexity of the in utero human heart, owing to a lack of maturity and faithfulness to human physiology, morphology, cellular organization, and functionality. These shortcomings severely limit the scope of relevance of traditional model systems. To circumvent these limitations, we designed and implemented simple and highly reproducible methods for developmental induction strategies inspired by in utero biological steps, producing human heart organoids with higher anatomical complexity and physiological relevance along first-trimester fetal development. Among these strategies, we found that EMM2/1 most closely recapitulated heart development in vitro, and enabled organoids to acquire high levels of complexity and anatomical relevance by inducing progressive mitochondrial and metabolic maturation, electrophysiological maturation, increased morphological and cellular complexity, and most importantly, by recapitulating anterior-posterior heart tube patterning by endogenous retinoic acid signaling and self-organization. Importantly, results from using the EMM2/1 strategy over control were found to be reproducible across organoids derived from three pluripotent stem cell lines. Additionally, as a proof-of-concept, we show that EMM2/1 organoids can be used as a model platform for investigating the role of ondansetron in congenital heart defects.

Single-cell gene expression across multiple cardiac cell clusters revealed that EMM2/1 organoids yielded high similarity to in vivo 6.5 post-conception week (GD45) developing human hearts[21]. Implementation of developmental induction strategies did not lead to the emergence of new cardiac lineages as the same cardiac cell types were observed in all conditions but did lead to expansion and reduction of certain populations, such as atrial and ventricular cardiomyocytes, and mesenchymal cell types (stromal cells) in what seems to be a process of fine tuning and remodeling. Interestingly, we could also observe the appearance of valvular and conductance cell types throughout all developmental induction conditions, a phenomenon not described before in heart organoids. Remarkably, organoids displayed a high degree of similarity to two independent embryonic human heart datasets across all major cardiac cell clusters.

The metabolic transition from glycolysis to fatty acid oxidation is a paramount step in the late stages of cardiac development, preparing the heart for increased energy expenditure as well as inducing transcriptional regulation and stimulating physiological maturation[35]. Efforts have been pursued to simulate these phenomena in vitro with cardiomyocytes and engineered heart tissues and have found beneficial effects from modified glucose concentrations and the addition of fatty acids[4,14,15]. However, these systems are simplistic models and do not possess the high physiological complexity as observed in human heart organoids. We showed that human heart organoids respond dramatically to developmental maturation stimuli and metabolically maturate and possess increased mitochondrial growth, density, gene expression profiles, and oxygen consumption rates, particularly through the EMM2/1 strategy. These dramatic responses compared to traditional methods may be the result of synergy between multiple cardiac cell subtypes, such as epicardial cells and cardiac fibroblasts, which have been shown to stimulate cardiomyocyte growth and function[57,58].

Proper and gradual electrophysiological maturation throughout the cardiac syncytium, including the complex interplay between various ion channels and their subtypes as well as depolarization through t-tubules, comprises critical aspects of cardiac development and functionality[43,59]. Here, we show that organoids from the EMM2/1 strategy develop distinct calcium transients with increasing physiological mimicry due to their increased amplitude and decreased frequency. Additionally, organoids in the EMM2/1 strategy develop higher levels of t-tubules, inward-rectifying potassium ion channels, and hERG channels compared to other maturation strategies. In fact, many efforts towards in vitro cardiomyocyte maturation have struggled or failed to elicit the presence of t-tubules[60] and inward-rectifying potassium ion channels remain critical to establishing low resting membrane potential. Moreover, cardiac hERG channels represent a paramount channel of importance for pharmacological screening due to its high arrhythmogenic potential if interfered with[41]. Nonetheless, the summation of multiple ion transients results in the cardiac action potential which is the ultimate driving force for human heart contraction and functionality. We show that cardiomyocytes within our EMM2/1 organoids possess ventricular-, atrial-, and nodal-like action potentials, opening the door for electrophysiological applications in drug screening.

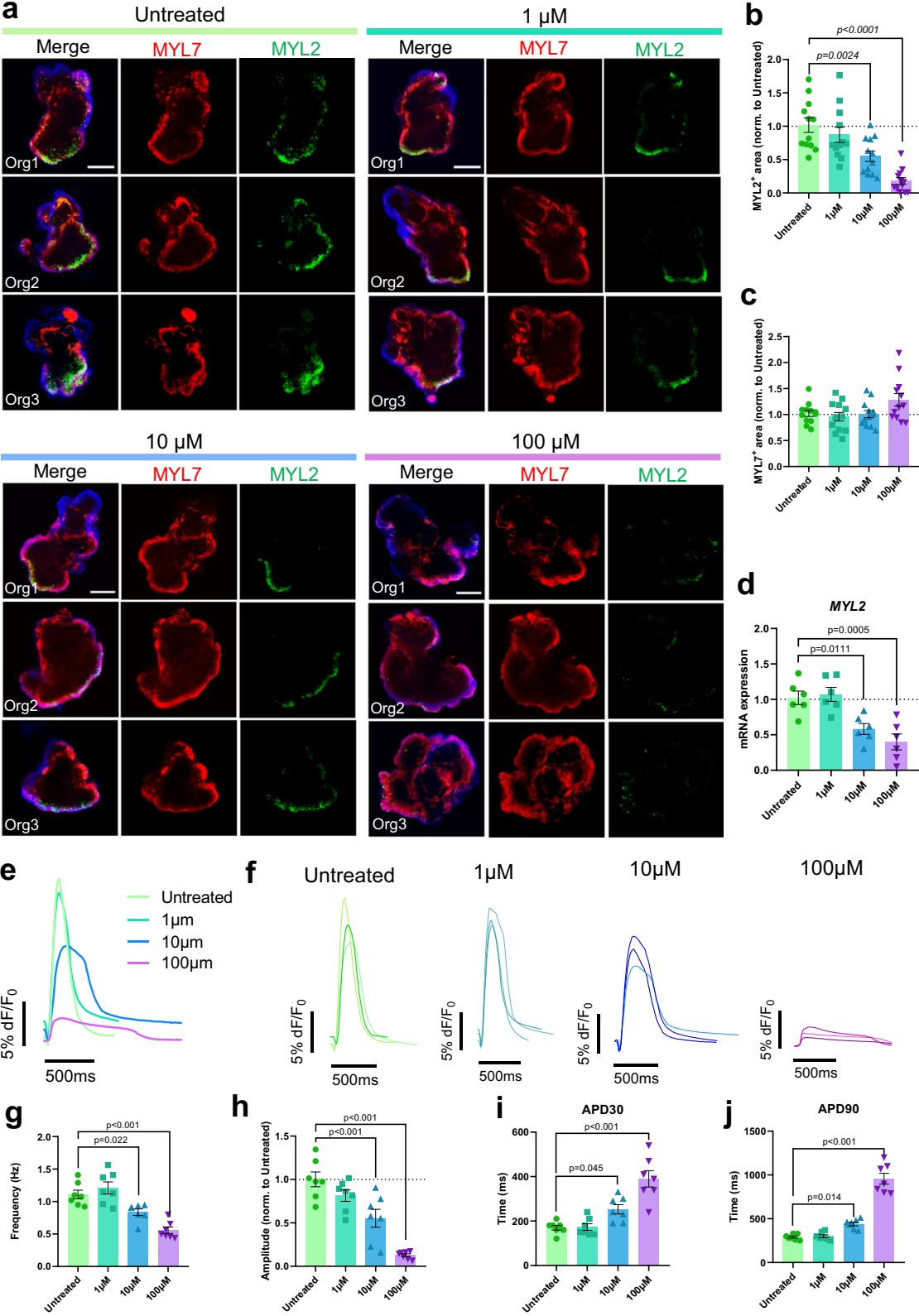

The embryonic heart begins as an unpatterned contracting tube and undergoes cellular and structural changes through morphogenetic signaling events to pattern along the anterior-posterior axis, loop, and eventually form the four-chambered primitive heart[61]. We investigated the morphological landscape of our organoids following the application of our maturation strategies and found that EMM2/1 organoids formed a two-chambered structure with cardiomyocytes forming one chamber with atrial identity and another with ventricular fate. Dense epicardial layering at the atrial chamber identified the proepicardial organ as the posterior pole of the heart tube[62] and revealed that EMM2/1 organoids were spontaneously patterning along the aforementioned anterior-posterior axis, a phenomenon that was exclusively observed in EMM2/1 organoids. Further investigations found that self-organization and patterning in EMM2/1 was driven by

**Fig. 8 | Primitive heart tube organoids demonstrate a direct link between ondansetron administration and ventricular cardiac defects. a** Representative immunofluorescence images of individual day 30 EMM2/1 organoids following exposure to varying concentrations of ondansetron (1 μM, 10 μM, or 100 μM) or no treatment (untreated) from day 9 to day 30 of culture. Staining was performed for ventricular marker MYL2 (green), atrial marker MYL7 (red), and DAPI (blue). Three organoids are displayed for each condition (*n* = 12 independent organoids per condition across two independent experiments). Scale bar = 200 μm. Quantification of MYL2+ area (**b**) and MYL7+ area (**c**) for each condition (*n* = 12 independent organoids per condition across two independent experiments). Data presented as fold change normalized to untreated. Values = mean ± s.e.m., one-way ANOVA with Dunnett's multiple comparisons test. **d** mRNA expression of ventricular marker

*MYL2* in all conditions at day 30 (*n* = 6 independent organoids per condition across two independent experiments). Data presented as log$_2$ fold change normalized to control. Values = mean ± s.e.m., one-way ANOVA with Dunnett's multiple comparisons test. Representative voltage tracings of organoids (**e**) showing three voltage traces from independent organoids in each condition (**f**) (representative of *n* = seven independent organoids per condition across two independent experiments). **g–j** Quantification of voltage tracings from individual organoids in each condition from traces presented in (**e**) and (**f**) (*n* = 7 independent organoids per condition across two independent experiments), displaying frequency (**g**), amplitude (**h**), APD30 (**i**), and APD90 (**j**). Values = mean ± s.e.m., one-way ANOVA with Dunnett's multiple comparisons test. Source data are provided as a Source Data file.

an endogenous retinoic acid signaling gradient, identified by using a combination of Raman and confocal microscopy. ALDH1A2, an enzyme required for retinoic acid synthesis, was observed to be spatially restricted to the posterior end of EMM2/1 organoids, and co-localized with TBX18, an epicardial transcription factor confirming that the proepicardial organ was functional. Further, the patterning role that retinoic acid plays in EMM2/1 organoids was confirmed through inhibiting ALDH1A2, with inhibited organoids showcasing diminished patterning and expression of hallmark chamber identity proteins. Moreover, organoid patterning and gene expression profiles in EMM2/1 organoids were shown to be reproducible across three independent cell lines and across multiple independent experiments (L1 (iPSC), BYS0111 (iPSC), H9 (ESC)). Taken together, these data support the hypothesis that our organoids recapitulate events that take place during in utero gestation where the proepicardial organ surrounds the posterior pole of the patterned heart tube where posterior atrial cardiomyocytes and proepicardial cells produce retinoic acid to form a signaling gradient that further instructs the remainder of the heart tube with patterning and specification information[21,48,49,62].

Cardiac organogenesis is a highly organized and carefully orchestrated process involving the interplay of millions of cells, and perturbations to this process can lead to congenital heart defects. Heart organoids represent a new avenue to explore heart development and congenital heart defects directly in humans and could be instrumental for disease modeling and pharmacological studies. For these reasons, we decided to investigate the effects of ondansetron (Zofran), a 5-HT$_3$ receptor antagonist and antiemetic used for treating nausea and vomiting[50] and the most common prescription medication for preventing nausea and vomiting during pregnancy, with up to 25% of pregnant women taking it off-label during this period[51]. Ondansetron has been epidemiologically linked to congenital heart defects (particularly ventricular septal defects) and orofacial defects[11–13], although consensus in the field is divided[52] and well-designed studies to thoroughly investigate its safety are lacking. We investigated the effects of ondansetron treatment during heart organoid development and found that ondansetron exposure causes electrophysiological alterations as expected due to its chemistry as a serotonin inhibitor. More interestingly, however, ondansetron also directly and strongly inhibited ventricular cardiomyocyte differentiation or maturation, even at low doses, by an unknown mechanism. This phenotype corresponds with epidemiological clinical findings which associate ondansetron use during pregnancy with ventricular septal defects[11,13]. This finding is important as it has significant implications in the clinic and suggests ondansetron should be administered with caution during pregnancy.

While our maturated human heart organoid technology opens exciting avenues for modeling the human heart in vitro, important limitations remain. First, further investigation is necessary to clarify the role of other important cellular populations, such as conductance cells and endothelial/coronary vasculature. EMM2/1 conditions had seemingly negative effects on organoid vascularization, a topic that deserves more attention as conditions can be further refined. Second, the lack of circulation constitutes a significant drawback that needs to

be addressed, possibly through using microfluidic devices. Third, further developmental steps need to be introduced to continue increasing the physiological relevance of the organoids. This includes correcting the lack of embryonic tissue-resident macrophage populations or the contributions of the neural crest. Finally, more anatomical events need to be modeled, such as outflow tract and atrioventricular canal formation, heart looping, and chamber septation.

In summary, we describe here a developmental induction strategy inspired by hallmarks of in utero development for producing patterned human heart tube organoids highly recapitulating human first trimester heart development. Our EMM2/1 developmental induction strategy yields several unique and crucial characteristics representative of an early fetal human heart, including the presence of anterior-posterior patterning with an endogenous retinoic acid gradient originating at the posterior end, polar separation of atrioventricular chambers, and a posterior, proepicardial pole; all shown in heart organoids from three independent cell lines. The EMM2/1 strategy also results in atrial and ventricular cardiomyocytes, valvular cells, conductance cells, epicardial cells, proepicardial cells and more, as well as large hollow chambers, functional electrophysiology, and increased mitochondrial density and metabolic transcriptional profiles. Moreover, EMM2/1 organoids display a high degree of similarity compared to embryonic human hearts at similar stages of development. To highlight the relevance and applicability of our model, we show that EMM2/1 organoids recapitulate and reveal potential mechanisms of ondansetron toxicity in the developing heart. To the extent of our knowledge, this is the first time the human heart tube has been reconstructed to this level of detail in vitro. All these developmental features highlight the ability to model the developing human heart in vitro and prove the potential of our methodology for establishing heart models for the study of normal heart development, congenital heart defects, cardiac pharmacology, regeneration, and other cardiovascular disorders in the future.

## Methods

### Pluripotent stem cell culture

The following human iPSC and human ESC lines were used for this study: iPSC-L1 (iPSC, male, iPSCORE), ATCC-BYS0111 (iPSC, male, ACS-1025), H9 (ESC, female, WiCell, WA09). iPSC-L1 was developed in house by Sendai reprogramming and quality controlled for pluripotency and normal karyotype following standard approaches[63], and has been described before by us an others[8,46]. Pluripotency, genomic stability, and Mycoplasma contamination were routinely tested for all hPSC lines used. hPSCs were cultured in Essential 8 Flex medium with 1% penicillin-streptomycin (Thermo) in 6-well plates on growth factor reduced Matrigel (Corning) inside an incubator at 37 °C and 5% CO$_2$. hPSCs were passaged using ReLeSR passaging reagent (STEMCELL Technologies) upon reaching 60-80% confluency.

### Self-assembling human heart organoid differentiation

Step-by-step, detailed protocols which describe the generation and differentiation of the human heart organoids are provided[14]. In brief,

hPSCs were grown to 60% confluency on six-well plates and dissociated using Accutase (Innovative Cell Technologies) to obtain a single-cell solution. hPSCs were collected and centrifuged at 300 $g$ for 5 min and resuspended in Essential 8 Flex medium containing 2 μM ROCK inhibitor (Thiazovivin) (Millipore Sigma). hPSCs were counted using a Moxi cell counter (Orflo Technologies) and were seeded at a concentration of 10,000 cells per well in round bottom 96 well ultra-low attachment plates (Costar) on day -2 in a volume of 100 μL. The plate was then centrifuged at 100 $g$ for 3 min and subsequently placed inside a 37 °C and 5% $CO_2$ incubator. After 24 h (day -1), 50 μL was removed from each well and 200 μL of fresh Essential 8 Flex Medium was added to each well to obtain a final volume of 250 μL per well. The plate was then placed inside a 37 °C and 5% $CO_2$ incubator. After 24 h (day 0), 166 μL of medium was removed from each well. Then, 166 μL of RPMI with B27 supplement without insulin (Gibco) supplemented with 1% penicillin streptomycin (Gibco) (hereafter termed "RPMI/B27 minus insulin") containing CHIR99021, BMP4, and Activin A was added to each well to obtain final concentrations of 4 μM CHIR99021, 36 pM (1.25 ng/mL) BMP4, and 8 pM (1.00 ng/mL) Activin A. The plate was subsequently placed inside a 37 °C and 5% $CO_2$ incubator. After exactly 24 h (day 1), 166 μL of medium was removed from each well and replaced with 166 μL of fresh RPMI/B27 minus insulin. On day 2, 166 μL of medium was removed from each well and 166 μL of RPMI/B27 minus insulin with Wnt-C59 (Selleck) was added to obtain a final concentration of 2 μM Wnt-C59 inside each well. The plate was then incubated for 48 h. On day 4, 166 μL was removed and replaced with fresh RPMI/B27 minus insulin and incubated for 48 h. On day 6, 166 μL was removed and replaced with 166 μL RPMI with B27 supplement (with insulin) and 1% penicillin streptomycin (hereafter termed RPMI/B27). The plate was incubated for 24 h. On day 7, 166 μL of media was removed from each well and 166 μL of RPMI/B27 containing CHIR99021 was added to obtain a final concentration of 2 μM CHIR99021 per well. The plate was incubated for 1 h. After 1 h, 166 μL of medium was removed from each well and 166 μL of fresh RPMI/B27 was added to each well. The plate was incubated for 48 h. From days 9 to 19, every 48 h, media changes were performed by removing 166 μL of media from each well and adding 166 μL of fresh RPMI/B27.

## Developmental induction conditions

Organoids were initially generated and differentiated according to the protocol outlined above. Beginning on day 20, organoids were subjected to developmental induction conditions: 1) control strategy is a continuation of culture within RPMI/B27 from day 20 to day 30, performing standard media changes every 48 h; 2) maturation medium (MM) strategy is employed from day 20 to day 30, performing media changes every 48 h using MM media, consisting of stock RPMI/B27 with 52.5 μM palmitate-BSA, 40.5 μM oleate-BSA (Sigma), 22.5 μM lineoleate-BSA (Sigma), 120 μM L-Carnitine (Sigma) and 30 nM T3 hormone (Sigma); 3) enhanced maturation medium 1 (EMM1) strategy is employed from day 20 to day 30, performing media changes every 48 h using EMM1 media, consisting of stock RPMI 1640 Medium, no glucose (Gibco) supplemented with B27 (with insulin), 1% penicillin streptomycin (Gibco), 52.5 μM palmitate-BSA, 40.5 μM oleate-BSA (Sigma), 22.5 μM lineoleate-BSA (Sigma), 120 μM L-Carnitine (Sigma), 30 nM T3 hormone (Sigma), 0.4 mM ascorbic acid (Thermo Fisher Scientific) and 4 mM Glucose (Gibco); 4) enhanced maturation medium 2/1 (EMM2/1) strategy is employed from day 20 to day 30, performing media changes every 48 h, utilizing a combination of two medias. From day 20 to day 26, EMM2 media is utilized which consists of stock RPMI 1640 Medium, no glucose (Gibco) supplemented with B27 (with insulin), 1% penicillin-streptomycin (Gibco), 52.5 μM palmitate-BSA, 40.5 μM oleate-BSA (Sigma), 22.5 μM lineoleate-BSA (Sigma), 120 μM L-Carnitine (Sigma), 30 nM T3 hormone (Sigma), 0.4 mM ascorbic acid (Thermo Fisher

Scientific), 4 mM Glucose (Gibco) and 50 ng/mL LONG R³ IGF-1 (*Repligen*). Continuing the EMM2/1 strategy, from day 26 to day 30, EMM1 media is utilized. Organoids are collected on day 30 for analysis.

## Immunofluorescence

Human heart organoids were transferred from the round bottom ultra-low attachment 96 well plate to 1.5 mL microcentrifuge tubes (Eppendorf) using a cut 200 μL pipette tip (to increase tip bore diameter as to not disturb the organoid). Organoids were fixed in 4% paraformaldehyde (VWR) in PBS for 30 min. Following, organoids were washed using PBS-Glycine (1.5 g/L) three times for 5 min each. Organoids were then blocked and permeabilized using a solution containing 10% Donkey Normal Serum (Sigma), 0.5% Triton X-100 (Sigma), and 0.5% BSA (Thermo Fisher Scientific) in PBS on a thermal mixer at 300 rpm at 4 °C overnight. Organoids were then washed three times using PBS and incubated with primary antibodies (Supplementary Table 1) within a solution containing 1% Donkey Normal Serum, 0.5% Triton X-100, and 0.5% BSA in PBS (hereafter termed "Antibody Solution") on a thermal mixer at 300 rpm at 4 °C for 24 h. Following, organoids were washed 3 times for 5 min each using PBS. Organoids were then incubated with secondary antibodies (Supplementary Table 1) in Antibody Solution on a thermal mixer at 300 rpm at 4 °C for 24 h in the dark. Subsequently, organoids were washed three times for 5 min each using PBS and mounted on glass microscope slides (Fisher Scientific). 90-μm polybead microspheres (Polyscience, Inc.) were placed between the slide and a No. 1.5 coverslip (VWR) to provide support pillars such that the organoids could retain three dimensionality. Organoids were transferred to the glass microscope slides using a cut 200 μL pipette tip and mounted using a clearing solution described previously[64]. T-tubule staining was performed using FITC-conjugated WGA lectins (Sigma).

## Confocal microscopy and image analysis

Immunofluorescence images were acquired using a confocal laser scanning microscope (Nikon Instruments A1 Confocal Laser Microscope) at 1024 × 1024 resolution and 16-bit. Images were analyzed using Fiji. When comparing images across or between conditions, for each channel of an image being measured, pixel intensity values of images were equalized to that of the control or EMM2/1 condition, where appropriate. To measure organoid diameter and area, the straight line and freehand tools were used, respectively. To measure mitochondrial (MitoTracker) area, Cav-3+ area, KCNJ2+ puncta, MYL2+ area, MYL7+ area, and PECAM1+ area, the auto threshold function was utilized, and the area was measured. To measure ALDH1A2 + TBX18+ area, the auto threshold function was utilized for ALDH1A2+ expression, and the freehand selection tool was used to select TBX18+ organoid area; the ALDH1A2+ thresholded area was calculated within this TBX18+ space. To measure TNNT2+ and WT1+ chamber size, puncta, and positive signal area the organoid was imaged at the middle plane (50% of thickness as measured by confocal microscopy; see end of section for more detail). Then, the oval selection tool was utilized, and the wall of the organoid was used as the boundary region of the respective area (TNNT2 or WT1) to be drawn (Supplementary Fig. 41). For area measurements that utilized the whole organoid (low magnification images), datapoints were normalized to organoid area. To measure FlipGFP fluorescence intensity, the mean gray value was calculated. To measure Pearson's coefficient, the JaCOP colocalization plugin was used. Thresholds were generated for the equalized image intensity values. A spatial resolution of 1.243 micrometers per pixel was utilized. The percentage of chambered organoids was quantified by visual screening for the distinct separation of MYL3 and NR2F2 staining. While organoids were imaged at both surface and interior planes, quantifications were made using a plane imaged at 50% of the

organoid's depth. Since it is difficult to control the orientation at which an organoid is placed onto the imaging slide, instances may arise where the exterior of the organoid is "pushed up" against the glass coverslip which may exacerbate or even hide certain features at the exterior. Therefore, acquiring and measuring at the interior allows for greater consistency in measurement and quantification.

### Single-cell RNA sequencing

Libraries were prepared using the 10x Chromium Next GEM Single Cell 3′ Kit, v3.1 and associated components. Completed libraries were QC'd and quantified using a combination of Qubit dsDNA HS, Agilent 4200 TapeStation HS DNA1000 and Invitrogen Collibri Library Quantification qPCR assays. The libraries were pooled in equimolar proportions and the pool quantified again using the Invitrogen Collibri qPCR assay. The pool was loaded onto two lanes of an Illumina NovaSeq 6000 SP flow cell (v1.5) and sequencing was performed in a custom paired-end format, 28 cycles for read 1, 2 10 cycle index reads and 90 cycles for read 2. A v1.5, 100 cycle NovaSeq reagent cartridge was used for sequencing. The 28 bp read 1 includes the 10× cell barcodes and UMIs, read 2 is the cDNA read. Output of Real Time Analysis (RTA) was demultiplexed and converted to FastQ format with Illumina Bcl2fastq v2.20.0. After demultiplexing, reads from each of the sample libraries were further processed using 10x Genomics cellranger count (v6.1.2). Counted data were processed using Seurat[65] and integrated using Harmony[66]. UMAP dimensional reduction plots were generated using the standard Seurat default Louvain algorithm. Cell-cell communication analysis and visualization was performed using Liana[67], SCpubr[68], and CellTalker (https://github.com/arc85/celltalker). For stage determination, 10x single-cell and 10x spatial Visium sequencing data of embryonic human hearts were obtained from refs. 21, 22 (GSE106118) and integrated as described above. Visium data was also processed without regard to spatial data. The UMAP datasets presented in gray are due to a lack of succinctly presented annotations from the publicly available datasets. For the stage determination analysis, we used the four most well-integrated clusters among all datasets (organoid and embryological) (ACM, VCM, PEDC, and EPC). At the next stage, single-cell data was converted to pseudobulk data using Seurat2Phantasus tool (https://github.com/ParkLaboratory/Seurat2Phantasus), batch effect was removed using Combat-seq[69]. Pseudobulk data analysis and PCA generation were performed using Phantasus[70]. After that, the top 1000 most highly expressed mRNAs were determined and mRNAs associated with ribosomes were removed, after which PCA was generated. Enrichr was used to assess gene ontologies. Pathview Web was used to generate biological pathway graphs[38]. When Feature plots are presented, the color intensity represents the relative value of gene expression per gene.

### Real-time RT-PCR

Organoids were collected on day 20, 21, 23, 25, and 30 and stored in RNAprotect (Qiagen) at −20 °C. RNA was extracted using the Qiagen RNEasy Mini Kit according largely to the manufacturer's instructions. Organoids were lysed using the Bead Mill 4 Homogenizer (Fisher Scientific) at speed 2 for 30 s. RNA concentration was measured using a NanoDrop One (Thermo Fisher Scientific). A minimum threshold of 10 ng/μL was required to proceed with reverse transcription. cDNA was generated using the Quantitect Reverse Transcription Kit (Qiagen) and stored at −20 °C. Primers for real time qPCR were designed using the Primer Quest tool (Integrated DNA Technologies). SYBR Green (Thermo Fisher Scientific) was used as the DNA intercalating dye and the amplifier for the reaction vessel. Real-time qPCR was performed using the QuantStudio 5 Real-Time PCR system (Applied Biosystems) using a total reaction volume of 20 μL. Gene expression levels were normalized to *HPRT1* expression in each independent sample. Log2

fold change values were obtained using the double delta CT method. At least four independent samples were run for each gene expression assay at each timepoint per condition. mRNA expression figures are displayed as log2-fold change relative to control. Primer sequences are provided in Supplementary Data 1.

### Mitochondrial imaging

Intracellular mitochondrial presence within human heart organoids was visualized using Mitotracker Deep Red FM (Thermo Fisher Scientific). Mitotracker was prepared according to the manufacturer's instructions. A 150 nM solution of Mitotracker was prepared in respective medium (control, MM, EMM1, etc.). Additionally, NucBlue (Thermo Fisher Scientific) was used to visualize cell nuclei. NucBlue was prepared by adding two drops per milliliter of 150 nM Mitotracker solution (described above). Organoids were washed twice using 166 μL of RPMI 1640 basal medium, then 166 μL of the Mitotracker/NucBlue solution was added to achieve a final concentration of 100 nM. Organoids were incubated for 30 min at 37 °C and 5% $CO_2$. Organoids were then washed twice using their respective medium (control, MM, EMM1, etc.) and transferred to a chambered coverglass slide (Cellvis) using a cut 200 μL pipette tip. Images were acquired using a Cellvivo microscope (Olympus). Samples were excited at 465 nm excitation and 630 nm emission was collected (for Mitotracker). Concurrently, samples were excited at 360 nm and emission was collected at 460 nm (for NucBlue). Data was processed using Fiji. An auto threshold was generated to measure mitochondrial area surrounding each nucleus.

### Raman microscopy

The Raman spectra of the organoids were acquired by using a Renishaw inVia Confocal Raman spectrometer connected to a Leica microscope (Leica DMLM, Leica Microsystems, Buffalo Grove, IL, USA). A 785 nm near-IR laser, Nikon Flour 60× NA = 1.00 water immersion objective lens, and 1000 ms exposure time with the average number of 100 accumulations were used for the data acquisition of each scanning position of the organoids. To circumvent a strong background signal, a quartz slide (Chemglass Life Sciences, NJ, USA) was used as a substrate for the Raman spectra acquisition. Organoids were collected on Day 30 for analysis.

### Compound treatments

Ondansetron Hydrochloride (Sigma) was prepared at 200 μM in DMSO and was further diluted in DMEM/F12 before being sterile-filtered via a 0.22 μm PVDF filter (Sigma). Ondansetron was applied to heart organoids at final concentrations of 1 μM, 10 μM, and 100 μM in EMM2/1 medium, and was applied from Day 9 to Day 20 of culture. Organoids were collected on Day 30 for analysis. Doxorubicin Hydrochloride (Sigma) was diluted to 1 mM in DMEM/F12 and was applied to heart organoids at a final concentration of 10 μM for 48 h from either Day 13 to Day 15 or from Day 28 to Day 30 of culture. 4-Diethylaminobenzaldehyde (DEAB) (Sigma) was prepared at 1 M in DMSO, diluted further to 10 mM using DMSO, and then finally diluted to 1 mM in DMEM/F12. Retinoic Acid (RA) (Sigma) was prepared at 1 M in DMSO and diluted to 100 μM in DMEM/F12. Diluted solutions of DEAB and RA were sterile-filtered via 0.22 μm PVDF filters (Sigma). DEAB was applied to heart organoids at a final concentration of 10 μM. RA was applied to heart organoids at a final concentration of 1 μM. DEAB, RA, and DEAB + RA were applied to heart organoids from Day 20 to Day 30 of culture using the EMM2/1 strategy. Organoids were collected on Day 30 for analysis.

### Statistics and reproducibility

GraphPad Prism 9 and Excel were used for all analyses. All data presented a normal distribution. Statistical significance was evaluated

using one-way ANOVA with Tukey, Dunnett, and Brown-Forsyth and Welch post-test corrections, or using unpaired t-tests, when appropriate ($p < 0.05$). All data presented as mean ± s.e.m. Specific number of independent organoids used and number of independent experiments (batches) for every experiment is indicated in figure legends. If not explicitly stated, images and data presented should be assumed to be from the iPSC L1 cell line.

## Reporting summary

Further information on research design is available in the Nature Portfolio Reporting Summary linked to this article.

## Data availability

scRNA-Sequencing data sets have been deposited in the National Center for Biotechnology Information Gene Expression Omnibus repository under accession code "GSE218582 [https://www.ncbi.nlm.nih.gov/geo/query/acc.cgi?acc=GSE218582]". All other data generated and/or analyzed in this study are provided in the published article and its supplementary information files. Any additional requests for information can be directed to, and will be fulfilled by, the corresponding author. Source data are provided with this paper.

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

## Acknowledgements

We wish to thank the IQ and MSU Advanced Microscopy Cores for access to confocal microscopes, and the MSU Genomics Core for sequencing services. We also wish to thank all members of the Aguirre Lab for their valuable comments and advice. Work in Dr. Aguirre's laboratory was supported by startup funds from MSU, the NIH under award numbers K01HL135464, R01HL151505, by the American Heart Association under award number 19IPLOI34660342 and by the Spectrum-MSU Foundation. Work in Dr. Zhou's laboratory was supported by startup funds from Washington University in St. Louis and NIH grants R01EB025209, R01HL156265, and R21EB03268401A1. Work in Dr. Qiu's laboratory was supported by startup funds from MSU and the U.S. National Science Foundation (1808436, 1918074). Work in Dr. Park's lab was supported by startup funds from MSU and by MSU's Discretionary Funding Initiative.

## Author contributions

B.V. and A.A. designed all experiments and conceptualized the work. B.V. performed all experiments and data analysis. A.R. performed data analysis, scRNA-seq data analysis, and cell and organoid culture. Al.K., P.M, C.O., A.H., Y.L.I., and A.H.W. performed cell and organoid culture. Ar.K., S.P., V.P., and S.B. performed scRNA-seq data analysis. F.W. and C.Z. performed optical coherence tomography experiments and data analysis. A.J. and Z.Q. performed Raman spectroscopy experiments and data analysis. A.L. provided chemicals and reagents. B.V. and A.A. wrote the manuscript.

## Competing interests

The authors declare no competing interests.
