## [Peer review file · Nature Communications]

REVIEWER COMMENTS

Reviewer #1 (Remarks to the Author):

In the manuscript Volmer et al describe a patterned human heart tube organoid model generated by pluripotent stem cell self-assembly. This manuscript depicts the extension of the culture for additional 10 days of the organoid model from the same lab published 2021 (PMID: 34446706), with the use of different media aimed to improve maturation.

It features beautiful imaging, including innovative ramen microscopy data, optical coherence tomography and molecular data, especially in-depth analysis of single cell RNA Sequencing of organoids. Functional evaluation is restricted to calcium and voltage imaging.

One key narrative is that maturation especially under EMM2/1 condition improves patterning of the cardiac organoid, which the authors claim as resulting from an endogenous retinoic acid gradient originating at the atrial pole, in an "proepicardial organ". While this is an interesting finding, this conclusion is not entirely convincing based on the data provided.

- The authors describe in Figure 6 "the formation of well-developed atrial and ventricular chambers by self-organization" with significantly more. However, in In Figure 7 of the previous manuscript (PMID: 34446706) the authors already depict well-evolved " ventricular (MYL2, green) and atrial (MYL7, red) chamber formation" and "producing heart organoids with well-defined regions of epicardial tissue adjacent to myocardial tissue".

Surprisingly, in this manuscript there is in the control condition 0% percent of organoids with a/p patterning. Is the existing patterning at day 15 lost during culture? This would speak rather for a degenerative process in the control condition, which makes it a questionable control to depict an positive effect of the maturation conditions like EMM 2/1. Please discuss and include into this evaluation day 15 data.

- The authors describe in figure 7 a ALDH1A2+ TBX18+ area, which is significantly bigger in EMM 2/1 organoids. This observation is utilized for the conclusion that patterning resulting from an endogenous retinoic acid gradient originating at the atrial pole, thus as the mechanisms. However, to my understanding there should at least be a clear correlation between the patterning described in figure 6 and the gradient in figure 7, which does not exist: The patterning of MM and EMM is better than in the control condition, while the ALDH1A2+ TBX18+ area is smaller than in the control, questioning the conclusion.

- To proof that the retinoic acid signaling leads to the patterning, experiments conducted adding diethylaminobenzaldehyde (DEAB) during the maturation period would strongly underscore the conclusion.

Authors describe increased maturation compared to control condition. However, if this is of added value and improves the function of the model compared to utilizing the model at day 15, remains unexplored.

- The authors state that they see more and bigger mitochondria, and describe changes in expression of metabolic pathways. However, there is in all conditions but control a very obvious accumulation of lipid droplets (see TEM pictures), which actually might indicate a rather a hyperlipidemic states, because of the delivery of surplus metabolic substrates. There is no evaluation of the consumption of energy (e.g of glucose / lactate concentration) or oxygen consumption (experiments like seahorse, as presented in the previous experiment). Thus, it remains unclear without if there is really more oxidative metabolism.

- While other manuscripts analyzing maturation conditions proved the impact of the increased maturation e.g. for disease modelling or tox screening (e.g. PMID: 326979, the authors leave this open in the presented manuscript. As such, the significance of the improved maturation is unclear. Methodological concerns:

- Figure 1H, mRNA expression of key sarcomeric genes involved in cardiomyocyte maturation between days 20 and 30 of culture is described as: “qRT-PCR revealed the differential expression of hallmark sarcomeric cardiomyocyte genes from day 20 to day 30 as expected, indicating cardiomyocyte function remained fundamentally unaffected throughout different developmental induction conditions at a general level (Fig. 1h).” However, for the condition EMM2/1 there is e.g. an almost complete loss of the ventricular sarcomeric transcript MYH7 for EMM2/1 over time and an almost complete loss of MYL2 for the control condition, in stark contrast to scRNA data. Please clarify this result, e.g. with western blot, and discuss the discrepancies.
- T-tubule quantification: The quantification of T-Tubules relies entirely on wga staining, which is not a specific stain for t-Tubules. Compared to others, the percent of the cell covered with t-tubules is very high. Another method would be strongly recommended to verify this finding, e.g. staining with caveolin or analysis of the TEM pictures utilized for quantification of mitochondria.
- hiPSC cell lines utilized: In contrast the previous manuscript, only two cell lines are referenced. One of these cell lines (AICS-0061-036) is fluorescently tagged which would interfere with IF, which suggests that mainly one in-house cell line was utilized. Regarding reproducibility and traceability, the manuscript would hugely profit from registration of the in-house cell line e.g. at hPSCreg, providing the mentioned stem cell QC data. Otherwise, please reference other manuscripts describing more in detail the cell line (donor data like gender, hiPSC karyotype, reprogramming method etc) or provide these details as supplemental information.
- Replication: Authors state: “Technical and experimental replicates performed are noted in figure legends”. However, only number of organoids are mentioned, which are technical and not experimental replicates. Please describe individual experiments (only indicated for the beating % across different experiments). Please specify if key findings (e.g. IF) were reproducible in several batches or even better in several cell lines. While scRNAseq on pooled organoids from one batch is reasonable, validation of some key findings or pathways with quantitative RT-PCR or western plot validation across 3 batches or even better more than one cell line would improve the reliability of results.
- Please provide explanation/legends for the supplemental movies.

Reviewer #2 (Remarks to the Author):

Volmert et al. developed a method to induce maturation of cardiac organoids which they had previously developed (Lewis-Israeli et al. Nat Commun. 2021), and analyzed them at the cellular level, the molecular level, and the functional level. As the authors describe in the Discussion section, the previous human heart models do not possess the true complexity of the in utero human heart, generating limitations for use in analysis of human heart development and disease. The current model constructed by the authors may show morphological and physiological maturation, and has the potential to become a better model than previous models. However, due to the inaccuracy of molecular-level assessment in experiments involving single-cell RNA-seq analysis, the current manuscript fails to present data to support the authors’ conclusions and fails to demonstrate the effectiveness of the model. Improvement points are shown below.

1. In Figure 2, the authors analyzed single-cell RNA-seq data, but we do not know how many biological replicates each condition consists of. To show the reproducibility of their model, they should show that multiple samples in each condition have similar profiles (cell-type proportion, cell-type expression profiles).
2. In Figure 2, the authors performed UMAP analysis and clustering of single-cell RNA-seq data separately for each condition, but this resulted in a loss of consistency in cell type classification in each condition. They should integrate all the data of each condition using Seurat, Harmony, etc.,

perform cell type classification, and show the distribution of each cell type for each condition and for each biological replicate. In addition, after classifying into each cell type, it is necessary to discuss the molecular mechanisms described in Figure 4 and later at the cellular level.

3. In Extended Data Figure 2, the authors used data of embryonic heart tissue at gestational day 45 in the human cell atlas. The heatmap shown in Extended Data Figure 2b does not allow us to understand the content at the single cell level. It is necessary to properly show the relationship using UMAP plot at the single-cell level and demonstrate that the gene expression profiles of actual human samples and human cardiac organoids are close at the single-cell level.

4. Figure 3a shows gene expression patterns for each cell type sorted by single-cell RNA-seq, but the annotation of this cell type may not be correct. For example, most of stromal cells do not express stromal marker genes. It is very strange that the definition of cell type is different for each condition. The authors need to integrate all the data as shown above (by adding multiple biological replicates for each condition), classify the cell types, and then analyze the cell-cell interactions shown in Figure 3b.

5. Figure 4 describes the maturation of myocardial mitochondria, Figure 5 describes the maturation of myocardial calcium transients, and Figure 6 describes the proepicardial population in cardiomyocytes. However, The data in Figures 4e,f and Extended Data Figure 3a do not appear to be cardiomyocyte data. I would like the authors to present evidence of cardiomyocyte data obtained by single-cell RNA-seq data. The data in Extended Data Figure 3c are the results of cell type classification by clustering for each condition, but the authors should show the results of integrated analysis instead of this.

6. Figure 5 shows electrophysiological maturation. Can the authors observe functional changes in forces such as contraction force during maturation of this human cardiac organoid model? Could this human heart organoid serve as an analytical model for drug response?

7. Figure 6 shows the emergence of WT1+TNNT2+ population and the differences in gene expression of atrial (MYL7) and ventricular (MYL2) cardiomyocytes. Extended Data Figure 4 shows the loss of PECAM1+ endothelial cells in the EMM1 and EMM2/1 conditions. Figure 7c shows the ALDH1A2+TBX18+ cardiomyocytes in the EMM2/1 condition. How are these represented by single-cell RNA-seq?

8. Figure 7 describes that endogenous retinoic acid gradient is responsible for the spontaneous anterior-posterior heart tube patterning in EMM2/1 organoids. However, to assert this conclusion, the authors need to investigate whether suppression of retinoic acid signaling abolishes anterior-posterior heart tube patterning.

Reviewer #3 (Remarks to the Author):

Referee Report on

A patterned human heart tube organoid model generated by pluripotent stem cell self-assembly
Summary

The authors describe a maturation process for human heart organoids that they claim results in the formation of heart-tube-like structures through self-organization. The maturation process involves exposure to previously described metabolic and hormonal signals and leads to changes in transcriptomics, cellular structure, and function over a 10-day period. The authors report the emergence of various cell types, including cardiomyocytes, valvular cells, epicardial cells, and conductance cells. Volmert et al. also claim the emergence of atrial and ventricular chambers with compact myocardial walls and a proepicardial organ. Finally, the authors explore anterior-posterior patterning in a self-organizing heart organoid.

Overall, in the present form, the manuscript contains a large proportion of preliminary data, overstatements, and factual inaccuracies, which limit its publication potential.

A. State of the art, evidence, reproducibility, conclusions, clarity, and suggested experiments

Major comments:

1) The title and the abstract are grossly misleading and inaccurate. For instance, in the abstract, the

authors write: "Our data reveals the emergence of atrial and ventricular cardiomyocyte populations, valvular cells, epicardial cells, proepicardial-derived cells, endothelial cells, stromal cells, conductance cells, and cardiac progenitors, all of them cell types present in the primitive heart tube." In the primitive heart tube, there are no valvular, epicardial, stromal, or conductance cells yet. The heart tube is a very transient structure (mouse E8.0-8.5, human 23-26 dpf) that starts looping rapidly and quickly turns into an embryonic multi-chamber heart at E8.5-9.0, after which, gradually, all these cell types emerge. Even atrial and ventricular cardiomyocytes, by definition, emerge only in a ballooning chambered heart after looping (human day 26-28 dpf). This also makes the title of the paper completely misleading because the analyzed organoids at day 30 or 35 are way beyond the primitive heart tube stage (usually around day 8-10 of differentiation in vitro). There are also other reasons why insisting on the heart tube stage is problematic in the points below. So, the basic developmental premise of the paper in its current form is factually wrong.

2) There are major concerns about the reproducibility and reliability of the data presented in the manuscript. The absence of biological replicates for almost all experiments raises questions about the representativeness of the results, and the information about the number of organoids per condition might not provide a clear picture of the reproducibility of the protocols. The lack of quantification in the functional data makes it difficult to determine its reproducibility. The origin of some data points is unclear, and the scRNA-seq being performed on day 34 while all other assays were performed on day 30 raises questions about the consistency of the study design. These issues raise concerns about the validity of the results and suggest a need for more biological replicates (not only technical) for almost all the experiments in the study. In addition, how does this protocol perform with different cell lines? This is a basic requirement today to demonstrate any utility of a new protocol.

3) Another concern is the validity of the claims made in the manuscript regarding the existence of various cell types in the heart organoids. The absence of ICC to confirm protein expression raises questions about the accuracy of the cell type identifications. The cluster assignment is not explained. Based on what markers? For instance, are SHOX2, HCN4, and CX45 used to identify conductance cells at all? IRX3/4 ventricular. NR2F2 for atrial. Random structural markers (which are known to change during early development and are expressed in both chambers) are solely used as A/V identifiers based on which in vivo data? The random assignment of clusters based on the mentioned markers is highly problematic. Are there non-cardiac cells? For instance, SOX2 and FOXA2 are used as stromal markers, but they also mark many non-cardiac cells. The authors could benefit from integrating human fetal scRNA-seq data to improve the reliability of their results. Additionally, performing stainings on sections instead of whole organoids would allow for a clearer understanding of cell location and patterning, a major point of this study. For figures 2b and 3b. there is no proper rationale or conclusion given. There are no details on the cell dissociation, viability, and sequencing QC, and just the very basic 10x pipeline is used. Overall, the scRNAseq analysis is preliminary and requires a much deeper investigation.

4) The authors claim the presence of endothelial cells (ECs) lining the myocardial regions of all organoids, as seen in the extended data in figure 4. However, it is difficult to see this in the figure, and therefore, there are questions regarding the reproducibility of this observation.

5) In Fig.6a, the authors state, "gives rise to a well-developed secondary chamber", just because of the increase in the area? This appears to be an overstatement. The chambers are not properly characterized by the relevant markers. IRX3/4/HEY2 and NR2F2/HEY1 are standard TF markers in the field for V and A, respectively. These are completely ignored in the study. Similarly, the "compacted" myocardial muscle walls are claimed, but there is no data to support it with specific markers.

6) There is a question of whether the maturation protocols are making the organoids more mature or just killing non-cardiomyocyte populations of cells, leading to an increase in the percentage of cardiomyocytes. The observation that the organoids get smaller during maturation raises concerns about the loss of important cell populations, such as endothelial cells, and suggests a need for further investigation.

7) The sarcomere markers checked are the usual ones, but in terms of fold changes, it should be way higher – for instance, the KCNJ2 staining is not convincing (should have a way higher fold increase during maturation. Regarding metabolic maturation, the authors could show the protein levels of some of the cited metabolic markers, or at least compare their expression with GD45. Plus, some metabolic

assays would have completed the picture and made it more informative. The T-tubules marker selected it's not the best choice. Further, 4/10 tubules per cell are irrelevant - it should be 1 per sarcomere.

8) There are also overstatements regarding the functional data due to improper characterization of the action potentials (different subtype/maturation criteria missing) and the absence of statistics to show their prevalence in the organoids. Additionally, the use of calcium as the major readout without normalizing it by the number of cells in the recording region can lead to irrelevant results, as the density of cells in the region of recording can drastically affect the amplitude. Electrochemical data are really affected by how many CM there are, and therefore single-cell data are imperative. This raises questions about the reliability of the functional data and suggests a need for further investigation and clarification

9) The authors claim self-organization, but no data is really shown to demonstrate the process of cell sorting and patterning, and how it is potentially connected to in vivo developmental stages. The assembly of the heart tube definitely happens very differently.

10) To show any benefits of the platform, a direct comparison to other maturation protocols is missing, as many use the exact same factors in their metabolic treatments. Also, no application of this system has been demonstrated, in line with the preliminary nature of the study.

11) The most interesting aspect and novel aspect is the endogenous anterior-posterior patterning shown at the later stages. However, there are severe doubts about whether this is actually showing the same process as in the heart tube. As mentioned before, tube formation and patterning happen much earlier, and the observed phenomenon could be a later patterning event. In any case, the observation would benefit from further characterization, blocking the endogenous RA signaling, using downstream RA signaling markers, etc.

Minor comments:

1) The introduction is, in general, well-written, and most of the relevant papers have been cited. However, the referencing format is inconsistent and has mistakes.

2) Fig1b: image quality insufficient. Images don't support statement (lines 127-129, 'continue to grow until d30').

3) Fig1c: would be more informative to quantitatively support the statement that organoids are becoming more elliptical over time.

4) Fig1g - sarcomere length quantification: not clear which data points belong to the same cardioid/same sections. Not described sufficiently how data was analyzed. Length measurement depends a lot on the cutting angle of the sarcomere (the more oblique). Is the sarcomere length consistent within a fiber? Are the differences within a fiber larger than between conditions?

5) In the comparison to the Human Cell Atlas there seems to be more of a selection of the organoid to VCM, which is not commented on but rather ignored. Also, some more interesting genes to address all of the cell types and compare them to the data set would be useful here.

6) Extended fig.2b wrong labeling of condition EMM2/1

7) Figure titles are too generic both in the figures and in the text (ex. fig4).

8) Line 330 doesn't match with what is shown in fig.6a (focusing only on one condition but showing data for all of them).

9) Figure 5b,c: says the recordings are taken with and without isoproterenol, but not an indication of isoproterenol in the graph or indicated in the text.

10) Figure 1h, 4e: not clear what the values on the y-axis mean, is it the fold change?

B. Significance

There is potential significance in all aspects of this work, although the novelty is limited since the maturation treatment is known and applicability over other protocols has not been demonstrated. However, a bigger issue is the preliminary nature of the study, with basic statistics and biological replicates missing to support most of the statements robustly. Also, there are basic flaws in the developmental biology interpretation, which require a major rewrite of the paper.

REBUTTAL LETTER

We want to thank the reviewers for their time, expertise, and constructive criticism. We have taken all comments into deep consideration and present to you an updated and improved manuscript including a significant amount of new data that, in our opinion, addresses the concerns raised.

Note: Original reviewers' comments are presented in **black**, while our responses are written underneath in blue highlight.

Reviewer #1 Comments:

1. In the manuscript Volmer et al describe a patterned human heart tube organoid model generated by pluripotent stem cell self-assembly. This manuscript depicts the extension of the culture for additional 10 days of the organoid model from the same lab published 2021 (PMID: 34446706), with the use of different media aimed to improve maturation. It features beautiful imaging, including innovative ramen microscopy data, optical coherence tomography and molecular data, especially in-depth analysis of single cell RNA Sequencing of organoids. Functional evaluation is restricted to calcium and voltage imaging. One key narrative is that maturation especially under EMM2/1 condition improves patterning of the cardiac organoid, which the authors claim as resulting from an endogenous retinoic acid gradient originating at the atrial pole, in an "proepicardial organ". While this is an interesting finding, this conclusion is not entirely convincing based on the data provided.

We thank the reviewer for the positive and constructive feedback, and we further address main concerns in all points below.

2. The authors describe in Figure 6 "the formation of well-developed atrial and ventricular chambers by self-organization". However, in Figure 7 of the previous manuscript (PMID: 34446706) the authors already depict well-evolved "ventricular (MYL2, green) and atrial (MYL7, red) chamber formation" and "producing heart organoids with well-defined regions of epicardial tissue adjacent to myocardial tissue". Surprisingly, in this manuscript there is in the control condition 0% percent of organoids with a/p patterning. Is the existing patterning at day 15 lost during culture? This would speak rather for a degenerative process in the control condition, which makes it a questionable control to depict an positive effect of the maturation conditions like EMM 2/1. Please discuss and include into this evaluation day 15 data.

We appreciate the concerns raised by the reviewer but there is a misunderstanding at work in this case. Figure 7 of Lewis-Israeli et al. corresponds to an experimental model of diabetes-induced congenital heart disease. As such, the organoids depicted there are exposed to different glucose concentrations than the ones provided here, and the observations cited above should only be taken in the context of that low glucose vs. high glucose comparison (that is, low glucose organoids exhibited better patterning than high glucose ones at day 15). Basically, the organoids the reviewer is referring to (with improved patterning) are not the same shown here as control organoids. Although early use of low glucose improved patterning, a number of other issues arised when low glucose was used. The current protocol includes lessons learned from that model and incorporates low glucose conditions in EMM1 and EMM2/1 at more optimized, later timepoints.

The ventricular and atrial chamber formation in that protocol, even for low glucose day 15 organoids, is much less sophisticated compared to what is shown here. EMM2/1 organoids exhibit better defined chamber walls, much larger chamber volumes and better overall organization compared to that shown in the experimental low glucose day 15 organoids in Lewis-Israeli et al, and we support this data with multiple new markers for atrioventricular specification and scRNAseq data (see **Fig. 6**). Regarding the “*well-defined regions of epicardial and myocardial tissue*” reported for experimental low glucose day 15 organoids in Lewis-Israeli et al, we expand on that significantly in this manuscript and show that organoids in all conditions display reproducible and well-organized regions of epicardial and myocardial tissue, including a proepicardial organ in EMM2/1 organoids. We have also removed the figure with 0% a/p patterning for control organoids as this representation was misleading and lacked granularity. Control organoids exhibit a certain degree of patterning, but it is more random and not as advanced as EMM2/1 patterning. We have also increased the number of experiments and have updated Fig 6 and its associated extended data to reflect these changes.

3. The authors describe in figure 7 a ALDH1A2+ TBX18+ area, which is significantly bigger in EMM 2/1 organoids. This observation is utilized for the conclusion that patterning resulting from an endogenous retinoic acid gradient originating at the atrial pole, thus as the mechanisms. However, to my understanding there should at least be a clear correlation between the patterning described in figure 6 and the gradient in figure 7, which does not exist: The patterning of MM and EMM is better than in the control condition, while the ALDH1A2+ TBX18+ area is smaller than in the control, questioning the conclusion.

We thank the reviewer for their valuable interpretation and critique of the data. We have repeated all experiments relating to ALDH1A2 and TBX18 to increase n, and have reinforced data that there is almost no positive ALDH1A2+ TBX18+ area in Control, MM, and EMM1 organoids (**Fig. 7**). There is also no spontaneous patterning, as the epicardium in these conditions is covering generally the surface of the whole organoid, rather than predominantly the atrial pole, as seen in EMM2/1 (**Fig. 7e**). Notice that in Figure 6a, the lack of proepicardial organ (PEO) is not immediately obvious in MM and EMM1 due to the co-staining and presentation as a merged image, Fig 7e offers a much better view at the distribution of the PEO. Atrioventricular patterning exhibits the same characteristics: in ctrl, MM and EMM1 atrial and ventricular separation is poor compared to EMM2/1 (see particularly Fig 6f, where the atrial and ventricular populations superimpose, but in EMM2/1 they clearly exclude each other, plus larger chambers). In summary, EMM2/1 organoids exhibit a PEO covering the atrial pole with an opposing ventricular chamber, all characteristics of the patterned heart tube, while organoids in other conditions lack this sharp organization.

4. To proof that the retinoic acid signaling leads to the patterning, experiments conducted adding diethylaminobenzaldehyde (DEAB) during the maturation period would strongly underscore the conclusion. Authors describe increased maturation compared to control condition. However, if this is of added value and improves the function of the model compared to utilizing the model at day 15, remains unexplored.

We thank the reviewer for this excellent suggestion. We have now performed experiments to manipulate the RA gradient as suggested, and this data is presented in fig 7h, I, j. We added DEAB to heart organoids during the induction period and observed a marked reduction in

patterning from organoids in the EMM2/1 condition. We also observed that the addition of exogenous retinoic acid did not lead to noticeable differences in patterning in EMM2/1, suggesting that EMM2/1 organoid RA production is sufficient for adequate patterning.

We also thank the reviewer for their comments surrounding Day 15 organoids. Day 15 organoids were explored and assessed thoroughly in our previous paper (Lewis-Israeli et al., 2021). As mentioned in the comments above, control organoids in this manuscript are not the low glucose organoids presented in Lewis-Israeli et al.

5. The authors state that they see more and bigger mitochondria, and describe changes in expression of metabolic pathways. However, there is in all conditions but control a very obvious accumulation of lipid droplets (see TEM pictures), which actually might indicate a rather a hyperlipidemic states, because of the delivery of surplus metabolic substrates. There is no evaluation of the consumption of energy (e.g of glucose / lactate concentration) or oxygen consumption (experiments like seahorse, as presented in the previous experiment). Thus, it remains unclear without if there is really more oxidative metabolism.

We thank the reviewer for this important comment. We have performed an evaluation of oxygen consumption by using Seahorse assays on organoids in all conditions (**Fig. 4f-j**). We show that EMM2/1 organoids display higher levels of basal respiration, maximal respiration, and percent respiratory capacity compared to Control. These functional increases were not observed for MM or EMM1 organoids.

6. While other manuscripts analyzing maturation conditions proved the impact of the increased maturation e.g. for disease modelling or tox screening (e.g. PMID: 326979, the authors leave this open in the presented manuscript. As such, the significance of the improved maturation is unclear.

We thank the reviewer for this comment. The term maturation in the context of this manuscript and those cited is not the same and this presents a problem of semantics. Our protocol refers to developmental maturation, not maturation to an adult-like state, as the term is typically used. We have introduced the term developmental induction to preserve the meaning in the context of development and use in conjunction with developmental maturation to mark the difference. In the developmental context of our manuscript, the significance of our induction protocols is increased physiological relevance and similitude to the embryonic heart, allowing for potentially better embryonic disease modeling or pharmacological safety testing, for example. To provide biological significance in this respect, we have added a new set of experiments (Fig 8) that investigate the effects of ondansetron exposure (a leading drug prescribed off-label to pregnant women during pregnancy to treat nausea) that has been associated with ventricular septal defects. Our EMM2/1 organoids recapitulate the epidemiological clinical findings linking ondansetron exposure to VSDs. We observed that ondansetron exposure during heart organoid development led to severely diminished expression of ventricular cells (**Fig. 8a-d**). To further support this, we performed qPCR for *MYL2* for Control and EMM2/1 organoids at day 30 for organoids from 3 different pluripotent stem cell lines (L1, BYS0111, and H9) and observed the same increase in *MYL2* expression across all experiments (**Extended Data Fig. 10a, shown below**). We also show that ondansetron leads to progressive electrophysiological abnormalities in EMM2/1 organoids (**Fig. 8e-j**).

7. Figure 1H, mRNA expression of key sarcomeric genes involved in cardiomyocyte maturation between days 20 and 30 of culture is described as: “*qRT-PCR revealed the differential expression of hallmark sarcomeric cardiomyocyte genes from day 20 to day 30 as expected, indicating cardiomyocyte function remained fundamentally unaffected throughout different developmental induction conditions at a general level (Fig. 1h).*” However, for the condition EMM2/1 there is e.g. an almost complete loss of the ventricular sarcomeric transcript MYH7 for EMM2/1 over time and an almost complete loss of MYL2 for the control condition, in stark contrast to scRNA data. Please clarify this result, e.g. with western blot, and discuss the discrepancies.

We thank the reviewer for this comment. We have repeated all qPCR experiments with more added independent experiments (**Fig. 1h**) and with a higher total number of organoids (n=7-14 organoids per gene per time point per condition). We show that the genes *MYL2*, *MYH7*, *MYL7*, and *MYH6* exhibit dynamic expression levels dependent on maturation condition. Although changes are not very large (except for *MYL2*), they are consistent with findings on atrial and ventricular specification in later figures. It is important to mention that the data are normalized against control day 20 organoids, thus reduction in specific sarcomeric genes (such as *MYH7* in EMM2) only illustrate fine tuning/regulation in that context, as many other ventricular genes are highly expressed. The scRNAseq data was performed later for all organoids at day 34 so there is no discrepancy, the differences observed in gene expression levels between qPCR data and scRNAseq data could simply be due to the dynamic nature of gene expression over time, as is shown in the qPCR data.

8. T-tubule quantification: The quantification of T-Tubules relies entirely on wga staining, which is not a specific stain for t-Tubules. Compared to others, the percent of the cell covered with t-tubules is very high. Another method would be strongly recommended to verify this finding, e.g. staining with caveolin or analysis of the TEM pictures utilized for quantification of mitochondria.

We have followed the reviewers advice and performed caveolin-3 staining (**Fig. 4g,h**) and show that EMM2/1 organoids display a marked increase in Cav-3+ area compared to Control organoids, confirming our WGA staining results. We have moved our WGA staining analysis to **Supplementary Fig. 23**. Please notice that WGA staining has been used many times in the past by us and others to support formation of t-tubules too (Lewis-Israeli et al., Nature Comms, 2021; Crossman et al., PLOS One, 2011; Heinzel et al., Circulation Research, 2008).

9. hiPSC cell lines utilized: In contrast the previous manuscript, only two cell lines are referenced. One of these cell lines (AICS-0061-036) is fluorescently tagged which would interfere with IF, which suggests that mainly one in-house cell line was utilized. Regarding reproducibility and traceability, the manuscript would hugely profit from registration of the in-house cell line e.g. at hPSCreg, providing the mentioned stem cell QC data. Otherwise, please reference other manuscripts describing more in detail the cell line (donor data like gender, hiPSC karyotype, reprogramming method etc) or provide these details as supplemental information.

We thank the reviewer for this comment. We have repeated all key findings in 2 additional cells lines (one ESC, one iPSC) and found excellent reproducibility of the findings. We have also updated the methods to describe the three cell lines (L1, BYS0111, and H9) in detail. The

previously mentioned AICS-0061-036 was mentioned by mistake and was not used in this manuscript, all mentions to it have been removed.

10. Replication: Authors state: “Technical and experimental replicates performed are noted in figure legends”. However, only number of organoids are mentioned, which are technical and not experimental replicates. Please describe individual experiments (only indicated for the beating % across different experiments). Please specify if key findings (e.g. IF) were reproducible in several batches or even better in several cell lines. While scRNAseq on pooled organoids from one batch is reasonable, validation of some key findings or pathways with quantitative RT-PCR or western blot validation across 3 batches or even better more than one cell line would improve the reliability of results.

We thank the reviewer for the valuable comment. In our experimental setup, individual organoids constitute independent experiments as they are grown completely independently from one another and develop independently in independent wells. Regardless of that, we have added additional organoids to all key findings to improve reproducibility and made sure all experiments represent at least three independent plates prepared on different dates.

Additionally, we have repeated key immunofluorescence findings and RT-PCR findings in three cell lines (L1, BYS0111, and H9) (**Extended Data Figs. 5, 8, and 9**). **Extended Data Fig. 8** is shown below. We show that the increase in ALDH1A2+TBX18+ from EMM2/1 organoids compared to Control organoids is reproducible across the three cell lines. We also show that NR2F2 and MYL3 chamber identity between Control and EMM2/1 organoids is similarly represented across the three cell lines (**Extended Data Fig. 5**), and that many key genes such as *ALDH1A2*, *MYL2*, and *PPARGC1A* show similar gene expression behavior across Control and EMM2/1 organoids across all three cell lines (**Extended Data Fig. 9**).

11. Please provide explanation/legends for the supplemental movies.

All supplementary movies are now provided with their associated legends.

Reviewer #2 comments:

12. Volmert et al. developed a method to induce maturation of cardiac organoids which they had previously developed (Lewis-Israeli et al. Nat Commun. 2021), and analyzed them at the cellular level, the molecular level, and the functional level. As the authors describe in the Discussion section, the previous human heart models do not possess the true complexity of the in utero human heart, generating limitations for use in analysis of human heart development and disease. The current model constructed by the authors may show morphological and physiological maturation, and has the potential to become a better model than previous models. However, due to the inaccuracy of molecular-level assessment in experiments involving single-cell RNA-seq analysis, the current manuscript fails to present data to support the authors' conclusions and fails to demonstrate the effectiveness of the model. Improvement points are shown below.

We thank the reviewer for acknowledging the potential of this manuscript and for their constructive criticism. Comments are addressed in full below.

13. In Figure 2, the authors analyzed single-cell RNA-seq data, but we do not know how many biological replicates each condition consists of. To show the reproducibility of their model, they should show that multiple samples in each condition have similar profiles (cell-type proportion, cell-type expression profiles).

We thank the reviewer for this comment. Indeed, the scRNA-seq data consists of 4 pooled organoids per condition (the Methods section has been updated with this information). This is a common approach in the field and has been used by many other recent publications and authors before us (Hofbauer et al., Cell, 2021; Camp et al., PNAS, 2015; Rossi et al., Cell Stem Cell, 2021). However, and to show the reproducibility of our model, we have now performed qPCR on a panel of important organoid genes across organoids from three independent cell lines (L1, BYS0111, and H9) and across 7 independent experiments (**Extended Data Fig. 9**). All genes show a high degree of reproducibility across multiple samples and multiple cell lines. Additionally, we reperformed all major experiments in the manuscript to show that our results are reproducible across independent experiments, including organoids from three total cell lines (**Extended Data Figs. 5 and 8**).

14. In Figure 2, the authors performed UMAP analysis and clustering of single-cell RNA-seq data separately for each condition, but this resulted in a loss of consistency in cell type classification in each condition. They should integrate all the data of each condition using Seurat, Harmony, etc., perform cell type classification, and show the distribution of each cell type for each condition and for each biological replicate. In addition, after classifying into each cell type, it is necessary to discuss the molecular mechanisms described in Figure 4 and later at the cellular level.

We thank the reviewer for this important comment. We integrated all the data of each condition using Seurat and Harmony (**Fig. 2a**) and have used this integrated dataset throughout the manuscript. We then performed cell type classification and showed the distribution of each cell types for each condition, as suggested, following integration (**Fig. 2b-d**). We have also provided a discussion on the molecular mechanisms described in Figure 4 and later at the cellular level.

15. In Extended Data Figure 2, the authors used data of embryonic heart tissue at gestational day 45 in the human cell atlas. The heatmap shown in Extended Data Figure 2b does not allow us to understand the content at the single cell level. It is necessary to properly show the relationship using UMAP plot at the single-cell level and demonstrate that the gene expression profiles of actual human samples and human cardiac organoids are close at the single-cell level.

We thank the reviewer for this critical comment, and we agree that this data was preliminary in the original manuscript. We have taken human embryonic heart scRNAseq data from two different published papers (PMID: 31835037; PMID: 30759401) and have integrated this data with our heart organoid dataset to show a UMAP plot (**Extended Data Fig. 2b**), keeping the annotations from the Asp et al. dataset. Through this, we show that our heart organoids and human samples possess similar cell types and expression profiles at the single-cell level. We investigated this further in **Extended Data Fig. 2c** to show that Control and EMM2/1 organoids are closer to Week 6 human hearts and that MM and EMM1 organoids are closer to Week 7-13 human hearts at the transcriptional levels. Additionally, we show individual gene expression feature plots for all major clusters across the heart organoid dataset and the two human heart datasets in **Extended Data**

Fig. 3 and in **Supplementary Fig. 7**, which shows a high degree of gene expression overlap and specificity across all scRNAseq datasets and further supports the hypothesis that human heart organoids can recapitulate significant features of human heart development. Feature plots from **Extended Data Fig. 3a and b** (for ACMs and VCMs) have also been performed.

16. Figure 3a shows gene expression patterns for each cell type sorted by single-cell RNA-seq, but the annotation of this cell type may not be correct. For example, most of stromal cells do not express stromal marker genes. It is very strange that the definition of cell type is different for each condition. The authors need to integrate all the data as shown above (by adding multiple biological replicates for each condition), classify the cell types, and then analyze the cell-cell interactions shown in Figure 3b.

We thank the reviewer for highlighting this oversight, and we agree that the initial marker gene selection needed improvement. As described above, we integrated all the data and classified the cell types using the top differentially expressed marker genes per cluster (**Fig. 2c** and **Fig. 3a**, both shown below). Additionally, cell-cell communication networks were re-analyzed, and the data is presented in **Fig. 3b** and **Supplementary Figs. 10-13**.

17. Figure 4 describes the maturation of myocardial mitochondria, Figure 5 describes the maturation of myocardial calcium transients, and Figure 6 describes the proepicardial population in cardiomyocytes. However, The data in Figures 4e, f and Extended Data Figure 3a do not appear to be cardiomyocyte data. I would like the authors to present evidence of cardiomyocyte data obtained by single-cell RNA-seq data. The data in Extended Data Figure 3c are the results of cell type classification by clustering for each condition, but the authors should show the results of integrated analysis instead of this.

We thank the reviewer for highlighting this important element to the paper. We have re-analyzed all scRNAseq data using the integrated dataset, and show evidence of cardiomyocyte data in **Figures 4j, 4k, 5d, and 6h-j**. This is also shown to an extent in **Extended Data Fig. 3a,b**. These data show the increasing expression levels of various genes within cardiomyocyte clusters at the single cell level across our maturation conditions.

18. Figure 5 shows electrophysiological maturation. Can the authors observe functional changes in forces such as contraction force during maturation of this human cardiac organoid model? Could this human heart organoid serve as an analytical model for drug response?

We haven't measured contraction forces by direct physical means, but EMM2/1 organoids present larger and more sustained action potential peak depolarization, that correlate with stronger contraction forces. We do show now that our human heart organoid system can serve as an analytical model for drug responses (see new data on ondansetron in **Fig. 8** and comment #6 above).

19. Figure 6 shows the emergence of WT1+TNNT2+ population and the differences in gene expression of atrial (MYL7) and ventricular (MYL2) cardiomyocytes. Extended Data Figure 4 shows the loss of PECAM1+ endothelial cells in the EMM1 and EMM2/1 conditions. Figure 7c

shows the ALDH1A2+TBX18+ cardiomyocytes in the EMM2/1 condition. How are these represented by single-cell RNA-seq?

We thank the reviewer for this important comment. We integrated the data from each condition and now show all gene expression discussed in this comment (WT1, TNNT2, MYL7, MYL2, PECAM1, ALDH1A2, TBX18) as well as many additional genes of interest in figures throughout the paper, such as **Fig. 2d**, **Fig. 3a**, **Fig. 6h-j**, and **Fig. 7d**, among others.

20. Figure 7 describes that endogenous retinoic acid gradient is responsible for the spontaneous anterior-posterior heart tube patterning in EMM2/1 organoids. However, to assert this conclusion, the authors need to investigate whether suppression of retinoic acid signaling abolishes anterior-posterior heart tube patterning.

We thank the reviewer for this excellent suggestion that was also raised by reviewer 1. As suggested, we suppressed retinoic acid signaling via a pharmacological approach through treating the organoids with the ALDH1A2 inhibitor called DEAB during the induction period. Please see comment #4 for more information, and **figure 7h-j** in the manuscript.

Reviewer #3 comments:

The authors describe a maturation process for human heart organoids that they claim results in the formation of heart-tube-like structures through self-organization. The maturation process involves exposure to previously described metabolic and hormonal signals and leads to changes in transcriptomics, cellular structure, and function over a 10-day period. The authors report the emergence of various cell types, including cardiomyocytes, valvular cells, epicardial cells, and conductance cells. Volmert et al. also claim the emergence of atrial and ventricular chambers with compact myocardial walls and a proepicardial organ. Finally, the authors explore anterior-posterior patterning in a self-organizing heart organoid. Overall, in the present form, the manuscript contains a large proportion of preliminary data, overstatements, and factual inaccuracies, which limit its publication potential.

We thank the reviewer for their time and their criticism, with critical points addressed below.

21. The title and the abstract are grossly misleading and inaccurate. For instance, in the abstract, the authors write: "Our data reveals the emergence of atrial and ventricular cardiomyocyte populations, valvular cells, epicardial cells, proepicardial-derived cells, endothelial cells, stromal cells, conductance cells, and cardiac progenitors, all of them cell types present in the primitive heart tube." In the primitive heart tube, there are no valvular, epicardial, stromal, or conductance cells yet. The heart tube is a very transient structure (mouse E8.0-8.5, human 23-26 dpf) that starts looping rapidly and quickly turns into an embryonic multi-chamber heart at E8.5-9.0, after which, gradually, all these cell types emerge. Even atrial and ventricular cardiomyocytes, by definition, emerge only in a ballooning chambered heart after looping (human day 26-28 dpf). This

also makes the title of the paper completely misleading because the analyzed organoids at day 30 or 35 are way beyond the primitive heart tube stage (usually around day 8-10 of differentiation in vitro). There are also other reasons why insisting on the heart tube stage is problematic in the points below. So, the basic developmental premise of the paper in its current form is factually wrong.

We thank the reviewer for highlighting these critical points. We agree with the reviewer on the transient state of the heart tube and the difficulty of specifically pinpointing the stage in which our organoids exist in relation to the heart tube. Thus, we have changed the title to reflect the status of our organoids as “primitive heart”, which is more general. Regarding cell populations, In the context of the post-heart tube, primitive heart stage, in the period of about 4-6 weeks in humans, abundant human embryonic heart data (Asp et al, 2019, Cell; Cui et al., 2019, Cell Reports) support the similitude of our organoids to those structures both transcriptionally as well as in cell composition, as shown in Extended Figure 2. On a separate note, we would like to highlight that the purpose of this work is not to replicate human heart development 100%, but to create a useful bioengineered model of that system in vitro for research and translational applications. In the pursue of that goal we use approaches and technologies that sometimes depart from the specific biological developmental steps, however if the end result brings us closer to the goal, that is still a successful model (reverse engineering). An example of this are significant attempts at accelerating iPSC-derived cardiomyocyte maturation through chemical and biophysical means which accelerate cardiomyocyte maturation beyond normal physiology.

22. There are major concerns about the reproducibility and reliability of the data presented in the manuscript. The absence of biological replicates for almost all experiments raises questions about the representativeness of the results, and the information about the number of organoids per condition might not provide a clear picture of the reproducibility of the protocols. The lack of quantification in the functional data makes it difficult to determine its reproducibility. The origin of some data points is unclear, and the scRNA-seq being performed on day 34 while all other assays were performed on day 30 raises questions about the consistency of the study design. These issues raise concerns about the validity of the results and suggest a need for more biological replicates (not only technical) for almost all the experiments in the study. In addition, how does this protocol perform with different cell lines? This is a basic requirement today to demonstrate any utility of a new protocol.

We thank the reviewer for this comment and we agree that it is critical to prove the reproducibility and reliability of our protocol. Our results report independent organoids in all cases (biological replicates). Organoids are seeded independently of each other and develop completely independent of each other in individual wells of a 96-well plate. The organoids presented always correspond to multiple independent plates (at least 3, but frequently more) prepared on different dates, thus completely independent. All of this information is available in figure legends and in the statistics section of the materials and methods. Nonetheless, we agree that our data could be strengthened by repeating experiments through performing additional experiments from new batches of cells, and by repeating experiments using different cell lines. We have repeated experiments throughout the manuscript to yield more data. Examples of Figures with additional experiments include **Fig. 1h, Fig. 4e,f, 5a-c,e, Fig. 6a-e, and 7e,g**. Furthermore, we have repeated all key experiments using three different cell lines now, including both ESCs and iPSCs (L1, BYS0111, and H9) (**Extended Data Figs. 5, 8, and 9**). We believe this data provides ample

evidence of reproducibility across multiple fronts throughout the paper. Additionally, we have performed more functional experiments for calcium transients to yield data across 3 independent plates of organoids (n=12 organoids per condition) (**Fig. 5a**). We have quantified this data to show amplitude and frequency variations in the organoids dependent on maturation condition (**Fig. 5b,c**). We show many independent cardiomyocyte voltage traces from multiple organoids across the Control and EMM2/1 conditions (**Fig. 5f**), which is standard practice in the field. To clarify the origin of all data points, we have thoroughly updated the figure legends and the methods section to provide detailed information on these fronts. scRNA-seq was performed on day 34 due to the limitations imposed by the coordination of personnel across different labs and core facilities and working with live a large number of dissociated organoids/cells in a protocol that extends throughout most of the day. This does not affect in any way the study design across the manuscript. All scRNA-seq datasets were collected on the same day and processed in parallel.

23. Another concern is the validity of the claims made in the manuscript regarding the existence of various cell types in the heart organoids. The absence of ICC to confirm protein expression raises questions about the accuracy of the cell type identifications. The cluster assignment is not explained. Based on what markers? For instance, are SHOX2, HCN4, and CX45 used to identify conductance cells at all? IRX3/4 ventricular. NR2F2 for atrial. Random structural markers (which are known to change during early development and are expressed in both chambers) are solely used as A/V identifiers based on which *in vivo* data? The random assignment of clusters based on the mentioned markers is highly problematic. Are there non-cardiac cells? For instance, SOX2 and FOXA2 are used as stromal markers, but they also mark many non-cardiac cells. The authors could benefit from integrating human fetal scRNA-seq data to improve the reliability of their results. Additionally, performing stainings on sections instead of whole organoids would allow for a clearer understanding of cell location and patterning, a major point of this study. For figures 2b and 3b. there is no proper rationale or conclusion given. There are no details on the cell dissociation, viability, and sequencing QCs, and just the very basic 10x pipeline is used. Overall, the scRNAseq analysis is preliminary and requires a much deeper investigation.

We thank the reviewer for this comment, and we agree that our initial claims on cell types in heart organoids were preliminary. We have performed a much deeper investigation into this front as this was a concern for reviewer #2 too (see above comments in this respect). To reiterate, we performed additional IF stainings for a number of atrial and ventricular markers, as well as epicardial cells (see **Fig. 6** and **Extended Data Fig. 5**), VCMs (**Fig. 6** and **Extended Data Fig. 5**), EPCs (**Fig. 6** and **7**), and ECs (**Extended Data Fig. 7**), and for post tube-like patterning. Furthermore, we re-analyzed all scRNA-seq data and provide a significant number of markers to support each of the populations mentioned. The new markers used have been derived from published human embryonic datasets (Asp et al., Cui et al) after integration of the datasets. Our organoids exhibit a remarkable level of similitude to this human datasets at the 4-6 weeks stage (**Extended Figure 2, 3**). The cluster assignments are explained in the main text now, along with references using *in vivo* data, and we support this with **Fig. 2c,d**, **Fig. 3a**, **Extended Data Fig. 2b**, **Extended Data Fig. 3**, and **Supplementary Fig. 7**. **Extended Data Fig. 3** and **Extended Data Fig. 7** surround the integration of our organoid data with two embryonic scRNAseq datasets to further support our cluster identifications. I'm not sure what the reviewer means by "there is no proper rationale for Figs. 2b and 3b". In the text we have updated our conclusions for Figures 2b and 3b. The dissociation, viability, sequencing QCs and analysis pipeline have been updated in the Methods. As described to an extent in response to Reviewer 2's comments, we have

performed a major overhaul to the scRNAseq analysis, and these updates can be found in the following Figures throughout the text: **Fig. 2, Fig. 3, Extended Data Fig. 2b,c, Extended Data Fig. 3, Fig. 4j,k, Fig. 5d, Fig.6 h-j, Fig. 7d, and Supplementary Figs. 1-22.**

24. The authors claim the presence of endothelial cells (ECs) lining the myocardial regions of all organoids, as seen in the extended data in figure 4. However, it is difficult to see this in the figure, and therefore, there are questions regarding the reproducibility of this observation.

We thank the reviewer for pointing this out. We have already reported this in detail in Lewis-Israeli et al, 2021, Nat Comms. We now provide high magnification images of regions containing endothelial cells (PECAM1+) and cardiomyocytes (TNNT2+) for all conditions in **Extended Data Fig. 7d**. We have modified the text to say, "... revealed the presence of endothelial cells amongst the myocardial regions of all organoids" to reflect this observation more accurately.

25. In Fig.6a, the authors state, "gives rise to a well-developed secondary chamber", just because of the increase in the area? This appears to be an overstatement. The chambers are not properly characterized by the relevant markers. IRX3/4/HEY2 and NR2F2/HEY1 are standard TF markers in the field for V and A, respectively. These are completely ignored in the study. Similarly, the "compacted" myocardial muscle walls are claimed, but there is no data to support it with specific markers.

We thank the reviewer for this comment, and we agree that chamber identity could be further explored. We updated the text to reflect our observations more faithfully, and we say "... organoids in all conditions possessed two distinct 'chambers' marked via WT1+ and TNNT2+ cells. TNNT2+ cells were densely packed and formed a thick myocardial wall in the lower chamber, while also present in the upper region in a less dense arrangement directly underneath WT1+ cells. ... We found no difference in TNNT2+ chamber area in MM and EMM1 organoids relative to Control but found that EMM2/1 organoids display a 1.54-fold increased area relative to Control. Additionally, we found no difference in WT1+ chamber area in MM organoids relative to Control, whereas EMM1 and EMM2/1 organoids displayed increased areas (fold change) of 1.77 and 1.98, respectively." As suggested, we have further explored the identity of chambers in our organoids with more well-characterized atrial and ventricular markers (**Fig. 6f-j** and **Extended Data Fig. 5**). In **Fig. 6f,g**, we show IF data that organoids in the EMM2/1 condition display superior separation of the atrial and ventricular chamber markers NR2F2 and MYL3 (supported by in vivo data in the main text). These results are repeated for three cell lines (**Extended Data Fig. 5**). Additionally, we provide scRNA-seq data for the ACM and VCM clusters (**Fig. 6h-j**) and show that ACMs display several hallmark chamber markers such as *NR2F2*, *TBX5*, *NPPA*, and *NR2F1* (references provided in main text). Meanwhile, VCMs display the chamber markers *MYL3*, *HEY2*, *IRX4*, and *HAND1*. This data combined with extensive immunofluorescence data supports the claim of the existence of two chambers with atrial and ventricular identities in the organoids. We did not intend to claim that the organoids possessed compacted myocardial tissue, which is a well-defined phenomenon in physiology, but rather that the walls themselves were distinct, thick, and chamber-like. We have thus updated the text and have deleted instances of the word "compact" or "compacted".

26. There is a question of whether the maturation protocols are making the organoids more

mature or just killing non-cardiomyocyte populations of cells, leading to an increase in the percentage of cardiomyocytes. The observation that the organoids get smaller during maturation raises concerns about the loss of important cell populations, such as endothelial cells, and suggests a need for further investigation.

We have addressed this concern in our previous publication (Lewis-Israeli et al, 2021, Nat Comms). Regardless, and to address this comment, we performed new experiments in this respect. We generated a genetic reporter iPSC line for apoptosis. This line fluoresces with GFP when the active form of caspase-3 (master regulator of apoptosis) is present (Flip-GFP, originally described by Zhang et al., 2019, J. Am. Chem. Soc). We show in **Extended Data Fig. 1** that there are no instances of noticeable apoptosis between timepoints of maturation or between maturation conditions from day 20 to day 30.

27. The sarcomere markers checked are the usual ones, but in terms of fold changes, it should be way higher – for instance, the KCNJ2 staining is not convincing (should have a way higher fold increase during maturation. Regarding metabolic maturation, the authors could show the protein levels of some of the cited metabolic markers, or at least compare their expression with GD45. Plus, some metabolic assays would have completed the picture and made it more informative. The T-tubules marker selected it's not the best choice. Further, 4/10 tubules per cell are irrelevant - it should be 1 per sarcomere.

We thank the reviewer for this comment and the valuable suggestions. The increases in KCNJ2 are reasonable in the context of development for the timepoints presented (based on comparing levels to human embryonic datasets mentioned before). We repeated this experiment and show that the increased levels of KCNJ2 for EMM2/1 organoids is reproducible across three independent plates of organoid involving 14 organoids per condition (**Fig. 5i, j**). Some of the metabolic comments were also raised by reviewer #1 (see comment #5) and are described above. To reiterate, we performed Seahorse assays (**Fig. 4f-i**) for oxidative maturation present in EMM2/1 organoids, and additional differential expression of metabolic genes (**Fig. 4j**), provided metabolic gene expression panels for both ACMs and VCMs (**Fig 4k**), show that PPARGC1A expression increases in EMM2/1 organoids relative to Control organoids for three different cell lines (**Extended Data Fig. 9f**), and show metabolic gene expression overlap across organoid and human cardiac embryonic datasets (**Supplementary Fig. 7**). We repeated t-tubule staining with an antibody for caveolin-3, as suggested by Reviewer 1 (see also comment #8 above and **Fig. 5g,h**). We show that Cav-3+ area increases in EMM2/1 organoids compared to Control organoids, and we have moved our WGA staining and analysis to **Supplementary Fig. 23**.

28. There are also overstatements regarding the functional data due to improper characterization of the action potentials (different subtype/maturation criteria missing) and the absence of statistics to show their prevalence in the organoids. Additionally, the use of calcium as the major readout without normalizing it by the number of cells in the recording region can lead to irrelevant results, as the density of cells in the region of recording can drastically affect the amplitude. Electrochemical data are really affected by how many CM there are, and therefore single-cell data are imperative. This raises questions about the reliability of the functional data and suggests a need for further investigation and clarification.

All calcium data represents individual cardiomyocyte data from n=12 organoids per condition across three independent experiments (**Fig. 5a-c**) using a super-resolution microscope at a 10 ms time resolution and a lateral resolution of ~300 nm. This is now clarified in the figure legend and in the methods. Since individual cardiomyocytes are quantified instead of taking an area measurement containing many cardiomyocytes, a normalization to the number of cells in the recording region is not necessary. We agree that single-cell data is imperative. We further show the reproducibility of these results in **Extended Data Fig. 4**, where we overlay calcium transients from individual cardiomyocytes from 6 independent organoids per condition.

29. The authors claim self-organization, but no data is really shown to demonstrate the process of cell sorting and patterning, and how it is potentially connected to in vivo developmental stages. The assembly of the heart tube definitely happens very differently.

We appreciate the reviewer's comment on this topic. We have shown and described extensively how the organoids develop by self-organization in Lewis-Israeli et al, 2021, Nat Comms. The purpose of the organoid platform is to provide a model to better understand how these developmental processes occur in humans, as such it is not a perfect model (no model is perfect could be argued), but still constitutes the first instance of this kind of advanced organization in the human heart achieved by bioengineering methods and pluripotent stem cells. We provide extensive data showing how the organoids are connected to in vivo developmental stages and how the model is relevant to a great extent in **Extended Data Fig. 2, 3** and **Supplementary Data Fig. 7**.

30. To show any benefits of the platform, a direct comparison to other maturation protocols is missing, as many use the exact same factors in their metabolic treatments. Also, no application of this system has been demonstrated, in line with the preliminary nature of the study.

We thank the reviewer for this important comment. It is difficult to compare other maturation protocols to our organoid maturation protocol since no other similar developmental system has been described to date. To demonstrate the application of our organoid platform, as discussed in response to Reviewers 1 and 2, we investigated the impact of ondansetron exposure during heart organoid development (see comment #6). Ondansetron is a drug prescribed off-label to pregnant women to treat nausea and vomiting but has been implicated in causing congenital heart defects and orofacial defects. However, the consensus in the field is divided as there have been minimal studies investigating the effects of ondansetron on the heart (see Fig. 8 in the manuscript).

31. The most interesting aspect and novel aspect is the endogenous anterior-posterior patterning shown at the later stages. However, there are severe doubts about whether this is actually showing the same process as in the heart tube. As mentioned before, tube formation and patterning happen much earlier, and the observed phenomenon could be a later patterning event. In any case, the observation would benefit from further characterization, blocking the endogenous RA signaling, using downstream RA signaling markers, etc.

We appreciate the constructive criticism and suggestions provided by the reviewer. This was a point raised by reviewer 1 and 2 too, please see comment #4. In summary, as suggested, we added DEAB to block RA production to heart organoids during the maturation period and observed a marked diminishment of patterning from organoids in the EMM2/1 condition (**Fig. 7h-**

j). We also observed that the addition of exogenous retinoic acid did not lead to noticeable differences in patterning, suggesting that EMM2/1 organoids are generating adequate levels of retinoic acid to direct patterning at an endogenous level. Furthermore, we repeated experiments described in **Fig. 7e,g** to show the reproducibility of these results. As described in our response to an above comment, we also show that the emergence of an ALDH1A2+TBX18+ pole in EMM2/1 organoids is reproduced across three different cell lines (**Extended Data Fig. 8**) and show that EMM2/1 organoids exhibit significantly increased *ALDH1A2* expression across organoids from the three different cell lines (**Extended Data Fig. 9e**).

Minor comments:

32. The introduction is, in general, well-written, and most of the relevant papers have been cited. However, the referencing format is inconsistent and has mistakes.

We have edited and revised the entire manuscript and have resolved all formatting mistakes.

33. Fig1b: image quality insufficient. Images don't support statement (lines 127-129, 'continue to grow until d30').

We have updated the image quality of Fig. 1b. Additionally, we have changed our statement to read, "... then began to condense and become more defined up to day 30 in all maturation conditions."

34. Fig1c: would be more informative to quantitatively support the statement that organoids are becoming more elliptical over time.

This figure now contains information on both short and long diameters as requested.

35. Fig1g - sarcomere length quantification: not clear which data points belong to the same cardioid/same sections. Not described sufficiently how data was analyzed. Length measurement depends a lot on the cutting angle of the sarcomere (the more oblique). Is the sarcomere length consistent within a fiber? Are the differences within a fiber larger than between conditions?

Data points are from 4 independent heart organoids across 16 total sections per condition. The methods section has been updated with relevant statistical and analytical information, and the figure legends have been updated with statistical information. The sarcomere length was consistent within a fiber, and the differences within a fiber were not different between conditions (data not shown).

36. In the comparison to the Human Cell Atlas there seems to be more of a selection of the organoid to VCM, which is not commented on but rather ignored. Also, some more interesting genes to address all of the cell types and compare them to the data set would be useful here.

Please see previous comments on the whole overhaul of all the scRNA-seq data above. We have performed an extensive overhaul of our scRNAseq analysis and our comparison to embryonic heart datasets, shown in **Extended Data Figs. 2 and 3** and in **Supplementary Fig. 7**. As

suggested, we have selected some interesting genes (which we also use as marker genes for our cell clusters) and address all cell types in comparison to the embryological datasets (**Extended Data Fig 3** and **Supplementary Fig. 7**).

37. Extended fig.2b wrong labeling of condition EMM2/1

Thank you for identifying this. This figure panel has been removed due to space constraints.

38, Figure titles are too generic both in the figures and in the text (ex. fig4).

Figure titles have been updated in both the figures and in the text.

39. Line 330 doesn't match with what is shown in fig.6a (focusing only on one condition but showing data for all of them).

This section has been updated to better describe all data shown.

40. Figure 5b,c: says the recordings are taken with and without isoproterenol, but not an indication of isoproterenol in the graph or indicated in the text.

This was a mistake and has been fixed.

41. Figure 1h, 4e: not clear what the values on the y-axis mean, is it the fold change?

Yes, the data is presented as \log_2 fold change. The figure legends have been updated to more clearly explain this.

42. There is potential significance in all aspects of this work, although the novelty is limited since the maturation treatment is known and applicability over other protocols has not been demonstrated. However, a bigger issue is the preliminary nature of the study, with basic statistics and biological replicates missing to support most of the statements robustly. Also, there are basic flaws in the developmental biology interpretation, which require a major rewrite of the paper.

Thank you for your comments. We would like to highlight that the developmental conditions described are new in many respects, although based on previous concepts as is usually the case. We have tweaked concentrations, time, application range and combination of all the different compounds while applying them to a wholly different system (heart organoid, vs. monolayer cultures), after a significantly different initial protocol compared to what has been described before and with a significant different goal in mind (developmental maturation vs adult maturation). In this revision, we have revised and addressed all the statistics and reproducibility issues described. Our protocol is currently remarkably robust and has been reproduced independently outside of our lab by a biotechnological company. Currently our statistics and biological experiments represent in most instances dozens of organoids per condition. We have also performed additional experiments to reinforce the biological significance of the findings (ondansetron), benchmarked them against human embryonic datasets with excellent similarity to our findings, and provide

additional evidence of post heart tube-like patterning. Thus, the manuscript is significantly revamped and improved and we hope the reviewer will see value for the scientific community in the reported findings.

REVIEWER COMMENTS

Reviewer #1 (Remarks to the Author):

The authors provided in the revised paper extensive new data. Especially, the new functional seahorse data, and the experiments treating the organoids with an retinoic acid inhibitor and Ondesantrol adds functional relevance to the description of this model and broadens the methodology used.

Major points:

1) Still, the manuscript strongly relies on imaging as methodology providing not only qualitative, but also quantitative data on which conclusions are taken. As such, this must be executed with high quality, and it is necessary to improve the evaluation of this data: It is unclear across all figure, exemplary see figure 6 a,c,e, how the area of positive cells and the chamber size was quantified. It would strongly improve the data, if the existing confocal images would be evaluated in 3D to measure the chamber volume and the area of positive cells across all planes). If this is not possible, it will be mandatory to provide a supplemental figure describing in detail the methodology of evaluation. Please clearly state how the plane that was measured was selected. In addition, it is mandatory to provide some examples comparing 3D evaluation with 2D for the most important experiments or a blinded evaluations to exclude experimental bias. Again, this is only the minimum, it would be much better to invest into 3D evaluation across all experiments.

2) The authors have nicely shown that EMM2/1 organoids form their specific pattern reproducibly and across different cell lines. However, it is not clear how high the efficiency of the formation of the chambered pattern of EMM2/1 organoids is. If you randomly take 100 organoids from the EMM2/1 differentiation, how many organoids will show the chambered pattern? Please provide this information for all cell lines used.

3) The authors claim that the proepicardial region was located above the atrial chamber. However, no co-stainings for epicardial/atrial markers and epicardial/ventricular markers were provided to prove this statement. Please either remove this statement or provide the data.

4) While staining for NR2F2 and MYL3 suggested the formation of a ventricular and atrial chamber, the staining for MYL2 and MYL7 (Fig. 6) does not show this clear separation. Notably, MYL2 and MYL7 are not appropriate markers for atrial and ventricular cells, especially not in in vitro systems, since expression of these genes is dynamic. MYL7 is expressed by early cardiomyocytes of both atrial and ventricular subtype, while MYL2 is restricted to more mature ventricular cells. Thus, the authors should not draw conclusions based on these markers. This applies specifically to the ondansetron experiments: A reduction of MYL2+ cells in treated organoids does not necessarily mean a loss of ventricular cells but might mean a decreased maturation state compared to controls. The authors should repeat the experiments and stain for NR2F2 and MYL3 or reframe the conclusion into: effect on ventricular differentiation or maturation, as the data does not support the conclusion of an effect on ventricular differentiation.

5) Why do the untreated EMM2/1 organoids (Fig. 8) show such a low amount of MYL2+ cells compared to the same organoids in Fig. 6d,e?

6) The authors write: "We applied ondansetron at three different concentrations [...] and noticed stark differences in beating behavior [...], with organoids in the 10 μ M and 100 μ M conditions exhibiting a marked decrease in contraction force and beating frequency (Supplementary Videos 17-20)." Please quantify these differences and provide statistics or change conclusion.

Minor points:

1) The authors write: "TNNT2+ cells were densely packed and formed a thick myocardial wall in the lower chamber, while also present in the upper region in a less dense arrangement directly underneath WT1+ cells. In EMM2/1, WT1+ cells were found densely covering the outer surface of the upper region." This is not easily spottable in the provided pictures. Could you indicate this with arrows or provide zoomed-in pictures?

2) The authors write: "While MM organoids displayed a single internal chamber, organoids grown in the control, EMM1 and EMM2/1 conditions possessed multiple, smaller, interconnected chambers. control and EMM2/1 organoids possessed chambers throughout the bulk of the organoids while EMM1 organoids showed chambers predominantly towards one side of the organoid. These data confirmed the formation of well-established cardiac chambers and further supported our observations on the effects of developmental induction conditions." While the formation of one chamber through fusion of many smaller chambers is known from many other comparable in vitro models, this does not reflect the in vivo development. It might be worth to rather discuss this as limitation than indicating this is supporting the biological relevance.

3) While the Ondesansron experiment is very helpful to indicate the biological relevance of the in vitro model discussed in this paper, it might be helpful for the readers to relate this finding to previous existing findings by citing other, although less complex cardiac models that studied the effect of this drug.

Reviewer #2 (Remarks to the Author):

As other reviewers have also pointed out, this paper raises the issue of reproducibility in organoid generation. Single-cell RNA-seq analysis for each organoid is crucial to clarifying the evidence to that point. I pointed this out in my previous comment, asking for the acquisition of single-cell RNA-seq data for each condition and for comparison, but the authors were unable to do this. However, the authors perform experiments to confirm reproducibility by other methods (Extended Data Fig. 5, 8, and 9); therefore, it is considered that reproducibility at a certain level has been confirmed.

I would like to request one additional correction.

My previous comment

Figure 6 shows the emergence of WT1+TNNT2+ population and the differences in gene expression of atrial (MYL7) and ventricular (MYL2) cardiomyocytes. Extended Data Figure 4 shows the loss of PECAM1+ endothelial cells in the EMM1 and EMM2/1 conditions. Figure 7c shows the ALDH1A2+TBX18+ cardiomyocytes in the EMM2/1 condition. How are these represented by single-cell RNA-seq?

The authors' response

We thank the reviewer for this important comment. We integrated the data from each condition and now show all gene expression discussed in this comment (WT1, TNNT2, MYL7, MYL2, PECAM1, ALDH1A2, TBX18) as well as many additional genes of interest in figures throughout the paper, such as Fig. 2d, Fig. 3a, Fig. 6h-j, and Fig. 7d, among others.

My additional comment

These data does not quantitatively show the specific findings such as the emergence of WT1+TNNT2+ population, the loss of PECAM1+ endothelial cells in the EMM1 and EMM2/1 conditions, and the emergence of ALDH1A2+TBX18+ cardiomyocytes in the EMM2/1 condition. The authors should quantitatively confirm these findings for example using violin plot after extracting the specific cell populations from single-cell RNA-seq data.

Reviewer #3 (Remarks to the Author):

Response to author revision

Major comments

1. In our opinion, it is still unclear what the authors consider to be technical and what a biological replicate is, which is very clear in the field. Technical replicates are organoids derived from the same batch of cells. Biological from different cell batches distinguished by passage number. From the figure

legends, it appears as though the authors consider two organoids in the same batch as biological replicates because they were generated “independently”. They state they repeated results in other cell lines in the rebuttal. But even now, in the text, we find “6 independent organoids per condition” – e.g., Fig 4b. The authors should clearly indicate from how many batches (same cell line, different passages) or cell lines these specific data points come from. In the Methods, this is clarified, but the manuscript would benefit from stating it clearly in the figure legends, e.g., 6 organoids from 3 independent batches (or biological replicates).

2. The authors constantly use mean +/- SEM -, which we think can be misleading to show the real variation. The use of SD would be more suitable. Fig1b/d shows a generally rather large morphological variability of roughly 0.5 to 1 mm², which is important.

3. In Fig 1e: % Organoids beating is an odd metric to use because with current protocols, anything that doesn't show almost 100% beating would be unusual. So, it is kind of useless info – or at least it should be part of the supplement.

4. Fig 1g: Quantification shows quite a large variability, especially in MM and condition. And in the EMM2/1, there is no difference to the control conditions. It is hard to make conclusions based on that.

5. Figure 1h: loss of MYH7 (while simultaneously showing an increase in MYL2) is not in line with ventricular specification but could rather be a sign of rather more random differentiation (not just fine-tuning of expression). Also, there is no scRNA-seq plot with MYL2 expression, which is the key ventricular-specific functional marker.

6. Figure 2a: Stromal cells seem to be a catch-all term for all sorts of mesendodermal cells and the term is not very precise. The endothelial population is minimal and likely not consistent with real heart development. Additionally, separating valvular cells from stromal cells is difficult as they share many markers. Only determination of functional properties such as differentiation potential could substantiate this conclusion? Or ICC staining? (How do these cells compare to, e.g. Neri et al. (<https://doi.org/10.1038/s41467-019-09459-5>)? The used markers for SC and VC overlap (e.g., SOX9, CD24, ID2, UGDH, and FLRT2 all seem to mark the same stromal/mesodermal population). Epicardial cells are also minimal. Claiming all these cell types based on some rather unspecific markers, minimal contribution, and no functional data is a clear overstatement.

7. The same applies to Conductance Cells, which show mostly neural markers. Where do these cells come from? However, given that this is an EB-type differentiation, it is equally likely that comes from random ectoderm/neuronal differentiation.

8. Figure 3: It's really hard to look at the markers used because of the quality of the image.

9. Figure 4k: Please show the min and max fold change values (not just “min” or “max”)

10. Figure 5e: The authors show SCN5A but the far better determinant for maturation is the 6A vs. 6B ratio (see Camprostrini et al.).

11. Figure 6a-c: To us, it looks like WT1 is expressed in a separate cell mass, and there is no real engulfing. Also, the epicardial cell mass (presumably the PEO) is much bigger in 6a than in 6d – inconsistent? In 6d, the MYL2 areas don't seem to have a separate chamber from the atrial ones, but there seems to be a chamber with regions that are more MYL2 and MYL7 positive. However, in 6f, that chamber is mostly only visible in a MYL3 positive section (also - why the switch to MYL3??? Is MYL2 not visible anymore?).

Also: MYL2/MYL7 is not really an anterior/posterior patterning marker. What if the ventricular part just forms spontaneously? What would happen if the structures just grow longer? Would the entire organoid become MYL2 positive? If so - then it would mean there's just a developmental timing delay, but it doesn't prove multiple heart fields or AP patterning.

12. Figure 7h and i vs. j: The authors use MYL3 again (instead of MYL2), and it is unclear why the ventricular portion of the organoid would not change upon RA treatment. Similarly, the Atrial portion doesn't seem to change upon RA treatment. RA is quite potent, so this is difficult to explain by the organoid dealing only with an inherent RA gradient.

13. Figure 8 – Please switch to MYL2 again (why? And the MYL2 region is now smaller).

14. Extended Data 7: We cannot tell how the ECs are really positioned in relation to CMs (other than the fact that they are in contact with CMs somehow).

15. Extended Data 9: Again, MYH7 drops.

16. Extended Data 10: Just by looking at apoptosis after ondansetron at a single time point at day 30, we cannot say if apoptosis didn't contribute to the decline in function previously (at earlier time points). There seems to be a low-level signal (and either way stronger than the signal presented in Extended Data Fig 1).

REBUTTAL LETTER

We would like to again thank the reviewers for their expertise, time, and constructive criticism. We have reviewed all additional comments and have taken them into deep consideration and present to you an improved manuscript that addresses many of the concerns raised.

Note: Original reviewers' comments are presented in black, while our responses are written underneath in blue highlight.

Temporary Color legend:

Blue = Rebuttal

Yellow Highlight = NEW FIGURE

Reviewer	#1	Comments:
-----------------	-----------	------------------

		1. The authors provided in the revised paper extensive new data. Especially, the new functional seahorse data, and the experiments treating the organoids with an retinoic acid inhibitor and Ondesantron adds functional relevance to the description of this model and broadens the methodology used.
--	--	---

We thank the reviewer for the constructive feedback.

		2. Still, the manuscript strongly relies on imaging as methodology providing not only qualitative, but also quantitative data on which conclusions are taken. As such, this must be executed with high quality, and it is necessary to improve the evaluation of this data: It is unclear across all figure, exemplary see figure 6 a,c,e, how the area of positive cells and the chamber size was quantified. It would strongly improve the data, if the existing confocal images would be evaluated in 3D to measure the chamber volume and the area of positive cells across all planes). If this is not possible, it will be mandatory to provide a supplemental figure describing in detail the methodology of evaluation. Please clearly state how the plane that was measured was selected. In addition, it is mandatory to provide some examples comparing 3D evaluation with 2D for the most important experiments or a blinded evaluations to exclude experimental bias. Again, this is only the minimum, it would be much better to invest into 3D evaluation across all experiments.
--	--	--

We thank the reviewer for highlighting these critical details. Indeed, it is very important to include a detailed evaluation on the details of quantification of these data. We have updated the Materials & Methods under the "Confocal Microscopy and Image Analysis" section to clarify and better reflect our analysis pipeline within ImageJ, providing details on functions and methods for all analyses, including the mentioned quantification process for **Figure 6a,c,e**. Additionally, as requested, we have provided a supplemental figure describing in detail the methodology of evaluation for **Figure 6a,b,c,e** in **Supplementary Figure 31**. As is now clearly stated in the Materials & Methods, the measured plane to be selected was selected to be 50% of the organoid depth to allow for greater consistency in measurement and quantification.

3. The authors have nicely shown that EMM2/1 organoids form their specific pattern reproducibly and across different cell lines. However, it is not clear how high the efficiency of the formation of the chambered pattern of EMM2/1 organoids is. If you randomly take 100 organoids from the EMM2/1 differentiation, how many organoids will show the chambered pattern? Please provide this information for all cell lines used.

We thank the reviewer for their positive and constructive feedback. We have provided the requested data that show quantifications for how many EMM2/1 organoids display chambered patterning for each cell line used in Supplementary Data Fig. 28.

4. The authors claim that the proepicardial region was located above the atrial chamber. However, no co-stainings for epicardial/atrial markers and epicardial/ventricular markers were provided to prove this statement. Please either remove this statement or provide the data.

We thank the reviewer for this comment. As requested, we have provided co-stainings for epicardial/ventricular markers that are now presented in Fig. 7h. Combined with other data presented in the manuscript, these data demonstrate that the proepicardial region is overlapping with the atrial chamber (as both are opposite that of the ventricular chamber). We have moved the panels containing ALDH1A2 and TBX18 immunostaining in the MM and EMM1 conditions to Supplementary Fig. 30. These data remain quantified and presented in Fig. 7g.

5. While staining for NR2F2 and MYL3 suggested the formation of a ventricular and atrial chamber, the staining for MYL2 and MYL7 (Fig. 6) does not show this clear separation. Notably, MYL2 and MYL7 are not appropriate markers for atrial and ventricular cells, especially not in in vitro systems, since expression of these genes is dynamic. MYL7 is expressed by early cardiomyocytes of both atrial and ventricular subtype, while MYL2 is restricted to more mature ventricular cells. Thus, the authors should not draw conclusions based on these markers. This applies specifically to the ondansetron experiments: A reduction of MYL2+ cells in treated organoids does not necessarily mean a loss of ventricular cells but might mean a decreased maturation state compared to controls. The authors should repeat the experiments and stain for NR2F2 and MYL3 or reframe the conclusion into: effect on ventricular differentiation or maturation, as the data does not support the conclusion of an effect on ventricular differentiation.

We greatly thank the reviewer for their valuable expertise and feedback regarding this topic, and we agree that our results do not allow to conclude whether ventricular cells are maturing less or not differentiating. To correct this, we have rephrased the conclusions regarding MYL2/MYL7 in Figure 6 and have reframed our conclusions regarding ventricular differentiation/maturation for Figure 8 in the manuscript to make it clear that we just observe a reduction in ventricular markers expression.

6. Why do the untreated EMM2/1 organoids (Fig. 8) show such a low amount of MYL2+ cells compared to the same organoids in Fig. 6d,e?

We thank the reviewer for addressing this point. The reason for this inconsistency was technical and due to the fact that the images were captured and processed at very different times (some more than a year ago) by different people using different settings in the analysis software. We have re-analyzed the images and data presented in **Fig. 8a,b,c** with the same identical settings now to be 100% consistent with the intensity values utilized in **Fig. 6d,e**.

7. The authors write: “We applied ondansetron at three different concentrations [...] and noticed stark differences in beating behavior [...], with organoids in the 10 μ M and 100 μ M conditions exhibiting a marked decrease in contraction force and beating frequency (Supplementary Videos 17-20).” Please quantify these differences and provide statistics or change conclusion.

We thank the reviewer for highlighting this oversight. We have changed the conclusion to remove the statement regarding contraction force.

Minor points:

8. The authors write: “TNNT2+ cells were densely packed and formed a thick myocardial wall in the lower chamber, while also present in the upper region in a less dense arrangement directly underneath WT1+ cells. In EMM2/1, WT1+ cells were found densely covering the outer surface of the upper region.” This is not easily spottable in the provided pictures. Could you indicate this with arrows or provide zoomed-in pictures?

We thank the reviewer for the constructive feedback. We have updated **Figure 6a** to include arrows and zoomed-in pictures for the TNNT2+ and WT1+ cells, respectively, hopefully improving visibility.

9. The authors write: “While MM organoids displayed a single internal 422 chamber, organoids grown in the control, EMM1 and EMM2/1 conditions possessed multiple, smaller, 423 interconnected chambers. control and EMM2/1 organoids possessed chambers throughout the bulk of 424 the organoids while EMM1 organoids showed chambers predominantly towards one side of the 425 organoid. These data confirmed the formation of well-established cardiac chambers and further supported our observations on the effects of developmental induction conditions.” While the formation of one chamber through fusion of many smaller chambers is known from many other comparable in vitro models, this does not reflect the in vivo development. It might be worth to rather discuss this as limitation than indicating this is supporting the biological relevance.

We thank the reviewer for this comment and agree with their opinion. Our intention in this part of the text was to simply describe the process by which chambers seem to form, while trying to stay away from making biological interpretations of this phenomenon since we agree that it could be a limitation. We have now edited the text in this area to clarify this aspect.

10. While the Ondesatron experiment is very helpful to indicate the biological relevance of the in vitro model discussed in this paper, it might be helpful for the readers to relate this finding to

previous existing findings by citing other, although less complex cardiac models that studied the effect of this drug.

We thank the reviewer for the positive feedback and for the suggestions on this front. We agree that the inclusion of previous existing findings would be helpful to the readers. We have added further references and explanation in the text for related studies, both in vitro and in vivo (Danielsson et al., 2018; Frommeyer et al., 2017; Williams et al., 1991; *Blinova et al., 2018; Lu et al., 2019; Leow et al., 2023; Patel et al., 2019*).

Reviewer	#2	Comments:
-----------------	-----------	------------------

11. As other reviewers have also pointed out, this paper raises the issue of reproducibility in organoid generation. Single-cell RNA-seq analysis for each organoid is crucial to clarifying the evidence to that point. I pointed this out in my previous comment, asking for the acquisition of single-cell RNA-seq data for each condition and for comparison, but the authors were unable to do this. However, the authors perform experiments to confirm reproducibility by other methods (Extended Data Fig. 5, 8, and 9); therefore, it is considered that reproducibility at a certain level has been confirmed.

We thank the reviewer for their constructive feedback, and we provide additional data for the requested correction below.

12a. I would like to request one additional correction. My previous comment Figure 6 shows the emergence of WT1+TNNT2+ population and the differences in gene expression of atrial (MYL7) and ventricular (MYL2) cardiomyocytes. Extended Data Figure 4 shows the loss of PECAM1+ endothelial cells in the EMM1 and EMM2/1 conditions. Figure 7c shows the ALDH1A2+TBX18+ cardiomyocytes in the EMM2/1 condition. How are these represented by single-cell RNA-seq?

We thank the reviewer for this comment. We would like to clarify two points of this comment before addressing in more directly below. **Figure 6** does not show the emergence of a double positive WT1+TNNT2+ population, but rather two distinct populations (one is WT1+ (epicardial cells) and one is TNNT2+ (cardiomyocytes)). **Figure 7c** does not show ALDH1A2+TBX18+ cardiomyocytes, but rather shows ALDH1A2+TBX18+ epicardial cells (as TBX18 is an epicardial/proepicardial marker as shown in the scRNA-seq data (**Fig. 2d** and **Fig. 3a**)).

12b. The authors' response: we thank the reviewer for this important comment. We integrated the data from each condition and now show all gene expression discussed in this comment (WT1, TNNT2, MYL7, MYL2, PECAM1, ALDH1A2, TBX18) as well as many additional genes of interest in figures throughout the paper, such as Fig. 2d, Fig. 3a, Fig. 6h-j, and Fig. 7d, among others. These data do not quantitatively show the specific findings such as the emergence of WT1+TNNT2+ population, the loss of PECAM1+ endothelial cells in the EMM1 and EMM2/1 conditions, and the emergence of ALDH1A2+TBX18+ cardiomyocytes in the EMM2/1 condition.

The authors should quantitatively confirm these findings, for example using violin plot after extracting the specific cell populations from single-cell RNA-seq data.

We thank the reviewer for this comment. We have now generated violin plots as suggested by the reviewer for the requested genes in each cluster of cells identified in the scRNA-seq data, which are now displayed in **Supplementary Fig. 8**.

Reviewer	#3	Comments:
-----------------	-----------	------------------

13. In our opinion, it is still unclear what the authors consider to be technical and what a biological replicate is, which is very clear in the field. Technical replicates are organoids derived from the same batch of cells. Biological from different cell batches distinguished by passage number. From the figure legends, it appears as though the authors consider two organoids in the same batch as biological replicates because they were generated “independently”. They state they repeated results in other cell lines in the rebuttal. But even now, in the text, we find “6 independent organoids per condition” – e.g., Fig 4b. The authors should clearly indicate from how many batches (same cell line, different passages) or cell lines these specific data points come from. In the Methods, this is clarified, but the manuscript would benefit from stating it clearly in the figure legends, e.g., 6 organoids from 3 independent batches (or biological replicates).

We thank the reviewer for discussing this important topic. We have extensively edited all figure legends to clearly state the respective information (organoids, batches, etc.).

14. The authors constantly use mean +/- SEM –, which we think can be misleading to show the real variation. The use of SD would be more suitable. Fig1b/d shows a generally rather large morphological variability of roughly 0.5 to 1 mm², which is important.

We thank the reviewer for this comment. SEM is used rather than SD when one has a significant number of multiple independent experiments (versus replicates or small n, such as 3), and given the amount of independent organoids across multiple batches in each experiment in our figures (information for all data and figures is now present in figure legends) we judged SEM is the more appropriate statistical method to represent the data. This is standard practice in the field and in this journal too. If the editor judges that this methodology is inappropriate we will be happy to change all figures to SD. Additionally, the morphological variability in **Fig.1 b,d** is ~0.6 to 0.9 mm², all data points are presented on the graph. Additionally, SEM/SD is not utilized in Violin plots, as the quartiles are shown.

15. In Fig 1e: % Organoids beating is an odd metric to use because with current protocols, anything that doesn't show almost 100% beating would be unusual. So, it is kind of useless info – or at least it should be part of the supplement.

We thank the reviewer for this comment. The beating data shows the efficiency and reproducibility of the presented methodology, and serves as a continuous benchmark for organoid quality, as such we believe this is a key aspect of our manuscript and our methods. Human heart organoid protocols vary significantly in differentiation efficiency across reports (Feng et al, *Communications Biology* 2022; Mendjan et al, *Cell* 2021; Kim et al, *Cell Stem Cell* 2021), and reproducibility is a significant challenge, so we believe every effort to address these issues is necessary and constitutes valuable data.

16. Fig 1g: Quantification shows quite a large variability, especially in MM and condition. And in the EMM2/1, there is no difference to the control conditions. It is hard to make conclusions based on that.

We thank the reviewer for the feedback on this data. We have re-analyzed the data using a more appropriate modified statistical test (Tukey's multiple comparisons) to achieve a more concrete conclusion on this dataset. This figure is still **Figure 1g**.

17. Figure 1h: loss of MYH7 (while simultaneously showing an increase in MYL2) is not in line with ventricular specification but could rather be a sign of rather more random differentiation (not just fine-tuning of expression). Also, there is no scRNA-seq plot with MYL2 expression, which is the key ventricular-specific functional marker.

We thank the reviewer for the constructive feedback. It is important to note that there is no loss of MYH7. Data in figure 1h are expressed relative to control day 20, and so this graph only points to a decreased rate of MYH7 transcript production when compared to control day 20, still plenty of MYH7 transcript is produced. Why the rate is reduced compared to control is unknown to us (although all conditions show the same tendency and this might reflect a more finely tuned developmentally relevant progression). There is however overwhelming data in the manuscript showing increased maturation over time, including abundant cardiac specific cytoskeletal protein production. We have added *MYL2* scRNA-seq data in **Figure 3a** as requested.

18. Figure 2a: Stromal cells seem to be a catch-all term for all sorts of mesendodermal cells and the term is not very precise. The endothelial population is minimal and likely not consistent with real heart development. Additionally, separating valvular cells from stromal cells is difficult as they share many markers. Only determination of functional properties such as differentiation potential could substantiate this conclusion? Or ICC staining? (How do these cells compare to, e.g. Neri et al. (<https://doi.org/10.1038/s41467-019-09459-5>)? The used markers for SC and VC overlap (e.g., SOX9, CD24, ID2, UGDH, and FLRT2 all seem to mark the same stromal/mesodermal population). Epicardial cells are also minimal. Claiming all these cell types based on some rather unspecific markers, minimal contribution, and no functional data is a clear overstatement.

We thank the reviewer for their comments. We use the term stromal cells to identify a population of cells that exhibit mesenchymal characteristics but express no clearly identifiable markers of a specific population, and thus are very hard to define. In the literature this and similar terms are used when referring to these cells, and it is consistent with other reports that genes may be shared across multiple populations. Endothelial cells in our organoids constitute ~1% of the total

cell population, which is consistent with the recent report by Asp et al., 2019 regarding human heart development at a matched time-point, showing a combined makeup of 1% capillary endothelial cells/pericytes/adventitia. Epicardial cells in our organoids constitute between 1-4% of total cells per condition, which is consistent with Asp et al., 2019 that shows 3.8% epicardial cells in developing human hearts at a similar stage to our heart organoids. We agree that our valve cells could benefit from a greater degree of definition. We have found new markers for valve cells using data supported by Neri et al. 2019, Cheng et al. 2021, and Alfieri et al., 2010 and Dill et al., 2018 and have updated **Fig. 2d** accordingly to show the expression of *DLK1*, *ID2*, *DKK2*, and *WNT7B*, specific valve cell markers not widely shared with other mesenchymal populations in the heart. **Fig. 3a** has also been updated to include these genes.

19. The same applies to Conductance Cells, which show mostly neural markers. Where do these cells come from? However, given that this is an EB-type differentiation, it is equally likely that comes from random ectoderm/neuronal differentiation.

We thank the reviewer for the comment. We performed analysis of mRNA expression by RNA-seq of markers for mesoderm, ectoderm, and endoderm (Tsankov et al., 2015) in early organoid development in our model (**Supplementary Fig. 9**). The data shows that ectoderm differentiation and endoderm differentiation is not present in our heart organoid protocol. With classical neuronal differentiation in the heart organoids being absent, these data suggest that the conductance cell population emerges from the mesodermal cells and warrants further, future investigation.

20. Figure 3: It's really hard to look at the markers used because of the quality of the image.

We thank the reviewer for pointing out this issue. While the image quality is high resolution on our end and reads clearly, there might be a loss of quality due to compression when uploading files. We will be happy to discuss this with the editor if the problem persists.

21. Figure 4k: Please show the min and max fold change values (not just “min” or “max”)

The heatmaps in figure 4k are expressed as relative color schemes specific to each gene row. As such, the color is specific to each row and depicts the max and min values. This is a classical way of representing heatmaps commonly used in literature and a default method for many bioinformatic tools used in the field (e.g. Morpheus software). It would not be possible to add the specific min/max value for each and every gene across the whole heatmaps without cluttering the figure, but this data can be readily found in the scRNA-seq datasets. Furthermore, the data from **Figure 4k** is strongly supported by the extensive amount of other metabolic data presented in **Figure 4a-j**, **Extended Data Fig. 9f** and **Supplementary Fig. 10, 17-24**.

22. Figure 5e: The authors show SCN5A but the far better determinant for maturation is the 6A vs. 6B ratio (see Camprostrini et al.).

We thank the reviewer for sharing this information We agree that the ratio of SCN5A/SCN5B is a better marker for maturation in cardiomyocytes when it refers to the fetal/adult state and switch to adult isoforms, however our findings are completely contained into the context of early development and it is highly likely that the SCN5B switch has not happened yet. The utility of

SCN5A as a cardiac maturation marker has been shown extensively too (e.g. Funakoshi et al, Nature Communications, 2021).

23a. Figure 6a-c: To us, it looks like WT1 is expressed in a separate cell mass, and there is no real engulfing.

We thank the reviewer for the comment. In **Figure 6a**, TNNT2+ cells can be seen to be located underneath WT1+ cells in the interior panels. To more easily visualize this, we have supplied high magnification images to support these claims, presented in **Supplementary Fig. 27**, showing the engulfing of TNNT2+ cells by WT1+ cells.

23b. Also, the epicardial cell mass (presumably the PEO) is much bigger in 6a than in 6d – inconsistent?

Figure 6a assesses epicardial (WT1, a PEO marker) and cardiomyocyte (TNNT2) expression and localization while **Figure 6d** assesses atrial (MYL7) and ventricular (MYL2) expression and localization, not PEO, so there is no real inconsistency.

23c. In 6d, the MYL2 areas don't seem to have a separate chamber from the atrial ones, but there seems to be a chamber with regions that are more MYL2 and MYL7 positive.

We thank the reviewer for this comment. Yes, while MYL7 is used loosely as an atrial marker in the literature, its expression pattern in early cardiac development is shared to a certain extent between atrial and ventricular cells. This fact was also noted by Reviewer 1 (see comment #5). It makes sense that there is a MYL7+MYL2+ region since ventricular cells differentiated from atrial cells progressively. To more definitively assess chamber identity, **Fig. 6f** shows staining for MYL3 and NR2F2 (ventricular and atrial markers, respectively) for a more insight towards this phenomenon.

23d. However, in 6f, that chamber is mostly only visible in a MYL3 positive section (also - why the switch to MYL3??? Is MYL2 not visible anymore?).

Figure 6f assesses NR2F2 (atrial chamber identity marker) and MYL3 (ventricular chamber identity marker) expression and localization. The ventricular chamber and atrial chambers in **Figure 6f** are more apparent likely due to the more-restricted expression pattern between NR2F2 and MYL3 as being more specific chamber identity markers for atrial and ventricular chamber development. Also, please see reviewer #1 comment 5 in this respect.

23e. Also: MYL2/MYL7 is not really an anterior/posterior patterning marker. What if the ventricular part just forms spontaneously? What would happen if the structures just grow longer? Would the entire organoid become MYL2 positive? If so - then it would mean there's just a developmental timing delay, but it doesn't prove multiple heart fields or AP patterning.

While MYL2/MYL7 are not patterning markers per se, they label specific areas of the heart that can be used to infer a pattern. In conjunction with the other markers for the PEO, retinoic acid signaling (WT1, ALDH1A1) and other ventricular/atrial markers, and other experiments presented in the figures (such as RA inhibition) this provides a compelling framework for a patterned heart organoid. This data is supported by a host of additional data supportive of anterior/posterior patterning (**Figure 6 f-j, Figure 7, Extended Data Fig. 5, and Extended Data Fig. 8**). Further investigation of ventricular formation, growth, and expression pattern are excellent topics to explore in additional papers yet are outside the scope of this work. Data on the existence of multiple heart fields is provided in **Supplementary Fig. 4 and Supplementary Fig. 5** and is supported by multiple well-researched heart field markers.

24. Figure 7h and i vs. j: The authors use MYL3 again (instead of MYL2), and it is unclear why the ventricular portion of the organoid would not change upon RA treatment. Similarly, the Atrial portion doesn't seem to change upon RA treatment. RA is quite potent, so this is difficult to explain by the organoid dealing only with an inherent RA gradient.

MYL3 and MYL2 are both well-characterized ventricular markers and can be used interchangeably. In this case MYL3 is used because of compatibility with the NRF2F2 antibody for atrial labeling (which was requested by some of the other reviewers). The data in figure 7i-j shows a significant reduction in ventricular and atrial populations upon RA signaling inhibition. The RA+ condition, which contains exogenous RA plus endogenous RA produced by the organoid, does not show additional patterning or enlargement suggesting that the endogenous RA is sufficient to fully activate the pathway.

25. Figure 8 – Please switch to MYL2 again (why? And the MYL2 region is now smaller).

Figure 8 already shows MYL2, so we are not sure what this comment is referring to. In any case, both MYL2 and MYL3 are well-described ventricular markers, and commonly used proteins for ventricular fate (we assume the reviewer wants us to change this to MYL3?). For the “MYL2 region is now smaller” comment, please refer to Reviewer 1 comment #6 (shown below).

26. Extended Data 7: We cannot tell how the ECs are really positioned in relation to CMs (other than the fact that they are in contact with CMs somehow).

We thank the reviewer for this comment. We invite the reviewer to check **Supplementary Videos 1-4**, which show 3-dimensional relative positioning of ECs and CMs in high detail. ECs can be seen forming vessels and are inserted into myocardial tissue. Furthermore, this aspect was also extensively investigated in our previous publication (Lewis-Israeli et al., Nature Comms., 2021).

27. Extended Data 9: Again, MYH7 drops.

Please see comment #17.

29. Extended Data 10: Just by looking at apoptosis after ondansetron at a single time point at day 30, we cannot say if apoptosis didn't contribute to the decline in function previously (at earlier time points). There seems to be a low-level signal (and either way stronger than the signal presented in Extended Data Fig 1).

We thank the reviewer for addressing this point, and we agree that this analysis could benefit from additional, earlier timepoints. We have included day 15 fluorescence data into this analysis and is included in **Extended Data Fig. 10b**. The fluorescence levels between Untreated, 1uM, 10uM, and 100uM were insignificant at both timepoints. We are using a highly sensitive iPSC reporter line for apoptosis (Zhang et al., J Am Chem Soc, 2019). Since the organoids are large and three-dimensional, a certain degree of light scattering is inevitable and will lead to unspecific background signal. We have re-calibrated the images in **Extended Data Fig. 10a** with the same settings used in **Extended Data Fig. 1a** to make images comparable.

REVIEWERS' COMMENTS

Reviewer #1 (Remarks to the Author):

The authors addressed the concerns of the reviewer with additional data and evaluation. No more major points, congratulations to the authors for the beautiful work.

Minor point for consistency throughout the manuscript:

While the discussion relates ondansetron effects also possibly to maturation, the results still need similar adjustments in row 553/554:

"Together, this data suggests that ondansetron perturbs critical steps of ventricular heart development in heart organoids reminiscent of most clinical phenotypes associated with its use, namely, ventricular septal defects. " needs to be rephrased.

Reviewer #2 (Remarks to the Author):

The authors have appropriately responded to my previous concerns.

Reviewer #3 (Remarks to the Author):

After the second revision, the authors have, in our opinion, addressed most of our points.

FINAL RESPONSE TO REVIEWER'S COMMENTS

Reviewer #1 (Remarks to the Author):

The authors addressed the concerns of the reviewer with additional data and evaluation. No more major points, congratulations to the authors for the beautiful work.

Minor point for consistency throughout the manuscript:

While the discussion relates ondansetron effects also possibly to maturation, the results still need similar adjustments in row 553/554:

"Together, this data suggests that ondansetron perturbs critical steps of ventricular heart development in heart organoids reminiscent of most clinical phenotypes associated with its use, namely, ventricular septal defects. " needs to be rephrased.

We have addressed this point in the manuscript text as requested.

Reviewer #2 (Remarks to the Author):

The authors have appropriately responded to my previous concerns.

Reviewer #3 (Remarks to the Author):

After the second revision, the authors have, in our opinion, addressed most of our points.